# Hyperbolic Aware Minimization: Implicit Bias for Sparsity

**Tom Jacobs    Advait Gadhikar    Celia Rubio-Madrigal    Rebekka Burkholz**
CISPA Helmholtz Center for Information Security, Saarbrücken, Germany
{tom.jacobs, burkholz}@cispa.de

## Abstract

Understanding the implicit bias of optimization algorithms is key to explaining and improving the generalization of deep models. The hyperbolic implicit bias induced by pointwise overparameterization promotes sparsity, but also yields a small inverse Riemannian metric near zero, slowing down parameter movement and impeding meaningful parameter sign flips. To overcome this obstacle, we propose Hyperbolic Aware Minimization (HAM), which alternates a standard optimizer step with a lightweight hyperbolic mirror step. The mirror step incurs less compute and memory than pointwise overparameterization, reproduces its beneficial hyperbolic geometry for feature learning, and mitigates the small–inverse-metric bottleneck. Our characterization of the implicit bias in the context of underdetermined linear regression provides insights into the mechanism how HAM consistently increases performance —even in the case of dense training, as we demonstrate in experiments with standard vision benchmarks. HAM is especially effective in combination with different sparsification methods, advancing the state of the art.

## 1 Introduction

The success of modern deep learning relies on large amounts of overparameterization, which has led to a computationally demanding trend to increase the size of models, and thus the number of trainable parameters by orders of magnitude (Hoffmann et al., 2022; Kaplan et al., 2020). A common explanation for this phenomenon are implicit biases that originate from a combination of the optimizer and the overparameterization (Pesme et al., 2021; Gunasekar et al., 2017a; Woodworth et al., 2020), which regularize the training dynamics and thus improve the generalization performance.

Training sparse models instead leads to suboptimal performance (Li et al., 2017; Frankle & Carbin, 2018). This fact has limited pruning at initialization (PaI) approaches (Tanaka et al., 2020; Lee et al., 2019; Liu et al., 2021a) that aim to reduce the heavy computational and memory demands by masking the network before training the remaining parameters. In contrast, state-of-the-art sparsification methods utilize overparameterization in some capacity, as they either gradually prune parameters in Dense-to-Sparse (DtS) training (Peste et al., 2021; Kuznedelev et al., 2024; Kusupati et al., 2020; Jacobs & Burkholz, 2025; Kolb et al., 2025) or dynamically explore multiple sparse masks to find high-performing sparse networks with Dynamic Sparse Training (DST) (Evci et al., 2020; Lasby et al., 2023; Chen et al., 2021). Key observations regarding these algorithms are that a) mild sparsity (which does not degrade performance relative to a dense baseline) (Jin et al., 2022) and b) longer training with standard optimizers can improve generalization performance significantly. The latter indicates that sparse models are difficult to train and take longer to converge (Kuznedelev et al., 2023). Consequently, sparse training ideally leverages overparameterization to improve generalization.

A recent development to improve sparse training is the pointwise overparameterization proposed in PILoT (Jacobs & Burkholz, 2025) and Sign-In (Gadhikar et al., 2025). All parameter weights $\theta \in \mathbb{R}^n$ are replaced by a pointwise product of parameters $m \odot w$, with both $m, w \in \mathbb{R}^n$. This changes the implicit bias of the optimization process and leads to substantial generalization benefits for sparse training. In PILoT, a continuous sparsification method, the overparameterization is used to jointly learn the mask $m$. Meanwhile, the PaI method Sign-In uses it to increase the plasticity of non-masked parameters and facilitate sign flips, which was shown to be a major obstacle in sparse training (Gadhikar & Burkholz, 2024a). Both methods, PILoT and Sign-In , achieve state-of-the-art

|  | Implicit sparsity bias | Sign flips | No hard perturbations | No extra parameters |
|---|---|---|---|---|
| Dense training | ❌ | – | ✅ | ✅ |
| PILoT (Jacobs & Burkholz, 2025) | ✅ | ❌ | ✅ | ❌ |
| Sign-In (Gadhikar et al., 2025) | ✅ (less strong) | ✅ | ❌ | ❌ |
| HAM (ours) | ✅ (less strong) | ✅ | ✅ | ✅ |

Table 1: HAM induces a less strong implicit sparsity bias (moderating between $L_2$ and $L_1$) and flips parameter signs more easily due to its inverse metric (see Fig. 1), which together lead to boosting sparse training without explicit overparameterization.

results in their respective categories. However, on their own, they fall short of baseline methods of sparse training that do not utilize this form of overparameterization, such as AC/DC (Peste et al., 2021) and RiGL (Evci et al., 2020). To understand this gap, we investigate their training dynamics.

The dynamics of the overparameterization $\boldsymbol{m} \odot \boldsymbol{w}$ can be derived within the mirror flow (Li et al., 2022) or time-dependent mirror flow framework (Jacobs et al., 2025). It is associated with the hyperbolic mirror map (Woodworth et al., 2020) and, depending on initialization, learning rate, and regularization, it changes from an implicit $L_2$ (Dense) to $L_1$ (Sparse) bias during training. The mirror flow induced by $\boldsymbol{m} \odot \boldsymbol{w}$ can also be characterized by a Riemannian gradient flow with an associated metric (Li et al., 2022). Comparing these metrics highlights a problem: $\boldsymbol{m} \odot \boldsymbol{w}$ suffers from a small inverse metric $g^{-1}(\boldsymbol{\theta})$ near the origin, where $g$ is a Riemannian metric tensor (Jacobs & Burkholz, 2025) (see Fig. 1 and Appendix F). As a consequence, parameters can get stuck at 0, preventing effective sign-flips. Sign-In partially mitigates this issue by iteratively re-initializing $m$ and $w$ such that $m \odot w$ remains fixed. We set $\gamma := (m^2 - w^2)^2 >> 0$. However, the remedy is unstable and introduces a hard perturbation to the training dynamics, limiting its positive effects.

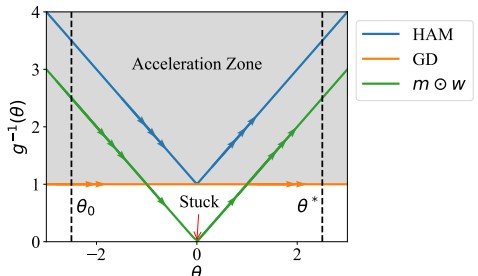

Figure 1: The inverse metric $g^{-1}(\boldsymbol{\theta})$ of HAM is above the one of gradient descent (GD), while the overparameterization $\boldsymbol{m} \odot \boldsymbol{w}$ is below for small $\boldsymbol{\gamma}$. This enables moving from the initialization $\boldsymbol{\theta}_0$ to the optimum $\boldsymbol{\theta}^*$ instead of getting stuck. Therefore, HAM fixes the vanishing inverse metric. Note the hyperbolic geometric structure of HAM and $\boldsymbol{m} \odot \boldsymbol{w}$ compared to the flatness of GD.

In this work, we propose to capture the essential structure of the two methods PILoT and Sign-In and thus the implicit bias of pointwise overparameterization $\boldsymbol{m} \odot \boldsymbol{w}$, which provably aids in finding generalizable sparse solutions. At the same time, we are able to avoid their drawbacks: their slow down near zero and their need for explicit overparameterization that negatively impacts memory and compute (see Table 1). We do this by deriving a plug-and-play hyperbolic optimization step, which we alternate with gradient descent or any other first-order optimizer. Our alternating method is called HAM: Hyperbolic Aware Minimization (§ 3, 3.1). HAM mitigates the small inverse metric problem of $\boldsymbol{m} \odot \boldsymbol{w}$ and keeps a similar but fully controllable implicit bias, as shown in § 4. We evaluate HAM on standard vision benchmarks and find that it consistently improves generalization, especially of sparse training (§ 5). Remarkably, HAM tends to enhance generalization complementary to sharpness aware minimization (SAM) (Foret et al., 2021), yet incurs only negligible computational overhead. These improvements can be explained by two major mechanisms: a) It accelerates training around 0, thus improving sign learning. This is facilitated by its geometry and a larger inverse metric. b) The implicit bias towards sparsity regularizes training inducing a mild sparsity. Both mechanisms boost generalization performance of sparsification techniques, such as AC/DC (Peste et al., 2021) and RiGL (Evci et al., 2020), and even of dense model training. In summary, our contributions are:

- We introduce HAM, a lightweight, plug-and-play general purpose optimization step that integrates with any optimizer at a negligible computational cost.

- We provide a theoretical analysis of HAM's training dynamics using Riemannian gradient flow for linear regression (§ 4), characterizing its implicit bias and sign-flipping mechanism (Appendix E).

- HAM inherits the geometric benefits of recent sparsity parameterizations while mitigating their vanishing inverse metric problem (see Figure 1 and Appendix F). The benefits are a implicit sparsity bias which facilitates a mild sparsity and complementary sign flips to those of dense training.

- Empirically, HAM improves state-of-the-art sparsity methods (AC/DC, RiGL, STR), enhances standard dense training, and is also compatible with optimizers like SAM.

## 2  RELATED WORK

**Sparsification**  Sparse training methods can be categorized into three broad classes: Pruning at Initialization (PaI), Dense-to-Sparse training (DtS), and Dynamic Sparse Training (DTS). PaI methods identify a sparse mask at initialization and train the remaining parameters to convergence. They include methods like SNIP (Lee et al., 2019), Synflow (Tanaka et al., 2020), NPB (Pham et al., 2023), PHEW (Patil & Dovrolis, 2021), GraSP (Wang et al., 2020) and random pruning (Liu et al., 2021a; Gadhikar et al., 2023). Their primary limitation is that standard optimizers do not find generalizable solutions on these fixed masks, as they struggle to effectively learn parameter signs (Gadhikar & Burkholz, 2024b; Gadhikar et al., 2025). In contrast, DtS methods learn the mask via a dense or denser phase of training, followed by any kind of pruning step and possibly more training. This includes iterative pruning methods like IMP (Frankle & Carbin, 2019), LRR (Renda et al., 2020; Han et al., 2015), AC/DC (Peste et al., 2021), CAP (Kuznedelev et al., 2024), and WoodFisher (Singh & Alistarh, 2020). Continuous sparsification methods, which start from a dense network and gradually sparsify it with a learnable mask, also fall under this category. They include PILoT (Jacobs & Burkholz, 2025), STR (Kusupati et al., 2020), CS (Savarese et al., 2021) and spred (Ziyin & Wang, 2022). The third class of methods, Dynamic Sparse Training, start from an already sparse mask but dynamically update it during training, and in this sense utilize a form of (dynamic) overparameterization (Liu et al., 2021b). Examples include RiGL (Evci et al., 2020), MEST (Yuan et al., 2021), and SET (Mocanu et al., 2018). While PaI methods cannot compete with the generalization performance of DtS and DST methods, Sign-In (Gadhikar et al., 2025) improves on PaI by using the pointwise overparameterization $m \odot w$, which leverages a hyperbolic mirror map to facilitate sign flips. In this work, we propose instead a simpler, more powerful hyperbolic optimization step to leverage a similar mirror map without doubling the number of parameters and solving an issue with an associated inverse metric.

**Implicit bias and mirror flow**  The implicit bias of neural networks is a well studied topic that aims to explain the regularization benefits resulting from overparameterization (Woodworth et al., 2020; Gunasekar et al., 2017b; 2018; Li et al., 2022). It is primarily characterized within the mirror flow framework, a well-established concept in convex optimization (Alvarez et al., 2004; Beck & Teboulle, 2003; Rockafellar & Fenchel, 1970; Boyd & Vandenberghe, 2009; Sun et al., 2022). A mirror flow can be seen as a gradient flow on a Riemannian manifold (Li et al., 2022; Alvarez et al., 2004) with the metric tensor being the Hessian of the Legendre function, which has also been extended to cover stochastic gradient descent (SGD) (Pesme et al., 2021; Even et al., 2023; Lyu & Zhu, 2023) and more recently to explicit regularization (Jacobs et al., 2025). This framework allows us to characterize the implicit bias. The main observation is that large learning rates, stochastic noise from SGD, and regularization can benefit generalization by implicitly inducing sparsity. However, overparameterization can also lead to small inverse metrics, slowing down convergence and potentially hampering generalization (Jacobs & Burkholz, 2025), which we can successfully avoid in HAM.

**Related optimizers**  The mirror flow framework also enables us to view our algorithm HAM through the lens of natural gradient descent. Accordingly, the inverse metric is adapted due to (an approximation of) the Fisher information matrix, which captures second-order information (Martens, 2014; Amari, 1999). A more general Bayesian framework (Khan & Rue, 2021) has been used to gain insights into invariant distributions by using Lie groups (Kıral et al., 2023) and to develop the IVON optimizer (Shen et al., 2024). Within this framework, HAM, our proposal that alternates exponential updates with gradient descent steps, can be interpreted as a mapping of the Fisher information (metric) to a known posterior distribution, as derived in Appendix C. Moreover, our proposed hyperbolic update is reminiscent of exponentiated gradient descent (Kivinen & Warmuth, 1994). Distinctly from HAM, it optimizes probability distributions and thus includes normalization to stay on a probability simplex, as seen, for example, in Chapter 7 of Vishnoi (2021). Exponential gradient descent, which has recently been applied to reweighting batches (Majidi et al., 2021) or

augmenting ADAM (Bernstein et al., 2020), also utilizes similar exponential updates but does not rely on an alternating scheme like HAM. This prevents it from facilitating more suitable sign flips than gradient descent. More advantages of HAM are analyzed in Appendix D and E.

**Two-step and alternating schemes**    Various previous works have explored alternating training schemes including proximal methods, soft thresholding, ADMM, alternating least squares, and expectation maximization, among others (Parikh & Boyd, 2014; Boyd et al., 2011; Cichocki et al., 2009; McLachlan & Krishnan, 1996) . The most related alternating algorithms to HAM are based on birth-death dynamics at a neuron level in two-layer neural network training (Rotskoff et al., 2019) or variational inference (Mielke & Zhu, 2025; Gladin et al., 2024; Yan et al., 2024). An important difference is that we work on a weight level while other approaches work on a neuron or distribution level and serve an entirely different purpose. Furthermore, one of the most well-known two-step approaches is sharpness aware minimization (SAM) (Foret et al., 2021), which promotes the search for flat solutions at the expense of almost doubling the compute of one optimization step. In contrast, HAM encourages an implicit sparsity bias and acceleration around 0, which are complementary mechanisms. Our experiments (Table 3) demonstrate that our proposed hyperbolic step can be effectively combined with SAM to further boost generalization.

## 3    MOTIVATION AND DERIVATION OF HAM

We derive our novel optimization step by building on insights from the implicit bias of recently developed sparse training methods. These methods have exploited a reparameterization of the neural network: They replace each weight $\boldsymbol{\theta}$ with a product of two weights $\boldsymbol{m} \odot \boldsymbol{w}$, where $\odot$ is the Hadamard product, i.e., a pointwise multiplication. For this reparameterization, it is known that stochastic noise and weight decay induce sparsity by an implicit $L_1$ penalty (Pesme et al., 2021; Jacobs & Burkholz, 2025). The next paragraphs restate the induced gradient flow of this reparameterization (where the learning rate $\eta \to 0$). Note that $\boldsymbol{u}^2$ abreviates $\boldsymbol{u}^{\odot 2}$.

**Gradient flow training**    Consider a continuously differentiable and L-smooth[1] loss function $f : \mathbb{R}^n \to \mathbb{R}$. It can be trained by means of gradient descent: $\boldsymbol{\theta}_{k+1} = \boldsymbol{\theta}_k - \eta \nabla f(\boldsymbol{\theta}_k)$, initialized at $\boldsymbol{\theta}_0 = \boldsymbol{\theta}_{init}$, where $\eta > 0$ is the learning rate. Taking $\eta \to 0$, we obtain its gradient flow: $d\boldsymbol{\theta}_t = -\nabla f(\boldsymbol{\theta}_t)dt$. Its integral form is used in the mirror flow analysis and descriptions of the implicit bias: $\boldsymbol{\theta}_t - \boldsymbol{\theta}_0 = -\int_0^t \nabla f(\boldsymbol{\theta}_s)ds$.

**Reparameterized gradient flow**    Li et al. (2022) derive a similar formulation for the reparameterization $\boldsymbol{m} \odot \boldsymbol{w}$ trained with gradient descent, while Theorem 2.1 in Jacobs & Burkholz (2025) integrates weight decay with strength $\beta$ in the analysis resulting of the following gradient flow:

$$\begin{cases} d\boldsymbol{m}_t = -\boldsymbol{w}_t \odot \nabla f(\boldsymbol{\theta}_t)dt - 2\beta \boldsymbol{m}_t dt, & \boldsymbol{w}_0 = \boldsymbol{w}_{\text{init}}, \\ d\boldsymbol{w}_t = -\boldsymbol{m}_t \odot \nabla f(\boldsymbol{\theta}_t)dt - 2\beta \boldsymbol{w}_t dt, & \boldsymbol{m}_0 = \boldsymbol{m}_{\text{init}}. \end{cases}$$

This corresponds to the integral equation for $\boldsymbol{\theta}_t = \boldsymbol{m}_t \odot \boldsymbol{w}_t$:

$$\boldsymbol{\theta}_t = \boldsymbol{u}_0^2 \odot \exp\left(-2\int_0^t \nabla f(\boldsymbol{\theta}_s)ds - 4\beta t\right) - \boldsymbol{v}_0^2 \odot \exp\left(2\int_0^t \nabla f(\boldsymbol{\theta}_s)ds - 4\beta t\right), \quad (1)$$

where $\boldsymbol{u}_0 := \frac{\boldsymbol{m}_0 + \boldsymbol{w}_0}{\sqrt{2}}$ and $\boldsymbol{v}_0 := \frac{\boldsymbol{m}_0 - \boldsymbol{w}_0}{\sqrt{2}}$ for $|\boldsymbol{w}_0| \le \boldsymbol{m}_0$ are chosen such that $\boldsymbol{u}_0^2 - \boldsymbol{v}_0^2 = \boldsymbol{\theta}_0$. $\beta > 0$ is the strength of weight decay. This results in a time varying Riemannian gradient flow for $\boldsymbol{\theta}_t$:

$$d\boldsymbol{\theta}_t = \sqrt{\boldsymbol{\theta}_t^2 + \boldsymbol{\gamma}_t^2} \odot \nabla f(\boldsymbol{\theta}_t)dt - 2\beta \boldsymbol{\theta}_t dt, \qquad \boldsymbol{\theta}_0 = \boldsymbol{\theta}_{\text{init}}, \quad (2)$$

where $\boldsymbol{\gamma}_t = 4\boldsymbol{u}_0^2 \odot \boldsymbol{v}_0^2 \exp(-4\beta t)$. Eq. (2) implies that we cannot move through zero when $\boldsymbol{\gamma}_t \to \boldsymbol{0}$.

**Exponential gradient descent**    The hyperbolic gradient flow in Eq. (1) not only corresponds to the gradient flow of $\boldsymbol{m} \odot \boldsymbol{w}$, but also to exponential gradient descent. This is presented by Wu & Rebeschini (2021) for matrix forms in a matrix sensing task without regularization ($\beta = 0$). We use this connection to derive an update without the need for the reparameterization (or regularization). The update is captured by the following theorem:

---

[1]See Definition A.1 in the appendix for a definition of L-smoothness.

**Theorem 3.1** *If $m_0 = \text{sign}(\boldsymbol{\theta}_0)w_0 = \sqrt{|\boldsymbol{\theta}_0|}$, then*

$$\boldsymbol{\theta}_{k+1} = \boldsymbol{\theta}_k \exp\left(-\eta\left(2\,\text{sign}(\boldsymbol{\theta}_k)\nabla f(\boldsymbol{\theta}_k) + 4\beta\right)\right) \tag{3}$$

*is equivalent to Eq. (1) up-to first order, i.e., the discretization error is $\mathcal{O}(\eta^2)$.*

*Proof.* See proof of Theorem B.1 in the appendix. □

Note that a more general update for a product of matrices is provided by Wu & Rebeschini (2021). Their exponential update suffers from the same problem as the parameterization $\boldsymbol{m} \odot \boldsymbol{w}$, as it corresponds to $\boldsymbol{\gamma} = \boldsymbol{0}$ and thus completely preventing sign flips (see Corollary B.2). Our proposal HAM overcomes this obstacle by alternating a gradient step with an exponential update step.

**Derivation of HAM**  The novelty stems from alternating the new hyperbolic update in Eq. (3) with another optimizer. This forms the basis of our proposal HAM. We derive its explicit form for gradient descent as follows:

$$\boldsymbol{\theta}_{k+\frac{1}{2}} = \boldsymbol{\theta}_k - \eta\nabla f(\boldsymbol{\theta}_k), \tag{GD}$$

$$\boldsymbol{\theta}_{k+1} = \boldsymbol{\theta}_{k+\frac{1}{2}} \odot \exp\left(-\eta\left(\alpha\,\text{sign}(\boldsymbol{\theta}_k)\nabla f(\boldsymbol{\theta}_k) + \beta\right)\right). \tag{HYP}$$

Note it is not necessary to use the same learning rate for gradient descent and for the exponential update. In fact, the learning rates control the strength of the implicit bias towards sparsity, for which we have introduced an additional hyperparameter $\alpha \in \mathbb{R}$. The exponential update now more closely resembles the hyperbolic gradient update in Eq. (1), as it can switch the sign in the exponential. § 4 studies the resulting gradient flow.

**Interpretation**  The exponential update (HYP) introduces a weight scaling which correspond to a metric $g(\boldsymbol{\theta}) = 1/|\boldsymbol{\theta}|$ for a Riemannian gradient flow, as we will see in §4. This changes how parameters evolve compared to standard gradient descent. When the sign of a parameter is correct, the update refines its magnitude. If the sign is incorrect, it drives the parameter exponentially fast toward zero. However, on its way to a sign flip, the parameter gets stuck in $0$ —because the parameter update is proportional to $\boldsymbol{\theta} = \boldsymbol{0}$ at the origin. To facilitate the sign flip, we need the intermediary gradient step (GD), which explains the advantages of HAM over pure exponential updates. In summary, our combined update can be interpreted as: *Learn the magnitude when the sign is correct; otherwise, move rapidly to zero to enable sign correction.* This mechanism is crucial for enabling sparse training (Gadhikar & Burkholz, 2024a; Gadhikar et al., 2025), as shown in our experiments (§ 5), where HAM significantly boosts performance compared to standard optimizers.

**Remark 3.2** *Note that the second step (HYP) depends both on $\boldsymbol{\theta}_k$ and $\boldsymbol{\theta}_{k+\frac{1}{2}}$, which would require twice the memory of gradient descent. To avoid this, we replace $\text{sign}(\boldsymbol{\theta}_k)$ with $\text{sign}(\boldsymbol{\theta}_{k+\frac{1}{2}})$. We restate the second HAM step actually deployed (HYP\*) in §3.1. It also has benefits for the optimization itself, promoting more stable sign flips, as we discuss in Appendix D and Figure 6. Building on recent work on sign flips (Gadhikar et al., 2025), we argue that the sign should be aligned with the gradient evaluation to assess whether the step should be accelerated or not, to promote meaningful sign flips. This insight is tightly linked to our choice of $\text{sign}(\boldsymbol{\theta}_{k+\frac{1}{2}})$ instead of $\text{sign}(\boldsymbol{\theta}_k)$ in the (HYP\*) step.*

### 3.1 ALGORITHM: HYPERBOLIC AWARE MINIMIZATION (HAM)

We propose HAM (Algorithm 1), which alternates between any standard optimizer step and a hyperbolic (signed) mirror map to improve the general trainability of neural networks. The proposed method is inspired by recent sparsification methods, as theoretically justified in § 4. We next state the main algorithmic innovations.

**Hyperbolic step**  Let $\eta > 0$ denote the learning rate and $\alpha, \beta \geq 0$ be positive constants. The hyperbolic step deployed with parameters $\boldsymbol{\theta}_k \in \mathbb{R}^n$ is given by

$$\boldsymbol{\theta}_k = \boldsymbol{\theta}_{k+\frac{1}{2}} \odot \exp\left(-\eta\left(\alpha\,\text{sign}(\boldsymbol{\theta}_{k+\frac{1}{2}})\nabla f(\boldsymbol{\theta}_k) + \beta\right)\right), \tag{HYP*}$$

where $\boldsymbol{\theta}_{k+\frac{1}{2}}$ is the step of any other optimizer. $\alpha$ controls the convergence speed and hyperbolic awareness of the method, and $\beta$ induces an explicit regularization similar to that of PILoT (Jacobs &

---

**Algorithm 1** HAM

---

**Require:** steps $T$, schedule $\eta$, initialization $\boldsymbol{\theta}_{init}$, constants $\alpha, \beta \geq 0$.
   **for** $k \in 0 \ldots T - 1$ **do**
      $\boldsymbol{\theta}_{k+\frac{1}{2}} = \text{OptimizerStep}(\nabla f(\boldsymbol{\theta}_k), \eta)$
      $\boldsymbol{\theta}_{k+1} = \text{HyperbolicStep}(\boldsymbol{\theta}_{k+\frac{1}{2}}, \nabla f(\boldsymbol{\theta}_k), \alpha, \beta, \eta)$ according to formula (HYP*)
   **end for**
   **return** Model weights $\boldsymbol{\theta}_T$

---

Burkholz, 2025). Note that we have replaced $\text{sign}(\boldsymbol{\theta}_k)$ with $\text{sign}(\boldsymbol{\theta}_{k+\frac{1}{2}})$, as mentioned in Remark 3.2. Our analysis of implicit bias § 4 still remains valid with this change, as we show in Appendix Theorem B.6.

**Memory and compute overhead**   HAM does not incur any memory overhead, as it reuses the known gradient and current signs of the weights. In contrast, the pointwise overparameterization $m \odot w$ doubles the number of parameters, which would only be negligible in case of large batch sizes —where activations dominate the memory requirements (Ziyin & Wang, 2022; Jacobs & Burkholz, 2025; Kolb et al., 2025). Moreover, the additional extra flops during training are negligible, as they are linear in the number of parameters.

## 4   THEORY: GRADIENT FLOW ANALYSIS

Our theory (Eqs. (GD,HYP)) identifies the implicit bias of HAM's Riemannian gradient flow in parameter space (Thm. 4.2) and provides a convergence analysis (Thms. 4.3 and 4.5). Accordingly, HAM solves the vanishing inverse metric problem of $m \odot w$, and thus converges faster, while retaining the same asymptotic implicit sparsity bias. In this section, we assume that the objective function $f : \mathbb{R}^n \to \mathbb{R}$ is continuously differentiable, i.e., $f \in C^1$, and $L-$smooth[1]. Appendix E proves that HAM also induces meaningful sign flips like $m \odot w$ (Gadhikar et al., 2025). We focus on the case $\beta = 0$ to simplify the exposition. Theorem 4.2 derives the flow for general $\beta$. The effects of nonzero $\beta$ are highlighted in Section B.1.

**Riemannian gradient flows**   In order to concisely study the behavior of HAM, we consider a gradient flow formulation. Gradient flow (flat) and $m \odot w$ (hyperbolic) flow can be described as Riemannian gradient flows depending on a general metric $g(\boldsymbol{\theta})$:

$$d\boldsymbol{\theta}_t = -g^{-1}(\boldsymbol{\theta}_t)\nabla f(\boldsymbol{\theta}_t)dt, \qquad \boldsymbol{\theta}_0 = \boldsymbol{\theta}_{init}.$$

We refer to the quantity $g^{-1}(\boldsymbol{\theta})$ as the inverse metric. It is trivial for gradient flow, since $g_{\text{GD}}^{-1}(\boldsymbol{\theta}) = 1$. For $m \odot w$, Jacobs & Burkholz (2025) have derived $g_{m \odot w}^{-1}(\boldsymbol{\theta}) = \sqrt{\boldsymbol{\theta}^2 + \boldsymbol{\gamma}^2}$, where $\boldsymbol{\gamma}$ depends on the initialization scale and can change due to noise and regularization. In contrast, the inverse metric of HAM is not changed by these factors. To give an overview, the inverse metrics are also reported in Table 2.

**The vanishing inverse metric problem**   For small $\boldsymbol{\gamma}$ and small weights $\boldsymbol{\theta}$, the inverse metric $\sqrt{\boldsymbol{\theta}^2 + \boldsymbol{\gamma}^2}$ of $m \odot w$ can get much smaller than 1 (see Figure 1). This implies that learning close to 0 is slowed down, which makes transitions through 0 (and sign flips of $\boldsymbol{\theta}$) much harder, slowing down convergence.

Table 2: Inverse metrics of gradient descent, the overparameterization $m \odot w$, and HAM.

|  | GD | $m \odot w$ | HAM |
|---|---|---|---|
| $g^{-1}(\boldsymbol{\theta})$ | 1 | $\sqrt{\boldsymbol{\theta}^2 + \boldsymbol{\gamma}^2}$ | $1 + \alpha\|\boldsymbol{\theta}\|$ |

**Theorem 4.1** *If $f$ is $L-$smooth (Definition A.1), satisfies the PL-inequality (A.2) or is convex and $\arg\min\{f(\boldsymbol{\theta}) : \boldsymbol{\theta} \in \mathbb{R}^n\}$ is non-empty, then the iterates $\boldsymbol{\theta}_t$ converge to a minimizer of $f$ for both metrics $g_{GD}$ and $g_{m \odot w}$. Under the PL-inequality , the linear convergence rates are respectively $\Lambda$ and $\Lambda \min_i \gamma_i$, where $\Lambda > 0$ is the PL-inequality constant.*

*Proof.* We can apply Theorem A.3 in Jacobs & Burkholz (2025) (Thm. A.7 in the appendix) and Theorem 4.14 in Li et al. (2022) (Thm. A.6 in the appendix). For the convergences rates it is sufficient to bound the inverse metrics from below such that we have $g_{GD} \geq 1$ and $g_{m \odot w} \geq \min_i \gamma_i$. $\qquad\square$

**Riemannian gradient flow of HAM**    In comparison, HAM speeds up learning around 0. To show this and characterize HAM's dynamics, we derive its gradient flow from Eqs. (GD;HYP) by writing out the iterates in sum notation and taking the learning rate $\eta \to 0$.

**Theorem 4.2** *The Riemannian gradient flow ($\eta \to 0$) of Eqs. (GD;HYP) is:*

$$d\boldsymbol{\theta}_t = - (1 + \alpha|\boldsymbol{\theta}_t|) \odot \nabla f(\boldsymbol{\theta}_t)dt - \beta\boldsymbol{\theta}_t dt, \qquad \boldsymbol{\theta}_0 = \boldsymbol{\theta}_{init}, \tag{4}$$

*where $|\cdot|$ is applied pointwise. Moreover, if $\beta = 0$, the inverse metric is $g_{\text{HAM}}^{-1}(\boldsymbol{\theta}) = 1 + \alpha|\boldsymbol{\theta}|$.*

*Proof.* This follows from writing out the sum update and then taking the limit to get an integral equation. The gradient flow then follows from the Leibniz rule. See Theorem B.3 in the appendix. □

**Convergence of HAM**    We analyze the inverse metric and convergence behavior of HAM when $\beta = 0$. In this case, the inverse metric is given by $g_{\text{HAM}}(\boldsymbol{\theta}) = 1 + \alpha|\boldsymbol{\theta}|$ (Table 2), indicating that HAM can converge faster than gradient descent depending on $\alpha$ and the magnitude of the weights. This stands in stark contrast with sparsification methods, where a decaying $\boldsymbol{\gamma} \ll 1$ slows down movement. We formalize this behavior under the same conditions as Theorem 4.1:

**Theorem 4.3** *Under the same setting as Theorem 4.1, the iterates of HAM in Eq. (GD,HYP) with $\beta = 0$, $\boldsymbol{\theta}_t$, converge to a minimizer of $f$. Moreover, the linear convergence rate is $\Lambda$ under the PL-inequality.*

*Proof.* Similar as in Theorem 4.1 we again can lower bound $g_{\text{HAM}}^{-1} \geq 1$ for the convergence rate.

This proves that HAM avoids the vanishing inverse metric problem from the pointwise overparameterization $\boldsymbol{m} \odot \boldsymbol{w}$, while keeping the geometric benefits as we will see next. Furthermore, we discuss the case for $\beta > 0$ in §B.1.

**Implicit bias of HAM**    We characterize the implicit bias of HAM by analyzing its associated Riemannian gradient flow. This confirms that HAM not only speeds up convergence with respect to $\boldsymbol{m} \odot \boldsymbol{w}$ and small $\boldsymbol{\gamma}$, but also influences the nature of the solution. To show this, we compute the Bregman function $R_\alpha$ (see Definition A.4) such that its Hessian yields the required metric, i.e., $g_{\text{HAM}} = \nabla^2 R_\alpha$.

**Lemma 4.4** *The function $R_\alpha$ for $\alpha \in \mathbb{R}$ is given by*

$$R_\alpha(\boldsymbol{\theta}) = \sum_i \frac{(\alpha|\theta_i| + 1)\ln(\alpha|\theta_i| + 1) - \alpha|\theta_i|}{\alpha^2} - \theta_i \frac{\text{sign}(\theta_{i,0})}{\alpha}\log(1 + \alpha|\theta_{i,0}|).$$

*If $\alpha > 0$, $R_\alpha$ is a Bregman function (Definition A.4).*

*Proof.* See Lemma B.4 in the appendix. □

Concretely, we can use Lemma 4.4 to characterize the implicit bias for under-determined linear regression. Let $\{(\boldsymbol{z}_i, y_i)\}_{i=1}^d \subset \mathbb{R}^n \times \mathbb{R}$ be a dataset of size $d$. The output of a linear model $\boldsymbol{\theta}$ on the $i$-th data is $\boldsymbol{z}_i^T\boldsymbol{\theta}$. The goal therefore is to solve the regression for the target vector $\boldsymbol{y} = (y_1, y_2, \ldots, y_d)^T$ and input vector $\boldsymbol{Z} = (\boldsymbol{z}_1, \boldsymbol{z}_2, \ldots, \boldsymbol{z}_d)$.

**Theorem 4.5** *Consider the same setting as Theorem 4.3 with $\beta = 0$. Then, if $f(\boldsymbol{\theta}) := f(\boldsymbol{Z}^T\theta - \boldsymbol{y})$, the gradient flow of $\boldsymbol{\theta}_t$ in Eq. (4) converges to the solution of the optimization problem: $\boldsymbol{\theta}_\infty = \arg\min_{\boldsymbol{Z}\boldsymbol{\theta}=\boldsymbol{y}} R_\alpha(\boldsymbol{\theta})$.*

*Proof.* Apply the mirror flow part of Theorem 4.17 (Li et al., 2022) for Bregman functions. □

Theorem 4.5 provides an intuition about the type of solutions to which HAM converges. We are particularly interested in the shape of $R_\alpha$ when $\boldsymbol{\theta}_0 = \mathbf{0}$ (to understand sign flips). Note that the gradient flow is well defined at $\boldsymbol{\theta}_0$ if $\beta = 0$.

**Theorem 4.6** *Let $\boldsymbol{\theta}_0 = \mathbf{0}$. Then, $R_\alpha \sim ||\boldsymbol{\theta}||_{L_2}^2$ if $\alpha \to 0$, and $R_\alpha \sim ||\boldsymbol{\theta}||_{L_1}$ if $\alpha \to \infty$, where $\sim$ indicates proportionality, i.e., there exists some positive functions $h_1 : \mathbb{R} \to \mathbb{R}$ and $h_2 : \mathbb{R} \to \mathbb{R}$ in the neighborhood of the limiting point such that for all $\boldsymbol{\theta} \in \mathbb{R}^n$, $\lim_{\alpha \to 0} h_1(\alpha)R_\alpha(\boldsymbol{\theta}) = ||\boldsymbol{\theta}||_{L_2}^2$ and $\lim_{\alpha \to \infty} h_2(\alpha)R_\alpha(\boldsymbol{\theta}) = ||\boldsymbol{\theta}||_{L_1}$.*

*Proof.* See proof of Theorem B.5 in the appendix. □

Theorem 4.6 is illustrated in Figure 2(b). $R_\alpha$ of HAM induces an implicit bias that interpolates between $L_2$ and $L_1$, similarly to $\boldsymbol{m} \odot \boldsymbol{w}$ (Jacobs & Burkholz, 2025).

**Remark 4.7** *In the $\alpha \to \infty$ setting, HAM induces an $L_1$ bias. However, in practice, due to discretizations, this setting would require a much smaller learning rate to ensure convergence. This makes HAM less suited to fully induce sparsity on its own than the related sparsification methods. Therefore, HAM is best used in combination with other methods to find sparse solutions, acting as a guide for sparse geometry during training.*

Remark 4.7 emphasizes that HAM needs a substantially large $\alpha$ to induce sparsity on its own. This is in line with our takeaway from §3: Our hyperbolic step primarily contributes to learning the correct magnitude of a weight and promotes sign flips. This differs from $\boldsymbol{m} \odot \boldsymbol{w}$ overparameterization, where sparsity emerges due to the inherently small inverse metric.

**Remark 4.8** *In practice, we apply additional weight decay with strength $\beta > 0$. This promotes sparsity but does not change the inverse metric for HAM. In contrast, for $\boldsymbol{m} \odot \boldsymbol{w}$ it worsens the vanishing inverse metric problem (see Appendix F), as we learn from comparing gradient flows (Eq. 2 and Eq. 4). HAM has the advantage that we can freely tune $\alpha$ for the right amount of implicit sparsity. For details on how $\beta > 0$ influences Thms 4.3 and 4.5 see §B.1.*

## 5 EXPERIMENTS

Our main goal is to highlight the versatility of our novel optimizer step, HAM, and verify our theoretical insights into its mechanisms. To this end, we compare HAM to two algorithms that explicitly utilize the parameterization $\boldsymbol{m} \odot \boldsymbol{w}$: a) PiLoT (Jacobs & Burkholz, 2025), a continuous sparsification method, and b) Sign-In (Gadhikar et al., 2025), an optimization approach designed to improve training sparse masks (especially in the context of PaI). Sign-In promotes sign flips complementary to dense training by rescaling $\boldsymbol{\gamma}$ to $\boldsymbol{1}$ in intervals, partially mitigating the vanishing inverse metric problem but inducing frequent perturbations to the optimization (see Appendix F).

HAM, due to its implicit sparsity bias (Theorem 4.7) and improve plasticity, is particularly compatible with sparse training methods, as we showcase in multiple scenarios. We choose hyperparameters $(\alpha, \beta)$ based on a grid search for dense training (see Figure 11 and Figure 12). The chosen hyperparameters are transferred to all dense and sparse training methods. For ImageNet and Vision Transformers, we use $(\alpha, \beta) = (200, 1e - 3)$ and for CIFAR100 $(\alpha, \beta) = (200, 16e - 3)$.

HAM improves generalization in a way that is complementary to Sharpness Aware Minimization (SAM) (Foret et al., 2021). We also apply HAM to other tasks such as pre-training vision transformers, LLM fine-tuning and graph and node classification in Appendices G.3, G.6 and G.7. This demonstrates the general utility of HAM as an optimization principle. In addition, Appendix G.1 verifies the improved (sparse) implicit bias proven in Theorem 4.5 for underdetermined linear regression.

**Dense training** Table 3 demonstrates that HAM improves dense training for a ResNet50 on ImageNet (Deng et al., 2009). Moreover, HAM works complementary to Sharpeness Aware Minimization (SAM) (Foret et al., 2021). Combining both algorithms (SAM-HAM) achieves the best overall performance. Table 3 further highlights that training HAM longer (using a similar compute budget as SAM, whose iterations are twice as expensive) achieves a similar improvement. Figure 10 tracks the total $L_1$ norm of the parameters during training to illustrate the complementary mechanisms of HAM and SAM. The same conclusions hold for similar experiments with the smaller vision benchmark CIFAR100 (Krizhevsky et al., 2009) (see Table 7). Furthermore, Table 8 showcases performance gains also for the transformer architecture DeiT (Touvron et al., 2021) trained on ImageNet with AdamW. A grid search for HAM's hyperparameters $\alpha$ and $\beta$ on ImageNet (Deng et al., 2009) and CIFAR100 (Krizhevsky et al., 2009) is visualized in Fig. 11 and 12. The best values for $\alpha$ are stable across different tasks, while $\beta$ needs tuning similar to weight decay.

**Sparsification** We demonstrate that HAM improves state-of-the-art pruning methods AC/DC and RiGL, as well as random pruning at initialization with the same hyperparameter configuration used

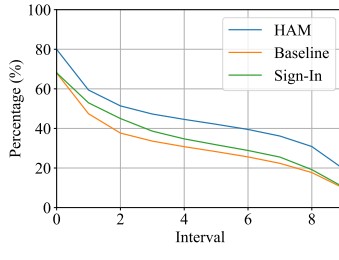 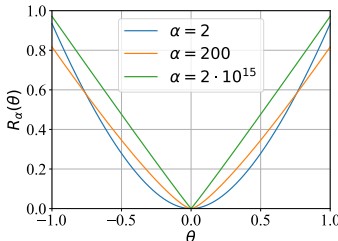

(a) Sign flips during training.      (b) HAM's Bregman function.

Figure 2: Demonstration of HAM's mechanisms. (a) The percentage of sign flips during training for Random PaI with sparsity level $90\%$ trained for $100$ epochs, where each interval correspond to ten epochs. HAM is able to consistently perform more sign flips than both the baseline and Sign-In. (b) Plot of the normalized Bregman function $R_\alpha$, where increasing $\alpha$ leads to an $L_1$ shape.

Table 3: HAM improves dense training of a ResNet50 on ImageNet.

|  | 100 epochs | 200 epochs | + SAM, 100 epochs | + SAM, 200 epochs |
|---|---|---|---|---|
| Baseline | $76.72 \pm 0.19$ | $77.27 \pm 0.13$ | $77.10 \pm 0.21$ | $77.94 \pm 0.16$ |
| HAM | $\mathbf{77.51 \pm 0.11}$ | $\mathbf{77.86 \pm 0.05}$ | $\mathbf{77.92 \pm 0.15}$ | $\mathbf{78.56 \pm 0.12}$ |

in dense training. Table 4 illustrates that dense-to-sparse training becomes significantly better with HAM. Improvements are most significant for AC/DC, which uses dense training phases effectively. We attribute this also to the fact that AC/DC turns on parameters indiscriminately, while RiGL does so based on gradient information. The improvements over PILoT and Sign-In show that we successfully extract the main beneficial mechanism of the hyperbolic geometry while mitigating the downsides.

**Sign flip mechanism** We show that HAM outperforms Sign-In (Gadhikar et al., 2025), which promotes sign flips complementary to dense training and tries to mitigate the vanishing inverse metric problem by repeated parameter rescaling. HAM still induces more sign flips than Sign-In and standard training, as demonstrated by Figure 2(a), which is in line with our theory (see Appendix E). Supporting this, we show in Appendix G.4 the improvement for training with various fixed masks.

Table 4: Dense-to-sparse training and pruning at initialization with HAM on ImageNet with ResNet50.

| Pruning type | Method | $s = 0.8$ | $s = 0.9$ | $s = 0.95$ |
|---|---|---|---|---|
| PaI | Random | $73.87(\pm 0.06)$ | $71.56(\pm 0.03)$ | $68.72(\pm 0.05)$ |
|  | Random + *Sign-In* | $74.12(\pm 0.09)$ | $72.19(\pm 0.18)$ | $69.38(\pm 0.1)$ |
|  | Random + HAM | $\mathbf{74.84(\pm 0.09)}$ | $\mathbf{72.72(\pm 0.03)}$ | $\mathbf{70.05(\pm 0.06)}$ |
| DtS | AC/DC | $75.83(\pm 0.02)$ | $74.75(\pm 0.02)$ | $72.59(\pm 0.11)$ |
|  | AC/DC + *Sign-In* | $75.9(\pm 0.14)$ | $74.74(\pm 0.12)$ | $72.88(\pm 0.13)$ |
|  | AC/DC + HAM | $\mathbf{77.2(\pm 0.14)}$ | $\mathbf{76.66(\pm 0.12)}$ | $\mathbf{75.45(\pm 0.13)}$ |
| DST | RiGL | $75.02(\pm 0.1)$ | $73.7(\pm 0.2)$ | $71.89(\pm 0.07)$ |
|  | RiGL + *Sign-In* | $75.02(\pm 0.1)$ | $74.27(\pm 0.08)$ | $\mathbf{73.07(\pm 0.17)}$ |
|  | RiGL + HAM | $\mathbf{76.22(\pm 0.07)}$ | $\mathbf{74.83(\pm 0.08)}$ | $72.93(\pm 0.1)$ |
| Cont. spars. | spred | $72.64$ | $71.84$ | $69.47$ |
|  | PILoT | $75.62$ | $74.73$ | $71.3$ |
|  | STR | $75.49(\pm 0.14)$ | $72.4(\pm 0.11)$ | $64.94(\pm 0.07)$ |
|  | STR + HAM | $\mathbf{76.37(\pm 0.18)}$ | $\mathbf{75.01(\pm 0.02)}$ | $\mathbf{71.41(\pm 0.1)}$ |

## 6 CONCLUSION

We propose a new hyperbolic update step that can be combined with any first-order optimizer and that improves generalization of dense and sparse training, making it suitable as a general purpose optimizer. Our algorithm HAM (Hyperbolic Aware Minimization) mitigates the vanishing inverse metric of the pointwise overparameterization $m \odot w$ used in recent sparsification methods, while inducing a similar implicit bias. Due to discretization, it is more suitable to control the strength and shape of the bias—and accordingly improve generalization in general, especially for dense-to-sparse training. The main mechanisms how HAM achieves this are an implicit bias towards sparsity and an acceleration of learning that promotes parameter sign flips. It remains an interesting open question if different mirror maps could create better task and optimizer-specific awareness. For example, for some tasks one might want to take into account robustness; for the optimizer, it might be the momentum or normalization. This could lead to more algorithmic advances to improve generalization via implicit bias control and to new theory for understanding the success of deep learning algorithms. In particular, optimizers with implicit biases that emulate the positive effects of other types of overparameterization, without explicitly requiring huge models, may represent an important leap in reducing the high computational costs associated with deep learning.

## ACKNOWLEDGEMENTS

The authors gratefully acknowledge the Gauss Centre for Supercomputing e.V. for funding this project by providing computing time on the GCS Supercomputer JUWELS at Jülich Supercomputing Centre (JSC). We also gratefully acknowledge funding from the European Research Council (ERC) under the Horizon Europe Framework Programme (HORIZON) for proposal number 101116395 SPARSE-ML.

## REPRODUCIBILITY STATEMENT

For the theory, detailed proofs have been provided for the main statements in Appendix B and used previously known statements have been provided in Appendix A. For the experiments, the details are provided in Appendix G with each experiment having its own subsection with accompanied specifics. The code use for the experiments is also attached.

## LLM STATEMENT

To improve fluency of the text sentence level editing has been done using large language models.

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

## A  OPTIMIZATION DEFINITIONS AND RESULTS

In this section we recall some basic definitions from convex and non-convex optimization.

**Definition A.1** *(L−smooth) A differentiable function $f : \mathbb{R}^n \to \mathbb{R}$ is said to be* L-smooth *if its gradient is Lipschitz continuous with constant $L > 0$. That is, for all $\boldsymbol{\theta}, \boldsymbol{\xi} \in \mathbb{R}^n$,*

$$\|\nabla f(\boldsymbol{\theta}) - \nabla f(\boldsymbol{\xi})\| \le L\|\boldsymbol{\theta} - \boldsymbol{\xi}\|,$$

*or equivalently,*

$$f(\boldsymbol{\xi}) \le f(\boldsymbol{\theta}) + \langle \nabla f(\boldsymbol{\theta}), \boldsymbol{\xi} - \boldsymbol{\theta} \rangle + \frac{L}{2}\|\boldsymbol{\xi} - \boldsymbol{\theta}\|^2.$$

*or equivalently,*

$$\frac{1}{2}\|\nabla f(\boldsymbol{\theta})\|^2 \le L\left(f(\boldsymbol{\theta}) - f^*\right),$$

*where $f^* = \min_{\boldsymbol{\theta} \in \mathbb{R}^n} f(\boldsymbol{\theta})$.*

**Definition A.2** *(PL-inequality) A differentiable function $f : \mathbb{R}^n \to \mathbb{R}$ satisfies the* PL inequality *with parameter $\Lambda > 0$ if for all $\boldsymbol{\theta} \in \mathbb{R}^n$,*

$$\frac{1}{2}\|\nabla f(\boldsymbol{\theta})\|^2 \ge \Lambda\left(f(\boldsymbol{\theta}) - f^*\right),$$

*where $f^* = \min_{\boldsymbol{\theta} \in \mathbb{R}^n} f(\boldsymbol{\theta})$.*

**Definition A.3** *(Legendre function Definition 3.8 ((Li et al., 2022))) Let $R : \mathbb{R}^d \to \mathbb{R} \cup \{\infty\}$ be a differentiable convex function. We say $R$ is a Legendre function when the following holds:*

- *$R$ is strictly convex on $\mathrm{int}(\mathrm{dom}\,R)$.*
- *For any sequence $\{\boldsymbol{\theta}_i\}_{i=1}^{\infty}$ going to the boundary of $\mathrm{dom}\,R$, $\lim_{i\to\infty} \|\nabla R(\boldsymbol{\theta}_i)\|_{L_2}^2 = \infty$.*

In order to recover the convergence result in Theorem 4.14 in (Li et al., 2022) the function $R$ also needs to be a Bregman function, which we define in Definition A.4. First, let us denote with $D_R$ denote the Bregman divergence with respect to the generator function $R$:

$$D_R(\boldsymbol{\theta}_1, \boldsymbol{\theta}_2) := R(\boldsymbol{\theta}_1) - R(\boldsymbol{\theta}_2) - \langle \nabla R(\boldsymbol{\theta}_2), \boldsymbol{\theta}_1 - \boldsymbol{\theta}_2 \rangle$$

for $\boldsymbol{\theta}_1, \boldsymbol{\theta}_2 \in \mathrm{dom}\,R$.

**Definition A.4** *(Bregman function Definition 4.1 (Alvarez et al., 2004)) A function $R$ is called a Bregman function if it satisfies the following properties:*

- *$\mathrm{dom}\,R$ is closed. $R$ is strictly convex and continuous on $\mathrm{dom}\,R$. $R$ is $C^1$ on $\mathrm{int}(\mathrm{dom}\,R)$).*
- *For any $\boldsymbol{\theta} \in \mathrm{dom}\,R$ and $\gamma \in \mathbb{R}$, $\{\boldsymbol{\xi} \in \mathrm{dom}\,R | D_R(\boldsymbol{\theta}, \boldsymbol{\xi}) \le \gamma\}$ is bounded.*
- *For any $\boldsymbol{\theta} \in \mathrm{dom}\,R$ and sequence $\{\boldsymbol{\theta}_i\}_{i=1}^{\infty} \subset \mathrm{int}(\mathrm{dom}\,R)$ such that $\lim_{i\to\infty} \boldsymbol{\theta}_i = \boldsymbol{\theta}$, it holds that $\lim_{i\to\infty} D_R(\boldsymbol{\theta}, \boldsymbol{\theta}_i) \to 0$.*

**Theorem A.5** *(Theorem 4.7 Alvarez et al. (2004)) If $R$ is a Legendre function with $\mathrm{dom}\,R = \mathbb{R}^n$, then if the domain of the convex conjugate $\mathrm{dom}\,R^* = \mathbb{R}^n$ implies that $R$ is a Bregman function*

From now on let $R$ be a Bregman function A.4. Consider the Riemannian gradient flow:

$$d\boldsymbol{\theta}_t = -\nabla^2 R^{-1}(\boldsymbol{\theta}_t)\nabla f(\boldsymbol{\theta}_t)dt, \qquad \boldsymbol{\theta}_0 = \boldsymbol{\theta}_{init}.$$

This covers all settings considered in the main text: gradient descent, $\boldsymbol{m} \odot \boldsymbol{w}$ and HAM as shown in Lemma B.4.

**Theorem A.6** *(Theorem 4.14 (Li et al., 2022)) Assume that $R$ is a Bregman function and that $f$ is quasi-convex, $\nabla f$ is locally Lipschitz and $\mathrm{argmin}\{f(\boldsymbol{\theta})|\boldsymbol{\theta} \in \mathbb{R}^n\}$ is non-empty. Then as $t \to \infty$, $\boldsymbol{\theta}_t$ converges to some critical point $\boldsymbol{\theta}^*$. Moreover, if $f$ is convex $\boldsymbol{\theta}_t$ converges to a minimizer of $f$.*

**Theorem A.7** *(Theorem A.3 (Jacobs & Burkholz, 2025)) Consider the same setting as Theorem A.6. Assume $R$ satisfies for all $\boldsymbol{\theta} \in \mathbb{R}^n$,*

$$\boldsymbol{z}^T \left(\nabla^2 R(\boldsymbol{\theta})\right)^{-1} \boldsymbol{z} \geq \sigma ||\boldsymbol{z}||_{L_2}^2 \qquad \forall z \in \mathbb{R}^n. \tag{5}$$

*Furthermore, assume $f$ satisfies the PL-inequality (A.2). Then $\boldsymbol{\theta}_t$ converges to a minimizer of $f$. Furthermore, the loss converges linearly with rate $\sigma\Lambda$.*

## B  PROOFS OF THEORETICAL STATEMENTS

Here we provide detailed proofs of the main statements in the paper. The theorems correspondence is:

- The proof of Theorem 3.1 is in B.1.
- The proof of Theorem 4.2 is in B.3.
- The proof of Lemma 4.4 is in B.4.
- The proof of Theorem 4.6 is in B.5.

**Theorem B.1** *(Theorem 3.1) If $\boldsymbol{m}_0 = \text{sign}(\boldsymbol{\theta}_0)\boldsymbol{w}_0 = \sqrt{|\boldsymbol{\theta}_0|}$, then*

$$\boldsymbol{\theta}_{k+1} = \boldsymbol{\theta}_k \exp\left(-\eta\left(2\,\text{sign}(\boldsymbol{\theta}_k)\nabla f(\boldsymbol{\theta}_k) + 4\beta\right)\right) \tag{6}$$

*is equivalent to Eq. (1) up-to first order Taylor approximation.*

First we use the Taylor approximation of the exponential function $\exp \boldsymbol{z} \simeq 1 + \boldsymbol{z} + \mathcal{O}(\boldsymbol{z}^2)$ to get the update:

$$\boldsymbol{\theta}_{k+1} \simeq \boldsymbol{\theta}_k - 2\eta|\boldsymbol{\theta}_k|\nabla f(\boldsymbol{\theta}_k) - 4\eta\beta\boldsymbol{\theta}_k + \mathcal{O}(\eta^2).$$

We show this is equivalent up to first order to the gradient descent of the overparameterization:

$$\boldsymbol{\theta}_{k+1} = \boldsymbol{m}_{k+1} \odot \boldsymbol{w}_{k+1} \simeq \boldsymbol{\theta}_k - \eta\left(\boldsymbol{m}_k^2 + \boldsymbol{w}_k^2\right)\nabla f(\boldsymbol{\theta}_k) - 4\eta\beta\boldsymbol{\theta}_k.$$

To do so, we show that $\boldsymbol{m}_k^2 + \boldsymbol{w}_k^2 = 2|\boldsymbol{\theta}_k|$ for all $k \in [T]$ up to zeroth-order approximation by induction. For $k = 0$, the statement holds per assumption on the initialization. The induction step is:

$$\boldsymbol{m}_{k+1}^2 + \boldsymbol{w}_{k+1}^2 = \boldsymbol{m}_k^2 + \boldsymbol{w}_k^2 - 4\boldsymbol{\theta}_k\eta\nabla f(\boldsymbol{\theta}_k) - 4\eta\beta\boldsymbol{\theta}_k + \eta^2\left(\boldsymbol{m}_k^2 + \boldsymbol{w}_k^2\right)\nabla f(\boldsymbol{\theta}_k)^2 \simeq 2|\boldsymbol{\theta}_k| + \mathcal{O}(\eta).$$

This concludes the induction and the proof. $\square$

**Corollary B.2** *Exponential gradient descent can not move through zero, preventing sign flips.*

Proof. The operation $\exp(\cdot)$ is always non-negative. Therefore multiplying with it will always keep the same sign since the sign operator is pointwise distributive:

$$\begin{aligned}
\text{sign}(\boldsymbol{\theta}_{k+1}) &= \text{sign}(\boldsymbol{\theta}_k)\exp\left(-\eta\left(2\,\text{sign}(\boldsymbol{\theta}_k)\nabla f(\boldsymbol{\theta}_k) + 4\beta\right)\right) \\
&= \text{sign}(\boldsymbol{\theta}_k)\text{sign}\left(\exp\left(-\eta\left(2\,\text{sign}(\boldsymbol{\theta}_k)\nabla f(\boldsymbol{\theta}_k) + 4\beta\right)\right)\right) \\
&= \text{sign}(\boldsymbol{\theta}_k).
\end{aligned}$$

Note we use $L-$smoothness and sufficient small learning rate to ensure bounded gradient preventing $\nabla f(\boldsymbol{\theta}_k) \to \infty$. If the gradient explodes we still can only end up in zero leading to $\text{sign}(\boldsymbol{\theta}_{k+1}) = \boldsymbol{0}$ so also no sign flip in that case. $\square$

**Theorem B.3** *(Theorem 4.2) The gradient flow ($\eta \to 0$) of Eqs. (GD;HYP) is given by:*

$$d\boldsymbol{\theta}_t = -\nabla f(\boldsymbol{\theta}_t)dt - |\boldsymbol{\theta}_t|\left(\alpha\nabla f(\boldsymbol{\theta}_t) + \text{sign}(\boldsymbol{\theta}_t)\beta\right)dt, \qquad \boldsymbol{\theta}_0 = \boldsymbol{\theta}_{init}.$$

Writing out the computation of iterates $\boldsymbol{\theta}_k$ give us:

$$\boldsymbol{\theta}_k = \boldsymbol{\theta}_0 \exp\left(\sum_{j=0}^{k-1} -\eta\alpha\,\text{sign}(\boldsymbol{\theta}_j)\nabla f(\boldsymbol{\theta}_j) - \eta\beta\right) - \sum_{j=0}^{k-1} \eta\nabla f(\boldsymbol{\theta}_j)\exp\left(-\sum_{l=j}^{k-1} \eta\alpha\,\text{sign}(\boldsymbol{\theta}_l)\nabla f(\boldsymbol{\theta}_l) - \eta\beta\right).$$

This allows use to take $\eta \to 0$ and get an integral equation for the dynamics:

$$\boldsymbol{\theta}_t = \boldsymbol{\theta}_0 \exp\left(-\int_0^t \alpha\,\text{sign}(\boldsymbol{\theta}_s)\nabla f(\boldsymbol{\theta}_s) + \beta ds\right) - \int_0^t \nabla f(\boldsymbol{\theta}_s)\exp\left(-\int_s^t \alpha\,\text{sign}(\boldsymbol{\theta}_c)\nabla f(\boldsymbol{\theta}_c) + \beta dc\right)ds. \tag{7}$$

Differentiating the first term under the Leibniz rule gives:

$$\frac{d}{dt}\boldsymbol{\theta}_0 \exp\left(-\int_0^t \alpha \operatorname{sign}(\boldsymbol{\theta}_s)\nabla f(\boldsymbol{\theta}_s) + \beta ds\right) =$$

$$\boldsymbol{\theta}_0 \exp\left(-\int_0^t \alpha \operatorname{sign}(\boldsymbol{\theta}_s)\nabla f(\boldsymbol{\theta}_s) - \beta ds\right)\left(-\alpha \operatorname{sign}(\boldsymbol{\theta}_t)\nabla f(\boldsymbol{\theta}_t) + \beta\right)$$

Next, differentiate the second term under the Leibniz rule:

$$\frac{d}{dt}\left(-\int_0^t \nabla f(\boldsymbol{\theta}_s) \exp\left(-\int_s^t \alpha \operatorname{sign}(\boldsymbol{\theta}_c)\nabla f(\boldsymbol{\theta}_c) + \beta dc\right)ds\right) =$$

$$-\nabla f(\boldsymbol{\theta}_t) - \int_0^t \frac{d}{dt}\nabla f(\boldsymbol{\theta}_s) \exp\left(-\int_s^t \alpha \operatorname{sign}(\boldsymbol{\theta}_c)\nabla f(\boldsymbol{\theta}_c) + \beta dc\right)ds =$$

$$-\nabla f(\boldsymbol{\theta}_t)dt - \int_0^t \nabla f(\boldsymbol{\theta}_s) \exp\left(-\int_s^t \alpha \operatorname{sign}(\boldsymbol{\theta}_c)\nabla f(\boldsymbol{\theta}_c) + \beta dc\right)ds\left(-\alpha \operatorname{sign}(\boldsymbol{\theta}_t)\nabla f(\boldsymbol{\theta}_t) - \beta\right)$$

Combining gives by noticing the form of $\boldsymbol{\theta}_t$:

$$d\boldsymbol{\theta}_t = -\nabla f(\boldsymbol{\theta}_t)dt - \boldsymbol{\theta}_t \left(\alpha \operatorname{sign}(\boldsymbol{\theta}_t)\nabla f(\boldsymbol{\theta}_t) + \beta\right)dt$$
$$= -\nabla f(\boldsymbol{\theta}_t)dt - |\boldsymbol{\theta}_t|\left(\alpha \nabla f(\boldsymbol{\theta}_t) + \operatorname{sign}(\boldsymbol{\theta}_t)\beta\right)dt$$

$\square$

**Lemma B.4** *(Lemma 4.4) The Bregman function $R_\alpha$ for $\alpha > 0$ is given by:*

$$R_\alpha(\boldsymbol{\theta}) = \sum_i \frac{(\alpha |\theta_i| + 1)\ln(\alpha |\theta_i| + 1) - \alpha |\theta_i|}{\alpha^2} - \theta_i \frac{\operatorname{sign}(\theta_{i,0})}{\alpha}\log(1 + \alpha|\theta_{i,0}|)$$

*Proof.*

We first construct the mirror map $R_\alpha$ by using the corresponding Hessian $g_{HAM}$. Next we check that $R_\alpha$ is a Bregman function. The Hessian of the mirror map $R_\alpha(\boldsymbol{\theta})$ is:

$$\nabla^2 R_\alpha(\boldsymbol{\theta}) = \frac{1}{1 + \alpha|\boldsymbol{\theta}|}$$

Moreover we need $\nabla R_\alpha(\boldsymbol{\theta}_0) = 0$. Therefore by integrating twice, $R_\alpha$ for $\alpha > 0$ is given by:

$$R_\alpha(\boldsymbol{\theta}) = \sum_i \frac{(\alpha |\theta_i| + 1)\ln(\alpha |\theta_i| + 1) - \alpha |\theta_i|}{\alpha^2} - \theta \frac{\operatorname{sign}(\theta_{i,0})}{\alpha}\log(1 + \alpha|\theta_{i,0}|)$$

It remains to be checked if $R_\alpha$ is Bregman. For this we use a relationship between Legendre and Bregman functions. We first show that $R_\alpha$ is Legendre and its convex conjugate as well. Then it follows from Theorem A.5 that $R_\alpha$ is Bregman for $\alpha > 0$.

Note that we have $dom R_\alpha = int(dom R_\alpha) = \mathbb{R}^n$. $R_\alpha$ is strictly convex as for all $\boldsymbol{\theta} \in \mathbb{R}^n$ the Hessian is positive definite. This shows the first statement. Next, since $R_\alpha$ is separable we can show the second statement for each parameter separately. Take a sequence $\{\theta_{i,j}\}_{i=1}^\infty$ for coordinate $j \in [n]$ such that $|\theta_{i,j}| \to \infty$ then by construction of $R_\alpha$ we have

$$\lim_{i \to \infty} \partial_j R_\alpha(\theta_{i,j})^2 = \infty$$

as $|\cdot|$ and $\log(\cdot)$ are increasing functions. Therefore $R_\alpha$ is Legendre.

The convex conjugate gradient $dom\nabla R_\alpha^* = (range\nabla R_\alpha)^{-1} = \mathbb{R}^n$. Therefore since $\mathbb{R}^n = dom\nabla R_\alpha^* \subset dom R_\alpha^*$ we can apply Theorem A.5. This concludes the result. $\square$

**Theorem B.5** *(Theorem 4.6) Let $\boldsymbol{\theta}_0 = \mathbf{0}$, then $R_\alpha \sim ||\boldsymbol{\theta}||_{L_2}^2$ if $\alpha \to 0$ and $R_\alpha \sim ||\boldsymbol{\theta}||_{L_1}$ if $\alpha \to \infty$, where $\sim$ indicates proportionality i.e. there exists some positive functions $h_1 : \mathbb{R} \to \mathbb{R}$ and $h_2 : \mathbb{R} \to \mathbb{R}$ in the neighborhood of the limiting point such that for all $\boldsymbol{\theta} \in \mathbb{R}^n$, $\lim_{\alpha \to 0} h_1(\alpha)R_\alpha(\boldsymbol{\theta}) = ||\boldsymbol{\theta}||_{L_2}^2$ and $\lim_{\alpha \to \infty} h_2(\alpha)R_\alpha(\boldsymbol{\theta}) = ||\boldsymbol{\theta}||_{L_1}$.*

*Proof.* The first statement follows from the Taylor approximation:

$$R_\alpha \simeq \sum_i \theta_i^2 + \frac{1}{\alpha}|\theta_i| - \frac{1}{\alpha}|\theta_i| \simeq ||\boldsymbol{\theta}||_{L_2}^2$$

which is valid if $|\theta_i| << \frac{1}{\alpha}$ so $h_1(\alpha) = 1$.

For large $\alpha > 0$ we have that $R_\alpha \simeq \sum_i \frac{|\theta_i|}{\alpha} \log(\alpha|\theta_i|)$. Therefore, we have

$$\frac{\alpha}{\log(\alpha)} R_\alpha(\boldsymbol{\theta}) \simeq \sum_i |\theta_i| = ||\boldsymbol{\theta}||_{L_1},$$

so $h_2 = \frac{\alpha}{\log(\alpha)}$, which is positive $\alpha > 1$. This concludes the proof. $\qquad\square$

The proof of Theorem B.5 follows similar steps as that of Woodworth et al. (2020).

**Sign discrepancy** In our implemented HAM algorithm (HYP*) we use $\text{sign}(\boldsymbol{\theta}_{k+\frac{1}{2}})$ instead of $\text{sign}(\boldsymbol{\theta}_k)$ from the derived step (HYP). We now argue why this does not change the implications of our theory, as their gradient flows are equivalent. We show in Thm. B.6 that, in continuous time, the iterates (GD;HYP*) follow a jump process. For this jump process, the jump vanishes in the flow setting, leading to no discrepancy between using either version. In discrete time, however, this may not be the case. We provide an explanation for the effect in discrete time in Appendix D. The main implication of the flow being the same is that at the end of training the implicit bias is the same, as the end corresponds to smaller learning rates.

**Theorem B.6** *Initialize $\boldsymbol{\theta}_{init} \neq 0$. Then the gradient flow of HAM i.e. Eqs. (GD;HYP*) ($\eta \to 0$) is given by:*

$$d\tilde{\boldsymbol{\theta}}_t = -\nabla f(\tilde{\boldsymbol{\theta}}_t)dt - |\tilde{\boldsymbol{\theta}}_t|\left(\alpha\nabla f(\tilde{\boldsymbol{\theta}}_t) + \text{sign}(\tilde{\boldsymbol{\theta}}_t)\beta\right)dt \qquad \tilde{\boldsymbol{\theta}}_0 = \boldsymbol{\theta}_{init}. \tag{8}$$

*Proof.* The statement follows from making the observation: Sign flips can only occur near zero; further away the processes are equivalent.

Let $\tilde{\boldsymbol{\theta}}_t$ denote the resulting process with $\eta \to 0$ and $\boldsymbol{\theta}_t$ the signed gradient flow. $\tilde{\boldsymbol{\theta}}_t$ does not have to be gradient flow as it can have discontinuous jumps due to the sign inconsistency. We can write the update as follows:

$$\tilde{\boldsymbol{\theta}}_k = \tilde{\boldsymbol{\theta}}_0 \exp\left(\sum_{j=0}^{k-1} -\eta\alpha \,\text{sign}(\tilde{\boldsymbol{\theta}}_j)\nabla f(\tilde{\boldsymbol{\theta}}_j) - \eta\beta - \eta\alpha\left(\text{sign}(\tilde{\boldsymbol{\theta}}_{j+\frac{1}{2}}) - \text{sign}(\tilde{\boldsymbol{\theta}}_j)\right)\nabla f(\tilde{\boldsymbol{\theta}}_j)\right) -$$

$$\sum_{j=0}^{k-1} \eta\nabla f(\boldsymbol{\theta}_j) \exp\left(-\sum_{l=j}^{k-1} \eta\alpha \,\text{sign}(\boldsymbol{\theta}_l)\nabla f(\boldsymbol{\theta}_l) - \eta\beta - \eta\alpha\left(\text{sign}(\tilde{\boldsymbol{\theta}}_{j+\frac{1}{2}}) - \text{sign}(\tilde{\boldsymbol{\theta}}_j)\right)\nabla f(\tilde{\boldsymbol{\theta}}_j)\right).$$

Then the discrepancy $\left(\text{sign}(\tilde{\boldsymbol{\theta}}_{j+\frac{1}{2}}) - \text{sign}(\tilde{\boldsymbol{\theta}}_j)\right)$ between the signs becomes a $\delta : \mathbb{R}^n \to \mathbb{R}^n$ function i.e.

$$\boldsymbol{\delta}(\boldsymbol{\theta}) = \begin{cases} 0 \text{ if } \theta_i \neq 0 \\ 2 \text{ if } \theta_i = 0^+ \\ -2 \text{ if } \theta_i = 0^- \end{cases}, \qquad \text{for } i \in [n]$$

The process for $\tilde{\boldsymbol{\theta}}_t$ with $\eta \to 0$ can be written as

$$\boldsymbol{\theta}_t = \boldsymbol{\theta}_0 \exp\left(-\int_0^t \alpha \,\text{sign}(\boldsymbol{\theta}_s)\nabla f(\boldsymbol{\theta}_s) + \beta - \alpha\boldsymbol{\delta}(\tilde{\boldsymbol{\theta}}_s)\nabla f(\tilde{\boldsymbol{\theta}}_s)ds\right)$$

$$- \int_0^t \nabla f(\boldsymbol{\theta}_s) \exp\left(-\int_s^t \alpha \,\text{sign}(\boldsymbol{\theta}_c)\nabla f(\boldsymbol{\theta}_c) + \beta - \alpha\boldsymbol{\delta}(\tilde{\boldsymbol{\theta}}_c)\nabla f(\tilde{\boldsymbol{\theta}}_c)dc\right)ds.$$

Differentiating under the Leibniz rule gives:

$$d\tilde{\boldsymbol{\theta}}_t = -\nabla f(\tilde{\boldsymbol{\theta}}_t)dt - |\tilde{\boldsymbol{\theta}}_t|\left(\alpha\nabla f(\tilde{\boldsymbol{\theta}}_t) + \text{sign}(\tilde{\boldsymbol{\theta}}_t)\beta + \alpha\boldsymbol{\delta}\nabla f(\tilde{\boldsymbol{\theta}}_t)\right)dt \qquad \tilde{\boldsymbol{\theta}}_0 = \tilde{\boldsymbol{\theta}}_{init}.$$

where $\boldsymbol{\delta} : \mathbb{R}^n \to \mathbb{R}^n$ is a delta function for every coordinate. Therefore, the jumps vanish as they are multiplied with $|\tilde{\boldsymbol{\theta}}|$ we have that

$$d\tilde{\boldsymbol{\theta}}_t = -\nabla f(\tilde{\boldsymbol{\theta}}_t)dt - |\tilde{\boldsymbol{\theta}}_t|\left(\alpha\nabla f(\tilde{\boldsymbol{\theta}}_t) + \mathrm{sign}(\tilde{\boldsymbol{\theta}}_t)\beta\right)dt \qquad \tilde{\boldsymbol{\theta}}_0 = \tilde{\boldsymbol{\theta}}_{init}.$$

which is equivalent to the gradient flow of Theorem B.3. $\square$

### B.1 THE EFFECT OF NON-ZERO $\beta$

The convergence and implicit bias results Thms 4.3 and 4.5 focus on the case $\beta = 0$. In the follow, we discuss general the case of $\beta \geq 0$. First, the flow takes the general form:

$$d\boldsymbol{\theta}t = -g_t^{-1}(\boldsymbol{\theta}_t)\nabla f(\boldsymbol{\theta}_t)dt - \beta\boldsymbol{\theta}_t dt, \qquad \boldsymbol{\theta}_0 = \boldsymbol{\theta}_{init}. \tag{9}$$

We have for $m \odot w$ that $g_{t,\boldsymbol{m}\odot\boldsymbol{w}}^{-1}(\boldsymbol{\theta}_t) = \sqrt{\boldsymbol{\theta}_t^2 + \boldsymbol{\gamma}_t^2}$ and for HAM we have $g_{HAM}^{-1}(\boldsymbol{\theta}_t) = 1 + \alpha|\boldsymbol{\theta}_t|$ in line with Table 2. For $\beta > 0$, we can define the on-manifold-regularization. This quantity determines the corresponding explicit regularization.

**Definition B.7** *For a time varying Riemannian gradient flow with off manifold weight decay Eq. (9), the on-manifold-regularization is given by*

$$M_t(\boldsymbol{\theta}) := \sum_{i\in[n]} \int^{\theta_i} g_{t,i}(z_i)z_i dz_i$$

*where $g_{t,i}$ is the $i$-th component of the seperable metric tensor.*

Using Definition B.7, we can compute $M_t$ in both cases. This gives

$$M_{t,\boldsymbol{m}\odot\boldsymbol{w}}(\boldsymbol{\theta}) = \sum_{i\in[n]} \sqrt{\theta_i^2 + \gamma_{t,i}^2} \text{ and } M_{\mathrm{HAM}}(\boldsymbol{\theta}) = \sum_{i\in[n]} \frac{\alpha|\theta_i| - \ln(|\alpha|\theta_i| + 1|)}{\alpha^2}. \tag{10}$$

Knowing this, we can adapt the convergence result. As both on-manifold-regularizations are convex, we converge to the minimizer of the objective function $f + \beta M$, assuming that there exists an $M$ such that $M_t \to M$ for $t \to \infty$. Note that, for $\boldsymbol{m} \odot \boldsymbol{w}$, we have that $M_t \to ||\cdot||_{L_1}$ for $t \to \infty$. This matches the LASSO optimization objective derived in spred (Ziyin & Wang, 2022). Additionally for HAM, we have that $\alpha M_{\mathrm{HAM}} \to ||\cdot||_{L_1}$ for $\alpha \to \infty$, indicating that we induce less sparsity, as we rescale with $\alpha$. Concretely, for a fixed $\beta$ and large $\alpha$, we approximately solve the LASSO objective with regularization coefficient $\beta/\alpha$. Note that for large $\alpha$, the explicit regularization strength decays while the implicit regularization gets closer to $L_1$.

Furthermore, to obtain the implicit bias result, we use the mirror flow formulation. We know from (Jacobs & Burkholz, 2025) that $\boldsymbol{m} \odot \boldsymbol{w}$ corresponds to a time-varying mirror flow for which we need $\beta \to 0$ to recover optimality. In contrast, for HAM we get the following mirror flow:

$$d\nabla R_\alpha(\boldsymbol{\theta}_t) = -\left(\nabla f(\boldsymbol{\theta}_t) + \beta\frac{\boldsymbol{\theta}_t}{1 + \alpha|\boldsymbol{\theta}_t|}\right)dt, \qquad \boldsymbol{\theta}_0 = \boldsymbol{\theta}_{init}.$$

This follows from the new objective function $f + \beta M_{\mathrm{HAM}}$. To fulfill the optimality condition in the implicit bias result for linear regression, we would need to show that the mirror flow is in the span of $Z^T$ to satisfy the KKT condition. This can only be guaranteed when the regularization is turned off. Therefore, similarly to $\boldsymbol{m} \odot \boldsymbol{w}$, we would need to turn-off the regularization at the end of training to obtain optimality.

## C   FISHER INFORMATION DERIVATIONS

Our algorithm can be interpreted through the lens of natural gradient descent (Amari, 1999; Martens, 2014). Each optimizer step corresponds to a natural gradient update $\boldsymbol{\theta}_{k+1} = \boldsymbol{\theta}_k - \eta g^{-1}(\boldsymbol{\theta}_k)\nabla f(\boldsymbol{\theta}_k)$, with $g$ is now the Fisher information. The key insight is that parameters follow a known parameterized distribution that is learned by the optimizer. For gradient descent, $g_{\mathrm{GD}}(\boldsymbol{\theta}) = \mathbf{1}$, which corresponds to a normally distributed random variable $\boldsymbol{\theta}$ with unit variance and learnable mean; i.e., $\boldsymbol{\theta} = \mathbb{E}[X]$. Thus we can interpret $\boldsymbol{\theta}$ as the mean of a normal distribution whose *position* is learned.

In contrast, the hyperbolic step (HYP) on its own corresponds to $g_{\mathrm{HYP}}(\boldsymbol{\theta}) := 1/|\boldsymbol{\theta}|$, which directly follows from the first order approximation of the exponential function. Similarly, we can match a random variables Fisher information to the metric $g_{\mathrm{HYP}}(\boldsymbol{\theta})$. It corresponds to a random variable $X$ parameterized as a normal distribution with unit variance: $\mathcal{N}(2\sqrt{|\boldsymbol{\theta}|}, \boldsymbol{I})$. In this view, weights are recovered via $\boldsymbol{\theta} = \frac{1}{4}\mathrm{sign}(\boldsymbol{\theta})\mathbb{E}[X]^2$, by using $\mathbb{E}[X] = 2\sqrt{|\boldsymbol{\theta}|}$. This means that we learn the *magnitude* of the expected position. Furthermore, if the sign is not correct, the hyperbolic step (HYP) will move the parameter exponentially fast towards zero, facilitating its sign flip. We provide derivations in next paragraph. To summarize, our combined update can be interpreted as follows:

*Learn the position (GD), and then the magnitude if the sign is correct; else move fast to zero (HYP).*

This mechanism is crucial to facilitate sparse training (Gadhikar & Burkholz, 2024a; Gadhikar et al., 2025), as portrayed in our experiments (§ 5), where HAM considerably boosts its performance.

**Derivations of the Fisher information**   We provide here the Fisher information $\mathcal{I} := g$ calculations for one dimensional random variables $\mathcal{N}(\theta, 1)$ and $\mathcal{N}(\sqrt{|\theta|}, 1)$. The Fisher information is defined as see for example Definition 1.1 in (Ly et al., 2017):

$$\mathcal{I}(\theta) = \mathbb{E}_X\left[\left(\frac{\partial}{\partial\theta}\log f(X;\theta)\right)^2\right] \tag{11}$$

where $f$ is the probability density function of the random variable $X$ and $\mathbb{E}$ is the expectation with respect to the random variable.

Let $X \sim \mathcal{N}(\theta, 1)$, where $\theta \in \mathbb{R}$ is the mean parameter. The likelihood function is:

$$f(X;\theta) = \frac{1}{\sqrt{2\pi}}\exp\left(-\frac{1}{2}(X-\theta)^2\right)$$

Taking the natural logarithm:

$$\ell(\theta) = \log f(X;\theta) = -\frac{1}{2}\log(2\pi) - \frac{1}{2}(X-\theta)^2$$

The score function is:

$$\frac{d\ell}{d\theta} = (X-\theta)$$

Then the Fisher Information is given by:

$$\mathcal{I}_{\mathrm{GD}}(\theta) = \mathbb{E}\left[\left(\frac{d\ell}{d\theta}\right)^2\right] = \mathbb{E}[(X-\theta)^2] = \mathrm{Var}(X) = 1$$

Let $X \sim \mathcal{N}(2\sqrt{|\theta|}, 1)$. The likelihood function is:

$$f(X;\theta) = \frac{1}{\sqrt{2\pi}}\exp\left(-\frac{1}{2}(X - 2\sqrt{|\theta|})^2\right)$$

The log-likelihood function is:

$$\ell(\theta) = -\frac{1}{2}\log(2\pi) - \frac{1}{2}(X - 2\sqrt{|\theta|})^2$$

Differentiate to get the score function:

$$\frac{d\ell}{d\theta} = (X - 2\sqrt{|\theta|}) \cdot \left(-\frac{\text{sign}(\theta)}{\sqrt{|\theta|}}\right) = -\text{sign}(\theta)\frac{X - 2\sqrt{\theta}}{\sqrt{\theta}}$$

Now square the score and take the expectation:

$$\mathcal{I}_{\text{HYP}}(\theta) = \mathbb{E}\left[\left(\frac{d\ell}{d\theta}\right)^2\right] = \mathbb{E}\left[\left(\frac{X - 2\sqrt{|\theta|}}{\sqrt{|\theta|}}\right)^2\right] = \frac{1}{|\theta|}\mathbb{E}[(X - 2\sqrt{|\theta|})^2] = \frac{1}{|\theta|}Var(X) = \frac{1}{|\theta|}$$

Note that instead of being part of the exponential family of distributions this distribution is part of the curved exponential family distributions.

# D  DIFFERENT SIGNS IN THE EXPONENTIAL UPDATE

In this section, we show that using the updated signs $\text{sign}(\boldsymbol{\theta}_{t+\frac{1}{2}})$ in the hyperbolic step (HYP) instead of the original ones $\text{sign}(\boldsymbol{\theta}_t)$ is actually beneficial for performance. The main difference occurs when a sign flip takes place due to the gradient step (that is, $\text{sign}(\boldsymbol{\theta}_{t+\frac{1}{2}}) \neq \text{sign}(\boldsymbol{\theta}_t)$), which can lead to a discrepancy in discrete time. Then, updating the sign leads to an acceleration away from zero, as the gradient $\nabla f(\boldsymbol{\theta}_t)$ does not change and still points in the same direction. This further aids in preventing parameters getting stuck at zero, apart from the benefits of the hyperbolic step on its own. We present the different cases in Figure 3.

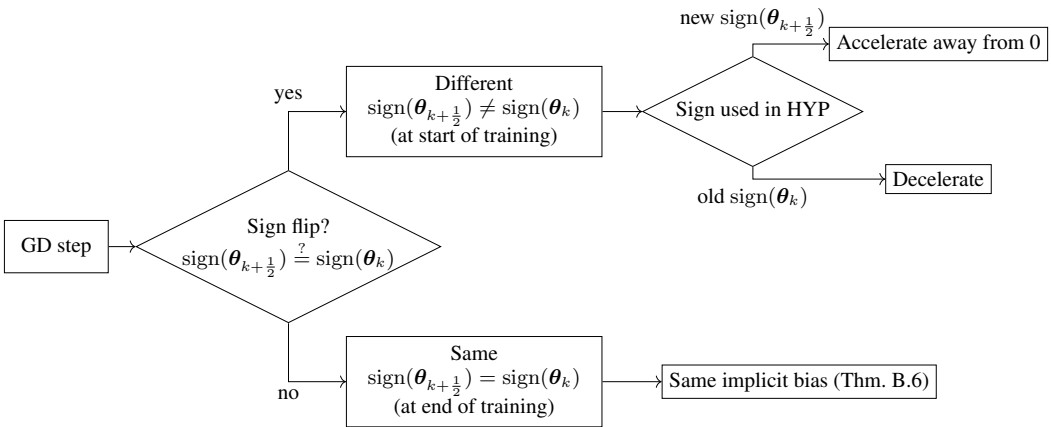

Figure 3: The difference between using $\text{sign}(\boldsymbol{\theta}_{k+\frac{1}{2}})$ and $\text{sign}(\boldsymbol{\theta}_k)$ in HYP. The main change we incur by using the new sign is that it accelerates away from zero when a sign flip occurs. Thus, when parameters are small, we can be more certain that they are actually redundant. Furthermore, when sign flips become less frequent due to decreasing learning rate at the end of training, we get the same implicit bias regardless, as shown in Theorem B.6.

# E  ONE-NEURON TOY EXAMPLE SIGN FLIPS

We show with a similar arguments as in (Gadhikar et al., 2025) that HAM allows for sign flips in a one neuron toy example. For this argument we similarly set $\beta = 0$ and we have to use a layerwise different $\alpha$. In the same vein we argue that in presence of more overparameterization using the same $\alpha$ constant is fine. This is also empirically substantiated by observing more sign flips with HAM.

**One dimensional neuron**   Consider a Gaussian i.i.d. data set $z_i \sim \mathcal{N}(0, 1)$ with $i \in [d]$. Let $f : \mathbb{R} \times \mathbb{R} \to \mathbb{R}$ be our objective function described by:

$$f(a, w) = \frac{1}{2d} \sum_{i=1}^{d} (y_i - a\sigma(wz_i))^2$$

where $\sigma(\cdot) = \max\{0, \cdot\}$ is the Rectified Linear Unit (ReLU). We want to learn a target one dimensional neuron $\tilde{a}\sigma(\tilde{w}\cdot)$, which generates the outputs $y_i$ for $i \in [d]$. Then gradient flow dynamics for HAM is described by:

$$\begin{cases} da_t = -\left(\alpha_1 |a_t| + 1\right) \partial_a f(a_t, w_t) dt, & a_0 = a_{\text{init}} \\ dw_t = -\left(\alpha_2 |w_t| + 1\right) \partial_{w_1} f(a_t, w_t) dt, & w_0 = w_{\text{init}}. \end{cases} \tag{12}$$

Note that standard gradient flow would get stuck at zero, as it has to satisfy a balance equation in Lemma E.1, which is based on Theorem 2.1 in (Gadhikar et al., 2025). The balance equation implies that if $a_0^2 - w_0^2 = C$, then for all $t \geq 0$ we have that $a_t^2 - w_t^2 = C$ (In our case: $C = 0$). In order that the student with parameters $(a, w)$ learn the ground truth, the parameters have to be able to sign flip when they don't match the ground truth's sign. This can be divided into four cases, i.e. the total amount of sign cases. In the balanced setting for gradient flow, the ground truth is recoverable only if the parameter signs align. Results of these four cases are shown in Figure 4.

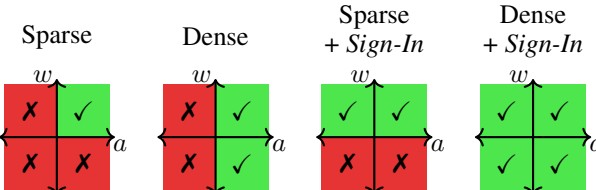

Figure 4: (Figure 2 from Gadhikar et al. (2025)), showing sign flipping benefits achieved with pointwise overparameterization $m \odot w$, for the sparse and dense case on a single-hidden neuron model.

**Lemma E.1** *Let $\tilde{g}(a, w) = \tilde{a}\sigma(\tilde{w}\cdot)$ with $\tilde{w} > 0$ be the teacher and $f$ be the student network objective such that $a$ and $w$ follow the gradient flow dynamics in Eq. (12) with a random balanced parameter initialization. For $\alpha_1 = \alpha_2 = 0$ (standard gradient flow) the student only can learn one of four cases i.e. when $w_{init} > 0$ and $\text{sign}(a_{init}) = \text{sign}(\tilde{a})$.*

Proof. If $w_{init}$ is negative then it needs to flip its sign which is prevented by the ReLU activation. We know from the balance equation that for $t \geq 0$:

$$|a_t| = |w_t|.$$

This implies that if $a_t = 0$ then $w_t = 0$ implying we can also not recover the case where $\text{sign}(a_{init}) \neq \text{sign}(\tilde{a})$. $\square$

We now show that Eq. (12) can find the ground truth, even if the sign of $a$ is misaligned, similarly as in (Gadhikar et al., 2025) for the $m \odot w$ reparameterization.

**Theorem E.2** *Let $\tilde{g}(a, w) = \tilde{a}\sigma(\tilde{w}\cdot)$ with $\tilde{w} > 0$ be the teacher and $f$ be the student network objective such that $a$ and $w$ follow the gradient flow dynamics in Eq. (12) with a random balanced parameter initialization. Moreover, let $\alpha_1 > \alpha_2 > 0$. If $w_{init} > 0$, then $f$ can learn the correct target with probability $1 - \left(\frac{1}{2}\right)^d$. In the other case ($w_{init} \leq 0$) learning fails.*

Proof. The proof idea is to show that for a balanced initialization with $w_{init} > 0$ the flow for $t > 0$ always enters the open set

$$\Gamma_0 := \{(a, w) \in \mathbb{R}^2 : a < -w, w > 0\}.$$

Furthermore, we show that the flow stays in the open set $\Gamma_0$. The system is a Riemannian gradient flow which implies that the flow converges towards a stationary point in $\Gamma_0$. It remains to be shown that the stationary point at the origin is a saddle and the stable manifold of the origin is not in $\Gamma_0$. Thus, the remaining stationary points are the global optimizers.

First we show that for balanced initializations $|a_{init}| = w_{init} > 0$ enter the region $\Gamma_0$, which can be divided into two cases. In case $a_{init} = w_{init} > 0$, we have $(a_{init}, w_{init}) \in \Gamma_0$. In case $-a_{init} = w_{init} > 0$, we have that $(a_{init}, w_{init}) \in \bar{\Gamma}_0 \setminus \Gamma_0$ i.e. the boundary of $\Gamma_0$. Therefore, we need to show the gradient field at $(a_{init}, w_{init})$ points into $\Gamma_0$.

The balanced initialization implies that

$$\partial_a f(a_{init}, w_{init}) = -\partial_w f(a_{init}, w_{init}).$$

Moreover, since $\tilde{a} > 0$, $\partial_a f(a_{init}, w_{init}) > 0$. Using that $\alpha_1 > \alpha_2 > 0$ we have that the gradient field satisfies:

$$\begin{aligned} da_{init} &= -(\alpha_1|a_{init}| + 1)\, \partial_a f(a_{init}, w_{init}) dt \\ &= (\alpha_1|w_{init}| + 1)\, \partial_w f(a_{init}, w_{init}) dt \\ &< (\alpha_2|w_{init}| + 1)\, \partial_w f(a_{init}, w_{init}) dt = -dw_{init} \end{aligned}$$

Therefore there is a $t_0 > 0$ such that $a_{t_0} < -w_{t_0} < 0$. Thus there is a $t_0 > 0$ such that $(a_{t_0}, w_{t_0}) \in \Gamma_0$.

We have entered the set $\Gamma_0$, we have to show that we cannot leave the set $\Gamma_0$. This can be shown by computing the gradient field at the boundaries. The boundary can be split up into three cases:

- $B_1 := \{(a, w) \in \mathbb{R}^2 : -a = w > 0\}$
- $B_2 := \{(a, w) \in \mathbb{R}^2 : w = 0, a > 0\}$
- The origin $\{(0, 0)\}$

The first case of $B_1$ is covered by the balanced initialization. For the second case of $B_2$ we can compute the gradient field again. We now only need that $dw_t > 0$. We linearize $dw_t$:

$$dw_t = Ca_t dt > 0,$$

where $C = \frac{1}{d}\sum_{i=1}^d \max\{0, z_i\}^2 > 0$ with probability $1 - \left(\frac{1}{2}\right)^d$. The last case is the saddle point at the origin which we show is not possible to be reached from the open set $\Gamma_0$. Thus for all $(a_{init}, w_{init}) \in \Gamma_0$ we have that for all $t \geq 0$, $(a_t, w_t) \in \Gamma_0$ or $\lim_{t\to\infty}(a_t, w_t) = (0, 0)$.

In the case that $w > 0$ the flow can be written as a dynamical system on a Riemannian manifold. This allows us to guarantee convergence to a stationary point. The flow is given by

$$\begin{cases} da_t = -C(\alpha_1|a_t| + 1)\left(a_t w_t^2 - w_t\right) dt & a_0 = a_{\text{init}} \\ dw_t = -C(\alpha|w_t| + 1)\left(a_t^2 w_t - a_t\right) dt & w_0 = w_{\text{init}}, \end{cases}$$

where $C = \frac{1}{d}\sum_{i=1}^d \max\{0, z_i\}^2 > 0$ with probability $1 - \left(\frac{1}{2}\right)^d$. This dynamical system has stationary points at the origin and the set $aw = 1$. The dynamical system is a Riemannian gradient flow system therefore the flow converges to a stationary point. The stationary point at the origin is a saddle point. Therefore, the only way of getting stuck at the origin is when we initialize on the associated stable manifold. We show that this not possible for the balanced initialization. We calculate the linearization of the stable manifold and use that the balanced initialization stays in $\Gamma_0$. The linearization at the origin $(0, 0)$ is given by

$$\begin{cases} da_t = Cw_t dt \\ dw_t = Ca_t dt. \end{cases}$$

By a direct calculation of the eigen vectors the linearization of the stable manifold is given by the vector $(-1, 1)$. This is the exact boundary of $\Gamma_0$, for which we showed that for finite $w_{init}$ and $a_{init}$

we enter $\Gamma_0$. Suppose that from $\Gamma_0$ the stable manifold is reachable. Then there is a continuous differentiable curve $\gamma_t$ with initialization $\gamma_0 = (a_{init}, w_{init}) \in \Gamma_0$ such that $\lim_{t\to\infty} \gamma_t = (0, 0)$. This is not possible as it violates the gradient field at the boundaries of $\Gamma_0$. Thus, the flow does not converge to the stationary point at the origin. This concludes the first part, since the only set of stationary points are the set of global optima.

The other two remaining cases fail as the boundary at $w = 0$ is not differentiable and the gradient flow stops there. $\square$

Theorem E.2 highlights a benefit of HAM over gradient flow. A key difference with the proof in (Gadhikar et al., 2025) is that now the stable manifold is exactly the boundary at $\Gamma_0$. Therefore, we are relying on the non-linearity of the model to push us into the open set $\Gamma_0$. A similarity between the proofs is that we rely on $\alpha_1 > \alpha_2 > 0$. In the next part we argue that for multidimensional inputs this is not necessary and we can use a single constant $\alpha$.

**Multidimensional neuron**  We can consider the gradient field at a balanced initialization for a multidimensional input case. Then we have the following inequality:

$$(\alpha|a_{init}| + 1) = (\alpha\|\boldsymbol{w}_{init}\|_{L_2} + 1) \geq \frac{1}{\sqrt{n}}(\alpha\|\boldsymbol{w}_{init}\|_{L_1} + 1) \geq \frac{1}{n}\sum_{i=1}^{n}(\alpha|w_{in,i}| + 1) \quad (13)$$

where we used the relation between the $L_1$ and $L_2$ norm. Note that now there is a significant gap due to switching between $\sqrt{n}$ and $n$. This inequality ensures that at initialization the gradient field is pointing in a similar direction as for the one dimensional case, promoting useful sign flips (Gadhikar et al., 2025).

## F ILLUSTRATION OF VANISHING INVERSE METRIC

We track the average inverse metric coefficient at $\boldsymbol{\theta} = \mathbf{0}$. This implies for HAM we get $g^{-1}(\mathbf{0}) = 1$ by definition of its inverse metric. For $\boldsymbol{m} \odot \boldsymbol{w}$ we get $g^{-1}(\mathbf{0}) = \boldsymbol{m}^2 - \boldsymbol{w}^2$. We track the average during training in the first layer of a ResNet50 trained on Imagenet. We consider $4$ scenarios: HAM, Sign-In for $90\%$ sparsity according to (Gadhikar et al., 2025), dense training $\boldsymbol{m} \odot \boldsymbol{w}$ with and without weight decay. The weight decay selected for $\boldsymbol{m} \odot \boldsymbol{w}$ is set to $2e - 5$, which is less than half of the strength it would be in case of dense training. Note that in both cases a Frobenius decay i.e. $||\boldsymbol{m} \odot \boldsymbol{w}||^2_{L_2}$ is applied in accordance with (Jacobs & Burkholz, 2025).

In Figure 5 we observe that the inverse metrics of $\boldsymbol{m} \odot \boldsymbol{w}$ decays severely when weight decay is applied. For the reparameterization, weight decay is needed to induce sparsity, so in order to use it for sparsity it needs to be used. Furthermore, note that even though Sign-In manually resets the rescaling at the start of training the metric decays at the end of training.

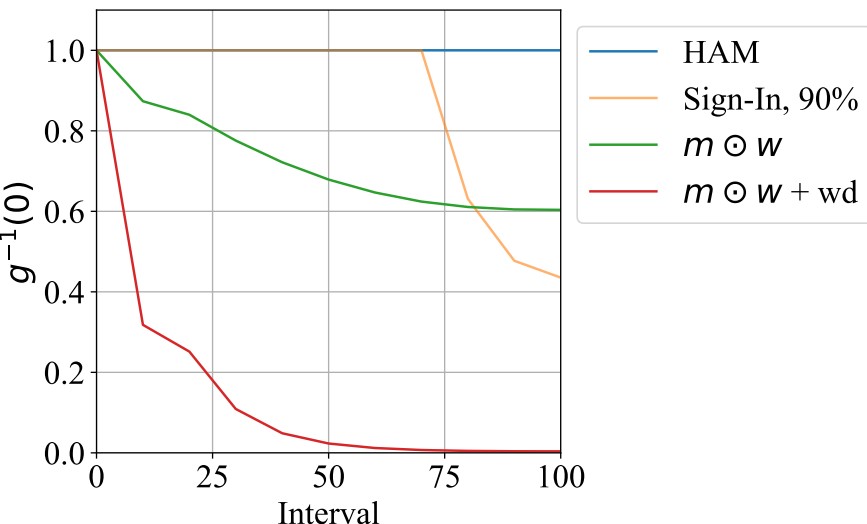

Figure 5: The first layer of a Resnet50's average inverse metric at zero reported at every tenth epoch.

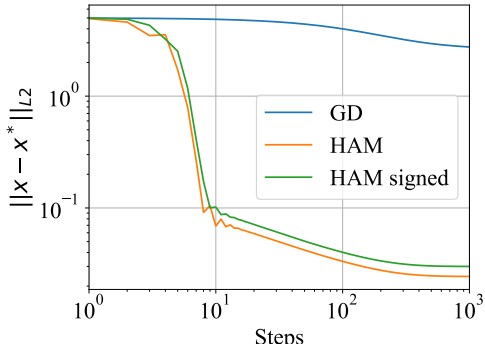

Figure 6: Gradient flow simulation of HAM, HAM with corrected sign and gradient descent. Observe the slight benefit of using $\text{sign}(\boldsymbol{\theta}_{k+\frac{1}{2}})$ instead of $\text{sign}(\boldsymbol{\theta}_k)$.

## G    EXPERIMENTAL DETAILS

We present the additional hyper parameters and other details of the experimental setup. Furthermore we provide ablations on various setups. In general HAM is applied to all layers except the batchnorm or layernorm layers.

### G.1    OVERPARAMETERIZED LINEAR REGRESSION

We illustrate Theorem 4.5 by considering a under-determined linear regression setup, similar to that of Pesme et al. (2021); Jacobs & Burkholz (2025). We consider a sparse groundtruth $\boldsymbol{\theta}^*$ and initialize at $\boldsymbol{\theta}_0 = \mathbf{0}$. Moreover, we use the mean squared error loss function. We generate data by sampling $\boldsymbol{z}_i \sim \mathcal{N}(\mathbf{0}, \boldsymbol{I})$ i.i.d. for $i \in [d]$, with $d = 40$ and $n = 100$. We compare gradient descent with and without HAM. Moreover we also show what happens if we replace $\text{sign}(\boldsymbol{\theta}_{k+\frac{1}{2}})$ with $\text{sign}(\boldsymbol{\theta}_k)$ denoted with HAM signed. The learning rate is set $\eta = 10^{-4}$, and both algorithms are run for $10e + 6$ steps. We track the distance to the ground truth during training. In Figure 6 we observe that HAM gets closer to the ground truth and converges faster then both gradient descent and HAM signed, where we set $\alpha = 1000$. This corresponds to a less strong sparse implicit bias than $L_1$.

In the same setting, we illustrate why we need both steps (GD;HYP*). We do this with an ablation, i.e., by using the hyperbolic step (HYP*) and gradient descent (GD) on their own. We initialize $\boldsymbol{\epsilon} = -10e - 5 \cdot \mathbf{1}$. This means we initialize with the opposite signs compared with the ground truth. In Figure 7, we observe that HAM reaches close to the ground truth, while the exponential step diverges, as it can not reach the ground truth due to having no sparse implicit bias. Moreover, gradient descent can also not reach the ground truth on its own. Therefore, both the hyperbolic and the gradient steps are necessary.

Furthermore, if we increase $\alpha$ and decrease the learning rate $\eta$, we can recover the ground truth solution. Concretely, consider HAM with the following configurations $(\alpha, \eta) = (10^{3+j}, 10^{-4-j})$ for $j \in [3]$. In Figure 8, we observe that we get closer to the ground truth.

### G.2    DENSE AND SPARSE TRAINING ON VISION TASKS AND ABLATION

We provide additional results on CIFAR100 (Krizhevsky et al., 2009) in Table 7. Furthermore, we train a small DeiT (Touvron et al., 2021) with AdamW in Table 8. All results are for 3 seeds. We provide the hyperparameter grid search for CIFAR 100 and Imagenet (Deng et al., 2009) in Figures 11 and 12. We find that the grid search is consistent i.e. there is a global best configuration. This implies it is easy to tune for specific tasks. We illustrate the convergence and implicit bias behaviour by tracking the training loss and $L_1$ norm in Figure 9. We also track the $L_1$ norm when comparing with SAM to show that SAM and HAM exploit different principles in Figure 10. The additional hyper-parameters of the experiments can be found in Table 5. The same parameters are used for the

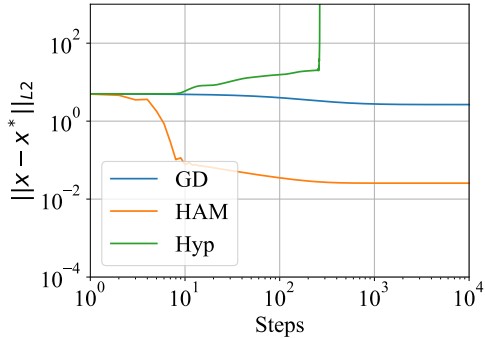

Figure 7: HAM vs hyperbolic step (HYP*) under the incorrect sign initializations.

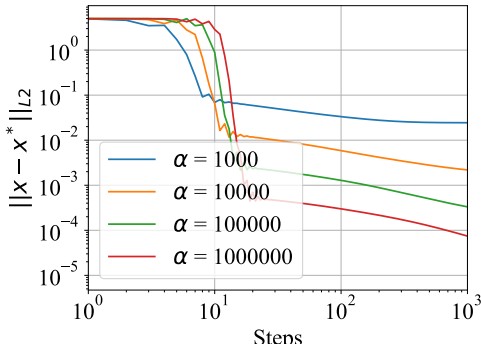

Figure 8: Gradient flow simulation of HAM with corrected sign for different $\alpha$. Larger $\alpha$ leads to closer ground truth recovery.

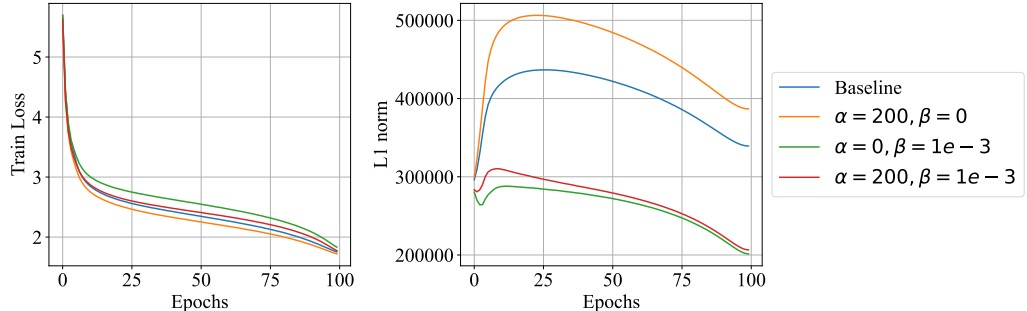

Figure 9: Training dynamics of HAM compared to the baseline for a ResNet 50 on Imagenet. Observe in the first figure (left) that the average training loss converges faster with higher $\alpha$ given the same $\beta$. This illustrates the convergence speed up predicted by our developed theory. Moreover, in the second figure (right) the average $L_1$ norm decays more due the regularization constant $\beta$, whereas larger $\alpha$ leads to a larger initial increase in the average $L_1$ norm it decays faster in the end. This is in line with being more uncertain about the sign of the weights in the beginning of training.

sparse training setup. To reproduce sparse training methods including AC/DC, RiGL and STR we use hyperparameters prescribed by the authors. Each experiment was run on 4 A100 GPUS. The code used is based on TurboPrune as in (Nelaturu et al.).

**Parameters for $m \odot w$**  The $m \odot w$ parameterization is not regularized with weight decay for the dense scenario, as this induces sparsity. Instead, weight decay is applied on the product $||m \odot w||_{L_2}^2$ with strength $5e-5$, the same strength as for dense training (Gadhikar et al., 2025).

**HAM optimization**  To optimize with HAM for dense and sparse training setups on vision tasks, we use $\alpha = 200$ and $\beta = 1e-3$ based on our grid search in Figure 12. Additionally, we clamp the exponent in the HAM step (see Equation HYP*) between $[-5, 5]$ to avoid exploding gradients. Note, in all experiments, (HYP*) is not applied to BatchNorm or LayerNorm layers.

Table 5: Training Details for the dense vision experiments presented in the paper.

| Dataset | Model | LR | Weight Decay | Epochs | Batch Size | Optim | LR Schedule |
|---------|-------|-----|--------------|--------|------------|-------|-------------|
| CIFAR100 | ResNet18 | 0.2 | $1e-4$ | 150 | 512 | SGD, $m = 0.9$, SAM | Triangular |
| ImageNet | ResNet50 | 0.25 | $5e-5$ | 100, 200 | 1024 | SGD, $m = 0.9$, SAM | Triangular |
|          | DeIT Small | 0.005 | $1e-1$ | 300 | 1024 | AdamW | Triangular |

Table 6: HAM improves dense training of a ResNet50 on Imagenet (Deng et al., 2009).

| Dataset | Baseline (no HAM) | $\alpha = 0, \beta = 1e-3$ | $\alpha = 200, \beta = 0$ | $\alpha = 200, \beta = 1e-3$ |
|---------|-------------------|----------------------------|---------------------------|------------------------------|
| HAM, 100 epchs | $76.72 \pm 0.19$ | $77.01 \pm 0.14$ | $76.72 \pm 0.07$ | $\mathbf{77.51 \pm 0.11}$ |
| HAM, 200 epchs | $77.27 \pm 0.13$ | $77.48 \pm 0.09$ | $77.24 \pm 0.09$ | $\mathbf{77.86 \pm 0.05}$ |
| SAM-HAM, 100 epchs | $77.10 \pm 0.21$ | $77.53 \pm 0.16$ | $77.21 \pm 0.09$ | $\mathbf{77.92 \pm 0.15}$ |
| SAM-HAM, 200 epchs | $77.94 \pm 0.16$ | $78.17 \pm 0.16$ | $77.60 \pm 0.03$ | $\mathbf{78.56 \pm 0.12}$ |

Table 7: Dense training with HAM on the CIFAR100 vision benchmarks.

| Dataset | Baseline | $\alpha = 0, \beta = 16e-3$ | $\alpha = 200, \beta = 0$ | $\alpha = 200, \beta = 16e-3$ |
|---------|----------|------------------------------|---------------------------|-------------------------------|
| HAM | $75.25 \pm 0.24$ | $75.36 \pm 0.04$ | $75.31 \pm 0.30$ | $\mathbf{76.12 \pm 0.27}$ |
| SAM-HAM | $75.12 \pm 0.68$ | $76.30 \pm 0.11$ | $75.25 \pm 0.20$ | $\mathbf{76.65 \pm 0.23}$ |

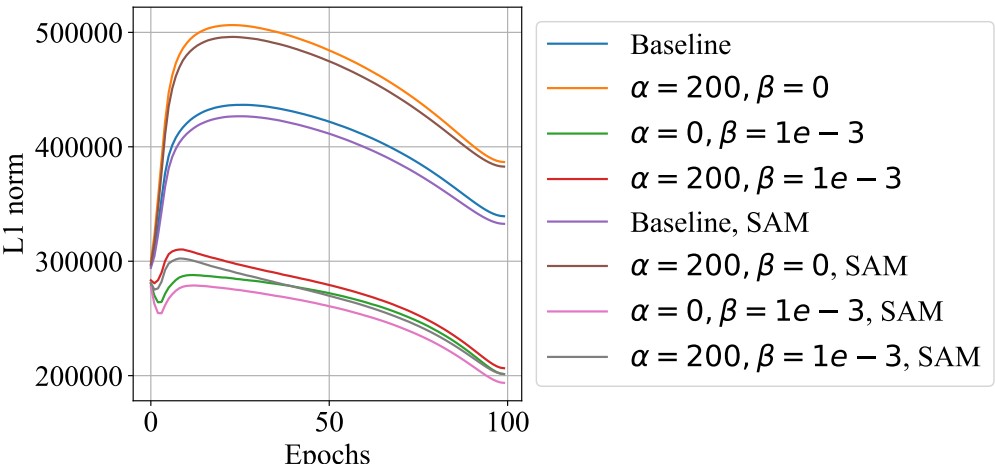

Figure 10: Training dynamics of HAM with and without SAM for a ResNet50 on Imagenet. Observe that the choice of our hyperparameters $\alpha$ and $\beta$ determine the general trend of the average $L_1$ norm while the choice between SGD and SAM make less of a difference. This provides additional evidence for their complementary working.

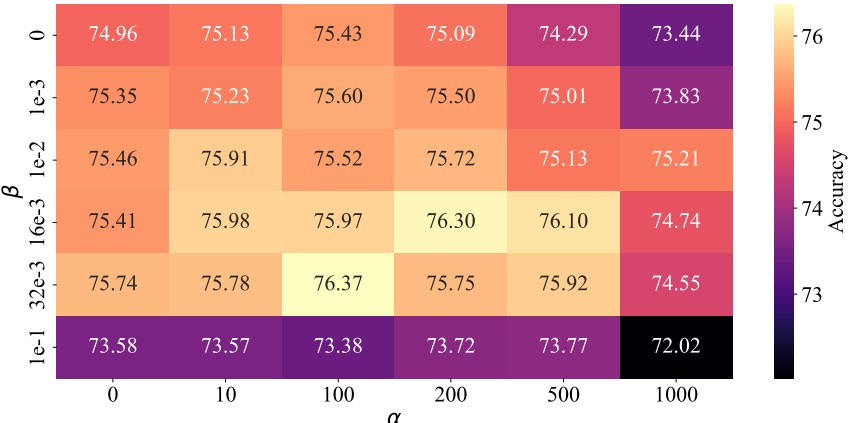

Figure 11: One seed hyperparameter search for a ResNet18 on CIFAR100.

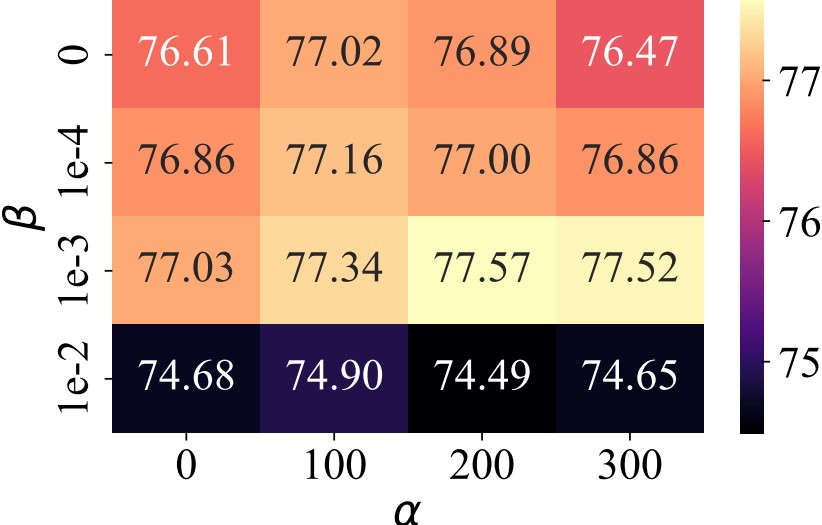

Figure 12: One seed hyperparameter search for a Resnet 50 on Imagenet.

### G.3 HAM FOR VISION TRANSFORMERS

We also verify that HAM can boost pre-training performance for ViTs trained with AdamW, for both dense training as well as sparse training with AC/DC as shown in Table 8 and Table 9 respectively. We use the same hyperparameters as for the ResNet-50 trained on Imagenet: $(\alpha, \beta) = (200, 1e-3)$.

Table 8: Pre-training a vision transformer from scratch for 300 epochs on ImageNet.

| Setup | AdamW + HAM | AdamW |
|---|---|---|
| ImageNet$_{+ \text{DeiT Small}}$ | $\mathbf{72.62}_{\pm\mathbf{0.22}}$ | $72.31_{\pm0.09}$ |

Table 9: Sparse pre-training of a vision transformer with AC/DC for 300 epochs on ImageNet at 50% sparsity.

| Setup | AC/DC + HAM | AC/DC |
|---|---|---|
| ImageNet$_{+ \text{DeiT Small}}$ | $\mathbf{73.24}_{\pm\mathbf{0.45}}$ | $72.5_{\pm0.16}$ |

### G.4 TRAINING WITH HAM AND DIFFERENT SPARSE MASKS.

HAM can be used to optimize sparse networks with different mask topologies. We train the nonzero weights of sparse mask topologies identified by different sparse methods including AC/DC, RiGL, STR and PaI masks. The weights are randomly initialized and optimized with HAM.

Note, this is equivalent to pruning at initialization with the mask obtained from the listed methods. We see a consistent improvement across all topologies with HAM except the SNIP mask, which was unstable to train also without HAM. HAM performs best for sparse masks identified by RiGL and random pruning, potentially due to better trainability and good layerwise sparsity ratios identified by these methods, which influences performance when the mask does not change during training. Results are provided in Table 10.

Table 10: HAM with different masks for a ResNet50 trained on ImageNet with $90\%$ sparsity and random initialization. ($*$ denotes a single run, as the runs for other seeds crashed.)

| Mask | Init | | |
|---|---|---|---|
| | Base | *Sign-In* | HAM |
| AC/DC | $70.66_{\pm 0.12}$ | $70.96_{\pm 0.09}$ | $\mathbf{71.84_{\pm 0.17}}$ |
| RiGL | $72.02_{\pm 0.23}$ | $72.48_{\pm 0.19}$ | $\mathbf{73.31_{\pm 0.01}}$ |
| STR | $68.36_{\pm 0.17}$ | $67.81_{\pm 0.34}$ | $\mathbf{68.75_{\pm 0.16}}$ |
| Snip | $52.9^{*}$ | $54.27^{*}$ | $44.48_{\pm 0.57}$ |
| Synflow | $60.66_{\pm 0.2}$ | $60.59_{\pm 0.07}$ | $\mathbf{62.4_{\pm 0.03}}$ |
| Random | $71.56_{\pm 0.03}$ | $72.19_{\pm 0.18}$ | $\mathbf{72.72_{\pm 0.03}}$ |

## G.5 Pruning dense models

Here we magnitude prune the best dense models acquired from both HAM and standard training. We observe in Table 11 that HAM's implicit sparsity bias makes the model perform relatively better then the standard dense model.

Table 11: Validation accuracy comparison between Pruned HAM and Dense models at different sparsity levels.

| Sparsity | HAM | Dense |
|---|---|---|
| 0.9 | 1.384 | 0.104 |
| 0.8 | 59.008 | 0.424 |
| 0.7 | 71.940 | 37.776 |
| 0.5 | 76.824 | 73.404 |

## G.6 Finetuning LLMs

As we show, HAM can also boost the performance of LLM finetuning. We evaluate on the common-sense reasoning benchmark (Hu et al., 2023) to finetune LlaMA 3.2 models (Grattafiori et al., 2024) and report accuracies across eight benchmarks in Table 12. On average, HAM improves on this task, demonstrating its compatibility with the optimizer ADAM and the LoRA architecture.

Table 12: Performance of LoRA + HAM on the commonsense reasoning (Hu et al., 2023) benchmark.

| LlaMa 3.2 | Size | HS | WG | PQ | AE | AC | OB | SQ | BQ | Avg |
|---|---|---|---|---|---|---|---|---|---|---|
| LoRA | 1B | 63.8 | 65.8 | 74.04 | 67.63 | **55.88** | 63.6 | 70.98 | **64.25** | 65.74 |
| LoRA + HAM | 1B | **64.8** | **68.35** | **74.21** | **68.39** | 51.79 | **65.8** | **71.2** | 62.17 | **65.83** |
| LoRA | 3B | 88.8 | **80.66** | **83.73** | **82.65** | 66.89 | 76.8 | 78.19 | 69.44 | 78.39 |
| LoRA + HAM | 3B | **89.40** | 80.58 | 82.69 | 81.77 | **68.43** | **80.2** | **78.25** | **69.48** | **78.85** |

**Experimental details** Each experiment was run on 4 A100 GPUs. We use th experimental setup of DoRA Liu et al. (2024) to finetune LlaMA 3.2 models of size 1B and 3B with $\alpha = 200, \beta = 1e-3$ and $\alpha = 100, \beta = 1e-4$ respectively. We find that larger models benefit from less strong regularization which is consistent with our different regularization strengths $\beta$ for ResNet18 and ResNet50.

## G.7 GRAPH AND NODE CLASSIFICATION

We report the experimental details of the 4 graph classification benchmarks from Table 13. We also include results on 13 node classification benchmarks, and an ablation on $\alpha < 0$. Each experiment was run on an A100. We consider 4 graph classification tasks on the GCN architecture (Kipf & Welling, 2017) in Table 13, and 13 node classification tasks on GCN, GATv2 (Brody et al., 2022), and GraphSAGE (Hamilton et al., 2017) in Tables 15, 16. We also include the hyperparameter grid search, as well as an ablation on negative $\alpha$. The success of negative values indicates that node classification prefers a different type of implicit bias. Nonetheless, we see consistent improvements across almost all datasets and architectures with $\alpha > 0$.

Table 13: Evaluation of HAM on 4 graph classification benchmarks from OGB (Hu et al., 2020).

| Dataset | ogbg-ppa | ogbg-molpcba | ogbg-molhiv | ogbg-code2 |
|---|---|---|---|---|
| Metric | Accuracy ↑ | Avg. Precision ↑ | AUROC ↑ | F1 score ↑ |
| GCN | $75.48 \pm 0.15$ | $27.57 \pm 0.04$ | $82.37 \pm 0.29$ | $13.89 \pm 2.11$ |
| GCN + HAM | $\mathbf{75.72 \pm 0.24}$ | $\mathbf{27.81 \pm 0.22}$ | $\mathbf{82.50 \pm 0.69}$ | $\mathbf{13.96 \pm 2.06}$ |

### G.7.1 GRAPH CLASSIFICATION

We report the performance of HAM on four graph classification datasets from Open Graph Benchmark (OGB) (Hu et al., 2020). The code to run these benchmarks is based on (Luo et al., 2025), using their choice of hyperparameters and ADAM as the optimizer. We use their GCN+ architecture, which is a GCN equipped with edge features, normalization, dropout, residual connections, feed-forward networks, and positional encodings. In order to implement HAM in combination with dropout, we mask the regularization term $\beta$ with (grad $\neq 0$). We only apply HAM on the weights and biases associated with the convolutional layers. The results, shown in Table 13, are averaged over 3 seeds. We report the best validation metric for the best values of $\alpha$ and $\beta$ for HAM, the selection of which is displayed in Table 14. The tuning range is $\alpha \in \{1, 10, 100, 200\}$, and $\beta \in \{0, 0.01, 0.1\}$. Table 14 also includes the size of the datasets in terms of number of graphs. Note that for three datasets, $\alpha$ is the same as for the vision tasks.

Table 14: $\alpha$ and $\beta$ best values for the graph classification tasks.

| Dataset | ogbg-ppa | ogbg-molpcba | ogbg-molhiv | ogbg-code2 |
|---|---|---|---|---|
| # graphs | 158,100 | 437,929 | 452,741 | 41,127 |
| $\alpha$ | 200 | 200 | 200 | 1 |
| $\beta$ | 0.1 | 0.1 | 0.1 | 0.01 |

### G.7.2 NODE CLASSIFICATION

We furthermore report the performance of HAM on thirteen node classification datasets: Cora, CiteSeer, and PubMed (Kipf & Welling, 2017), Wiki-CS (Mernyei & Cangea, 2022), Coauthor-CS, Coauthor-Physics, Amazon-Computers, and Amazon-Photo (Shchur et al., 2019) (homophilic); Amazon-Ratings, Squirrel, Chameleon, Minesweeper, and Roman-Empire (Platonov et al., 2023) (heterophilic). We evaluate over GCN, GATv2, and GraphSAGE. We only apply HAM on the weights and biases associated with the convolutional layers and on the attention parameters. The code to run these benchmarks is based on (Luo et al., 2024), using their choice of hyperparameters and ADAM as the optimizer. The results, shown in Tables 15 (homophilic) and 16 (heterophilic), are averaged over different runs according to the original setup. We report the best validation accuracy for the best values of $\alpha$ and $\beta$ for HAM, the selection of which is displayed in Table 17. The tuning range is $\alpha \in \{1, 10, 100, 200\}$, and $\beta \in \{0, 0.01, 0.1\}$. The best value per architecture is in bold, and ties are underlined. Tables 15 and 16 also include the size of the datasets in terms of number of nodes and edges.

Table 15: Evaluation of HAM on 8 homophilic node classification benchmarks.

| Dataset | cora | citeseer | pubmed | wikics | coauthor-cs | coauthor-physics | amazon-computer | amazon-photo |
|---|---|---|---|---|---|---|---|---|
| # nodes | 2,708 | 3,327 | 19,717 | 11,701 | 18,333 | 34,493 | 13,752 | 7,650 |
| # edges | 5,278 | 4,522 | 44,324 | 216,123 | 81,894 | 247,962 | 245,861 | 119,081 |
| GCN | 81.32 ± 0.30 | 68.60 ± 0.94 | 77.96 ± 0.46 | 80.71 ± 0.29 | 95.32 ± 0.12 | 97.17 ± 0.01 | 83.47 ± 0.70 | 94.33 ± 0.61 |
| GCN+HAM | **81.44 ± 0.43** | **68.64 ± 0.97** | **78.16 ± 0.65** | **80.84 ± 0.27** | 95.32 ± 0.04 | **97.18 ± 0.02** | **83.93 ± 0.58** | **95.33 ± 0.23** |
| GAT | 81.24 ± 0.68 | 68.68 ± 0.30 | 79.00 ± 0.95 | 82.35 ± 0.39 | 95.33 ± 0.07 | 97.15 ± 0.01 | 83.67 ± 0.23 | 94.47 ± 0.31 |
| GAT+HAM | **81.56 ± 0.67** | **68.92 ± 0.27** | **79.08 ± 0.88** | **82.47 ± 0.38** | 95.33 ± 0.07 | **97.16 ± 0.01** | **84.40 ± 0.92** | **94.60 ± 0.69** |
| SAGE | 80.44 ± 1.03 | 67.44 ± 0.26 | 79.36 ± 0.67 | 81.72 ± 0.46 | 95.50 ± 0.08 | 97.01 ± 0.10 | 83.07 ± 0.90 | 95.33 ± 0.31 |
| SAGE+HAM | **80.60 ± 0.63** | **67.48 ± 0.18** | **79.96 ± 0.62** | **81.78 ± 0.41** | **95.58 ± 0.09** | **97.03 ± 0.08** | **83.60 ± 1.11** | **95.40 ± 0.40** |

Table 16: Evaluation of HAM on 5 heterophilic node classification benchmarks.

| Dataset | amazon-ratings | squirrel | chameleon | minesweeper | roman-empire |
|---|---|---|---|---|---|
| # nodes | 24,492 | 2,223 | 890 | 10,000 | 22,662 |
| # edges | 93,050 | 46,998 | 8,854 | 39,402 | 32,927 |
| GCN | 53.23 ± 0.54 | 44.52 ± 1.12 | 46.12 ± 2.38 | 97.46 ± 0.24 | 90.96 ± 0.33 |
| GCN+HAM | **53.43 ± 0.44** | **44.55 ± 1.28** | **46.78 ± 2.22** | **97.78 ± 0.53** | **91.22 ± 0.40** |
| GAT | 55.47 ± 0.20 | 42.22 ± 1.73 | 45.84 ± 3.02 | 97.98 ± 0.21 | 90.58 ± 0.91 |
| GAT+HAM | **55.58 ± 0.47** | **43.17 ± 1.37** | **46.37 ± 3.32** | **98.37 ± 0.46** | **90.83 ± 0.89** |
| SAGE | 55.05 ± 0.50 | **40.91 ± 1.27** | 42.80 ± 2.90 | 97.02 ± 0.59 | 90.51 ± 0.33 |
| SAGE+HAM | **55.50 ± 0.55** | 40.85 ± 1.16 | **43.07 ± 2.84** | **97.77 ± 0.16** | **90.57 ± 0.44** |

Table 17: $\alpha$ and $\beta$ best values for the node classification tasks.

| | GCN | | GAT | | SAGE | |
|---|---|---|---|---|---|---|
| Dataset ↓ | $\alpha$ | $\beta$ | $\alpha$ | $\beta$ | $\alpha$ | $\beta$ |
| cora | 200 | 0.1 | 200 | 0.1 | 200 | 0.1 |
| citeseer | 1 | 0.01 | 10 | 0 | 200 | 0.01 |
| pubmed | 200 | 0.1 | 100 | 0 | 10 | 0.01 |
| wikics | 10 | 0.01 | 1 | 0.1 | 1 | 0.1 |
| coauthor-cs | 1 | 0.1 | 10 | 0.1 | 10 | 0.1 |
| coauthor-physics | 10 | 0 | 10 | 0.01 | 1 | 0 |
| amazon-computer | 10 | 0.01 | 10 | 0 | 200 | 0 |
| amazon-photo | 10 | 0.01 | 10 | 0 | 1 | 0.01 |
| amazon-ratings | 10 | 0.01 | 200 | 0.01 | 200 | 0 |
| squirrel | 10 | 0.01 | 200 | 0 | 200 | 0.1 |
| chameleon | 200 | 0 | 200 | 0 | 200 | 0.1 |
| minesweeper | 200 | 0.1 | 100 | 0.1 | 1 | 0.01 |
| roman-empire | 10 | 0.1 | 10 | 0.1 | 10 | 0.1 |

**Node classification with $\alpha < 0$** We perform an ablation on the node classification tasks by assigning negative values to $\alpha$, denoted as nHAM. Tables 18 (homophilic) and 19 (heterophilic) show the results of this ablation, while Table 20 displays the optimal pair of ($\alpha < 0$, $\beta$) hyperparameters. Note that the baselines never perform better than HAM or nHAM. Surprisingly, heterophilic datasets appear to be able to benefit more consistently from nHAM, especially for GCN and GraphSAGE. In homophilic datasets and GATs, it still provides consistent but smaller improvements, or matches the best performance of $\alpha > 0$. This intriguing phenomenon requires further investigation, as it may indicate the need for a different kind of implicit bias in certain graph-based architectures. Other methods particularly effective in heterophilic settings, such as modifying the adjacency matrix (Qian et al., 2024; Jamadandi et al., 2024; Rubio-Madrigal et al., 2025), may offer insight into these results.

Table 18: HAM ($\alpha > 0$) compared to nHAM ($\alpha < 0$) on 8 homophilic node classification benchmarks.

| Dataset | cora | citeseer | pubmed | wikics | coauthor-cs | coauthor-physics | amazon-computer | amazon-photo |
|---|---|---|---|---|---|---|---|---|
| GCN | 81.32 ± 0.30 | 68.60 ± 0.94 | 77.96 ± 0.46 | 80.71 ± 0.29 | 95.32 ± 0.12 | 97.17 ± 0.01 | 83.47 ± 0.70 | 94.33 ± 0.61 |
| GCN+HAM | 81.44 ± 0.43 | 68.64 ± 0.97 | **78.16 ± 0.65** | 80.84 ± 0.27 | 95.32 ± 0.04 | **97.18 ± 0.02** | 83.93 ± 0.58 | **95.33 ± 0.23** |
| GCN+nHAM | 81.44 ± 0.26 | **68.68 ± 0.99** | 78.08 ± 0.58 | **80.89 ± 0.30** | **95.34 ± 0.09** | 97.17 ± 0.00 | **84.20 ± 0.40** | 95.27 ± 0.50 |
| GAT | 81.24 ± 0.68 | 68.68 ± 0.30 | 79.00 ± 0.95 | 82.35 ± 0.39 | 95.33 ± 0.07 | 97.15 ± 0.01 | 83.67 ± 0.23 | 94.47 ± 0.31 |
| GAT+HAM | 81.56 ± 0.67 | **68.92 ± 0.27** | 79.08 ± 0.88 | **82.47 ± 0.38** | 95.33 ± 0.07 | 97.16 ± 0.01 | **84.40 ± 0.92** | 94.60 ± 0.69 |
| GAT+nHAM | 81.56 ± 0.43 | 68.92 ± 0.18 | 79.08 ± 0.88 | 82.45 ± 0.41 | 95.33 ± 0.07 | 97.16 ± 0.01 | 83.93 ± 0.31 | **94.80 ± 0.72** |
| SAGE | 80.44 ± 1.03 | 67.44 ± 0.26 | 79.36 ± 0.67 | 81.72 ± 0.46 | 95.50 ± 0.08 | 97.01 ± 0.10 | 83.07 ± 0.90 | 95.33 ± 0.31 |
| SAGE+HAM | 80.60 ± 0.63 | 67.48 ± 0.18 | **79.96 ± 0.62** | 81.78 ± 0.41 | **95.58 ± 0.09** | **97.03 ± 0.08** | 83.60 ± 1.11 | 95.40 ± 0.40 |
| SAGE+nHAM | **81.00 ± 0.35** | 67.48 ± 0.23 | 79.84 ± 0.93 | **81.82 ± 0.45** | 95.56 ± 0.12 | 97.02 ± 0.10 | 83.60 ± 1.04 | **95.67 ± 0.61** |

Table 19: HAM ($\alpha > 0$) compared to nHAM ($\alpha < 0$) on 5 heterophilic node classification benchmarks.

| Dataset | amazon-ratings | squirrel | chameleon | minesweeper | roman-empire |
|---|---|---|---|---|---|
| GCN | 53.23 ± 0.54 | 44.52 ± 1.12 | 46.12 ± 2.38 | 97.46 ± 0.24 | 90.96 ± 0.33 |
| GCN+HAM | 53.43 ± 0.44 | 44.55 ± 1.28 | 46.78 ± 2.22 | 97.78 ± 0.53 | 91.22 ± 0.40 |
| GCN+nHAM | 53.43 ± 0.25 | **44.58 ± 1.09** | 46.78 ± 1.85 | **97.85 ± 0.10** | **91.23 ± 0.36** |
| GAT | 55.47 ± 0.20 | 42.22 ± 1.73 | 45.84 ± 3.02 | 97.98 ± 0.21 | 90.58 ± 0.91 |
| GAT+HAM | 55.58 ± 0.47 | **43.17 ± 1.37** | **46.37 ± 3.32** | 98.37 ± 0.46 | **90.83 ± 0.89** |
| GAT+nHAM | **55.76 ± 0.55** | 42.84 ± 1.25 | 46.23 ± 2.91 | **98.53 ± 0.25** | 90.74 ± 0.82 |
| SAGE | 55.05 ± 0.50 | 40.91 ± 1.27 | 42.80 ± 2.90 | 97.02 ± 0.59 | 90.51 ± 0.33 |
| SAGE+HAM | **55.50 ± 0.55** | 40.85 ± 1.16 | 43.07 ± 2.84 | 97.77 ± 0.16 | 90.57 ± 0.44 |
| SAGE+nHAM | 55.25 ± 1.00 | **41.11 ± 1.61** | **43.32 ± 2.92** | **97.81 ± 0.14** | **90.64 ± 0.56** |

Table 20: $\alpha$ and $\beta$ best values for the node classification tasks with $\alpha < 0$.

| | GCN | | GAT | | SAGE | |
|---|---|---|---|---|---|---|
| Dataset ↓ | $\alpha$ | $\beta$ | $\alpha$ | $\beta$ | $\alpha$ | $\beta$ |
| cora | -10 | 0.01 | -10 | 0 | -200 | 0.01 |
| citeseer | -1 | 0.01 | -10 | 0 | -100 | 0 |
| pubmed | -100 | 0.01 | -100 | 0 | -1 | 0.1 |
| wikics | -100 | 0.1 | -10 | 0.01 | -10 | 0.01 |
| coauthor-cs | -10 | 0.01 | -100 | 0.01 | -100 | 0.01 |
| coauthor-physics | -1 | 0.1 | -1 | 0.01 | -100 | 0.1 |
| amazon-computer | -200 | 0 | -1 | 0.01 | -1 | 0.01 |
| amazon-photo | -200 | 0 | -200 | 0 | -1 | 0.01 |
| amazon-ratings | -10 | 0 | -200 | 0.1 | -1 | 0 |
| squirrel | -100 | 0 | -200 | 0.01 | -200 | 0.1 |
| chameleon | -200 | 0 | -100 | 0 | -200 | 0.1 |
| minesweeper | -100 | 0.1 | -10 | 0.1 | -200 | 0.1 |
| roman-empire | -200 | 0.1 | -10 | 0.1 | -1 | 0.1 |

