# OpenReview forum: "Hyperbolic Aware Minimization: Implicit Bias for Sparsity"
_ICLR.cc/2026/Conference — ICLR 2026 Poster_

### Official Review · Reviewer_qGRF · 2025-10-21

**Soundness:** 3
**Presentation:** 2
**Contribution:** 3
**Rating:** 4
**Confidence:** 4

**Summary:**

This paper introduces Hyperbolic Aware Minimization (HAM), an alternating optimization step that combines the beneficial sparsity-promoting implicit bias of Hadamard $m \odot w$ overparameterizations in a lightweight two-step minimization routine that aims to avoid the optimization and computational difficulties of direct $m \odot w$ parameterization. Classical $m \odot w$ parameterizations, in which each parameter is represented as a product of two factors, are known to be able to improve generalization and induce sparsity regularization, but suffer from a small inverse metric near zero that slows down small weights and impedes often important sign changes.
HAM aims to overcome this issue by alternating a standard optimizer update like SGD/Adam with a hyperbolic mirror step whose inverse metric is lower bounded by 1 at the origin and thus promotes flexible learning dynamics even for parameters close to zero, while maintaining a similar implicit bias to the $m \odot w$ parameterization. This mechanism facilitates sign plasticity, allowing parameters to adapt their sign in response to changing gradients, and introduces an implicit bias that interpolates between $L_2$ and $L_1$ regularization, promoting ‘mild sparsity’. Theoretical analyses based on Riemannian gradient flow and connections to exponential gradient descent as an approximation show that HAM circumvents the small-metric problem and that it can be shown to converge using standard methods and assumptions in the literature. Empirical results on ImageNet and CIFAR benchmarks demonstrate consistent generalization improvements in both dense and sparse regimes, whereas further experiments on a range of tasks showcase the broad use of HAM from linear regression to LLMs and graph neural networks. Combined with other sparse learning approaches, HAM is demonstrated to enhance popular methods like  AC/DC, RiGL, and STR, and works complementarily with Sharpness Aware Minimization. Overall, the work positions HAM as a geometry-utilizing optimization method that reproduces the positive effects of overparameterized training, like its implicit bias, but ensures stable sign dynamics, while remaining computationally efficient and broadly applicable.

**Strengths:**

The submission’s main strength lies in its conceptual originality and theoretical depth. It proposes a, to the best of my knowledge, novel and interesting idea of replacing the pointwise product parameterization with a direct modification of the training routine that reproduces a similar optimization geometry and implicit bias on the original weights. This reframing of the $m \odot w$ parameterization as an alternating optimization scheme is both interesting and practically relevant, as it retains the sparsity-promoting benefits of overparameterized training without increasing model size, computational cost, or running into small inverse metric issues. The work also provides a clear and well-structured overview of existing sparsification methods, placing HAM within a broad literature that includes pruning at initialization, dense-to-sparse, and dynamic sparsity training. By focusing on the gradient dynamics and implicit bias of reparameterized models, the paper contributes to an important and active area of research that links optimization geometry to generalization (and sparsity), advancing the theoretical understanding of how optimizer design influences learning behavior and generalization. The analysis is mathematically rigorous, building on (Riemannian) gradient flow and mirror descent concepts, and derives the implicit regularization induced by HAM.
The empirical evaluation is performed on a large variety of tasks ranging from the simplest linear regression models to common vision benchmarks, LLM finetuning, and graph learning tasks. Overall, the paper nicely balances theoretical results with experimental evaluations, offering an innovative and well-substantiated contribution to the study of how geometry induces implicit bias for optimization in deep learning.

**Weaknesses:**

Overall, I think the submission proposes an interesting and novel method that makes a neat paper contribution. However, some major weaknesses prohibit recommending acceptance in the submission’s current state. However, I think the majority of my concerns could be addressed in the rebuttal, which would change my rating if addressed adequately. In the following, the weaknesses are presented point-by-point.

- W1: The paper’s main weakness is its exposition. This pertains to several aspects.

- W1a.i  quality of exposition: In parts, the submission reads rather “unfinished”, including awkward theorem referencing, using inconsistent notation, referencing undefined equations, or unfinished sentences. Some proofs were poorly written and hard to follow due to a lack of contextualization (e.g., Lemma E.1, Theorem E.2, Theorem B.6), missing definitions, or a lack of explanations of why certain steps are performed. Actionable examples are in a separate suggestions list below.

- W1a.ii clarity of exposition: I got the feeling that some statements or sections implicitly only cover a certain setting (like $\beta=0$) without making this clear enough. It was not always obvious to me, e.g., if a statement is made about discrete vs continuous time, $\beta=0$ vs $\beta \neq 0$ (e.g., line 298), if weight decay on $m$ and $w$  or Frobenius (product) weight decay was used for the spred/$m \odot w$ experiments and analyses, or whether GD+HYP/GD+HYP* was used.


- W1a.iii precision of exposition: the submission contains many statements that come off a bit too bold or sweeping, or make imprecise claims (e.g., a potential exaggeration of the small inverse metric problem for $m \odot w$, or claiming that Frobenius weight decay is required for $m \odot w$ to induce sparsity (line 1360); whereas,  to my knowledge, sparsity using this parameterization can be achieved either through implicit regularization, or by using weight decay on $m$ and $w$ separately, bur not with the mentioned Frobenius decay of the product). Actionable details are in the suggestions list below.


- W1b: The submission seems a bit “all over the place” with the use cases of HAM. It remains unclear to me if its proposed use is mainly a general-purpose optimization step improving generalization, a method to induce and control implicit biases without direct overparameterization, a way to enhance existing pruning pipelines, or a sparse learning method on its own.


- W1c: I am not entirely convinced by one of the main motivating arguments for HAM, namely, to mitigate the small inverse metric problem caused by direct $m \odot w$ parameterization, which causes parameters to get stuck/decelerate around 0 while crossing signs. Fig. 1 displays the authors’ argument nicely. However, the inverse metric only vanishes for $\gamma=0$, which only happens for balanced initializations of $m,w$, i.e., $m_0^2-w_0^2=0$ (or also for $t \to \infty$ if $\beta>0$ by Eq. (2)). By choosing a random/imbalanced initialization for $m,w$, as used by Ziyin and Wang (2023) or Kolb et al. (2025), sign crossings are unproblematic since $\gamma_0>0$. Since both works report no issues of achieving competitive performances directly using $m \odot w$ in both linear or DNN models, it makes me wonder how relevant the sign flip/inverse metric problem for the $m \odot w$ parameterization is in practice when not using the balanced initialization that was avoided in previous works? Ziyin and Wang (2023) even report “Our initial experiments find no significant difference between making the norm balanced or not at initialization”, contradicting the small inverse metric problem, which would be intensified by using a balanced initialization. Furthermore, the submission derives the inverse metric as $1+\alpha |\theta|$ when $\beta=0$, being lower bound by 1. Similarly, the inverse metric for $m \odot w$ with $\beta=0$  is $\sqrt{\theta^2+\gamma^2}$, reducing to $|4*(m_0^2-w_0^2)|$ at the origin, also serving as a positive lower bound for imbalanced initializations. What is the advantage of HAM over imbalanced initializations then?

- W2: The selection of hyperparameters is not discussed for the main experiments; they are merely listed in the Appendix. Without further explanation/justification of their selection, this significantly weakens the empirical comparisons, since the results can only be assumed to hold for some arbitrary hyperparameter setting (which could potentially be biased to favor HAM). While extensive hyperparameter tuning for each method is infeasible for large-scale experiments, typically, some established set of hyperparameters, e.g., as found in related benchmark papers, is used to establish some form of comparability. This weakness is aggravated by the observation that the performances of their comparison methods sometimes differ strongly from the reported values in the original works (e.g., spred for ResNet50 on ImageNet at 80% sparsity is reported as >77% in Fig. 5 of Ziyin and Wang (2023), while the submission reports only 72.6% in Table 4; similarly STR for ResNet50 on ImageNet at 90% sparsity is reported as 74.31% (Table 1 of Kusupati et al, 2020), while the submission reports only 68.36% for the same setting in Table 10).


**Sources**:

Ziyin and Wang (ICML, 2023): https://proceedings.mlr.press/v202/ziyin23a/ziyin23a.pdf

Kolb et al. (ICLR, 2025): https://openreview.net/pdf?id=vNdOHr7mn5

Kusupati et al. (ICML, 2020): https://arxiv.org/pdf/2002.03231

**Questions:**

In the following, questions (Q) and comments/suggestions (S) are given point-by-point. A dedicated response to all points, especially the suggestions, is not expected.


- Q1: Related to W1b: What is the main purpose or application of HAM? Is it proposed mainly as a general-purpose optimization step, a modification to boost sparsification methods, a sparsification method in its own right, or a replacement of the $m \odot w$ parameterization by the HAM step? Clearly, the method could be all of the above, but the exposition should highlight the principal use case.

- Q2: Lines 17-18: The abstract states that HAM incurs less computational overhead than $m \odot w$, but, e.g., Ziyin and Wang (2023) state that $m \odot w$ on ResNet-18 takes less than 5% more time than without, which is almost negligible. Why or when is the claimed computational cost advantage (which is not empirically verified in the submission) of HAM over $m \odot w$ relevant?

- Q3: Why is Lemma 4.4 limited to $\beta=0$?

- Q4: Line 237: What is scaled and why by that factor? Context is missing here.

- Q5: Why do the results between Tables 4 and 10 differ significantly for some of the overlapping methods, i.e.,  AC/DC and STR, as well as their sign-in/HAM variants? This seems particularly strange since the reported performances for Random(+Sign-In/HAM) are identical, suggesting identical settings.

- Q6: Why does random pruning (baseline) outperform 4 out of 5 competitor methods in Table 10? This seems strange, particularly since Random performs worst across settings in Table 4.

- Q7: Clearly, HAM provides an advantage over $m \odot w$ with balanced initializations by enabling sign flips. But a simple remedy to the sign-flip problem could be to use a standard initialization for $m,w$. What additional purpose does HAM serve over $m \odot w$ in that case?

- Q8: Line 933: What justifies the use of the linearization $\exp(x) \approx 1+x$, which holds only around 0, specifically?

- Q9: I don’t fully understand the difference between Theorem 4.2/B.3 (GF for Eqs GD;HYP) and Theorem B.6 (GF of HAM). Does Theorem B.6 correspond to Eqs GD;HYP*?

- Q10: line 1252: Why does $w_i$ have an index $i$ but not $x$? And why does the setup define $z_i$ but the objective contains only $y_i$ and $x$?

- Q11: Lemma E.1 is incomprehensible to me. Why a teacher-student set-up? Which four cases? The writing is also hard to understand.

- Q12: Line 1435: In Fig. 5, it seems like none of the methods converge anywhere near close to the ground truth, with the lowest error still being larger than $0.01$ and already having leveled out. Why do seemingly all methods perform badly on this example?

Comments/suggestions:

- S1: The exposition would benefit tremendously by using boldface small letters for vectors to distinguish them from scalars, and boldface capital letters for matrices.
- S2: The re-statement of the theoretical results using a different label in the Appendix is very confusing (lines 923-927). Simply use the same names/labels in both the main text and the appendix for the same result
- S3: Hyperbolic aware minimization implies that the proposed optimization step somehow utilizes the hyperbolicity of the loss landscape to improve generalization, but the proposed optimization step imitates increased hyperbolicity through point-wise overparameterization and then mitigates the resulting small inverse metric problem at the origin. Maybe the choice of the method's name is not optimal. Also, I think it should have been “hyperbolicity aware” and not “hyperbolic aware”.
- S4: Lines 162-171: This paragraph misrepresents the scope of alternating or two-step optimization schemes, which have a long-standing history in optimization for ML. Important examples are proximal methods, soft thresholding, ADMM, alternating least squares/ridge, expectation maximization, etc. The paragraph should either give an appropriate overview or restrict itself to a certain scope of optimizers and modifications thereof.
- S5: Consistently use $\theta_{\text{init}}$ instead of variants like $\theta_{init}$ or $\theta_{in}$
- S6: Theorem 3.1 (line 211): The assumption on the initialization does not make sense to me, I think it should read $\sqrt{|\theta_0|}$ since theta is real-valued.
- S7: Clarify that Fig. 1 only holds for $\gamma=0$ where the vanishing inverse metric problem occurs.
- S8: lines 39-40: The methods by Kusupati et al. (2020) and Kolb et al. (2025) do not involve iterative parameter pruning as claimed
- S9: line 199: Why is the condition $|w_0| \leq m_0$ required? More explanation is needed.
- S10: line 271: Many analyses are performed for $\beta=0$, so shouldn’t the algorithm line read $\geq 0$?
- S11: line 291: Formulations like “mild sparsity” should be avoided and made more precise and rigorous.
- S12: line 298: Why does this equation not contain the previously used weight decay parameter $\beta$ anymore? Especially since the sentence in line 300 states that the inverse metric can change due to regularization.
- S13: lines 315-316: The references are mixed up, Theorems A.7 and A.6 are applied, which are Theorems A.3 and 4.14 in the stated references.
- S14: line 342: Sweeping statements like the one on convergence speed should be avoided if they are not made precise.
- S15: lines 355-356: The target and input vectors should have $d$ not $n$ entries as defined in 353. Also the input should be a matrix, since each $z_i$ already is a vector.
- S16: line 431: it is claimed that the optimal $\alpha, \beta$ are stable across the two tasks. But the best setting for CIFAR100 (Fig. 8) is $\alpha=100, \beta=3.2 \times 10^{-2}$, whose closest neighbor in the ImageNet grid (Fig. 9) actually yields the worst performance. Looking at other experiments, the optimal values of $\alpha$ seem to be unstable, ranging from as small as 1 to 200, or even -200.
- S17: line 870: $\lambda$ is undefined, I think this should be $\Lambda$.
- S18: line 954: The hyperlink in Eq. (3) actually references Eq. (HYP), not Eq. (3).
- S19: line 1018: ‘indicate’ should be ‘indicates’
- S20: lines 1035-1036: Sentence unclear to me, writing could be improved.
- S21: line 162: Shouldn’t $\delta$ be a function of $\theta_{j+1/2}$ and $\theta_{j}$, and the conditions in the parenthesis depend on both their signs?
- S22: line 1044: $\hat{\theta}$ does not appear in Equation 8.
- S23: lines 1088-1094: I can’t follow the writing. The exposition here could be improved.
- S24: line 1246: ‘vain’ should be ‘vein’.
- S25: line 1249: $\mathcal{N}$ should be used consistently for a Gaussian. Moreover, previously, the main text used $d$ to denote the number of samples, not $k$.
- S26: line 1252: I think there are incorrect indices in the equation (see Q10).
- S27: lines 1260-1262: The text references an undefined balance equation and stops in the middle.
- S28: line 1268: Text references undefined ‘balance equation’.
- S29: line 1286-1287: I think the statement might be incorrect. If $a_{in}=w_{in}>0$, then it can’t be true that $a_{in}<-w_{in}$, which is the requirement for belonging to $\Gamma_0$. Also $\bar{\Gamma}_{0}$ is undefined.
- S30: line 1357: Why is suddenly Frobenius decay applied to the product weight instead of weight decay separately for $m$ and $w$ as in lines 192-193? This would not induce sparsity, only separately applying weight decay induces sparsity.


**Sources**:

Ziyin and Wang (ICML, 2023): https://proceedings.mlr.press/v202/ziyin23a/ziyin23a.pdf

Kolb et al. (ICLR, 2025): https://openreview.net/pdf?id=vNdOHr7mn5

Kusupati et al. (ICML, 2020): https://arxiv.org/pdf/2002.03231

---

> ### Author Response · Authors · 2025-11-20
> **Part 1**
>
> We would like to express our gratitude for your time and efforts in providing valuable comments on our manuscript. Below, we elaborate on your concerns in a detailed point-by-point response. In case of any open questions, we would be happy to discuss them.
>
> Weaknesses:
>
> **W1 Exposition.**
> W1a
> We have incorporated the suggestions in the revised manuscript, which we believe have improved the exposition.
>
> In Section 4, all statements apply to gradient flow except the limiting result, i.e., mapping the discrete dynamics to the continuous. Moreover, we have added additional clarifications for the case distinction between $\beta = 0$ and $\beta > 0$. Also see S12. Finally, for the Frobenius decay, see the next point.
>
> To clarify, Frobenius decay is used in [1,2] to prevent the weights from becoming small too fast. Without it, the reparameterization does not work on a large scale. Intuitively, this can be explained by wanting to have the standard weight decay regularization on the original parameters $ \theta$, which helps with generalization, not sparsity.
>
> W1b See Q1.
>
> W1c See Q7.
>
> **W2 Hyperparameter tuning.** We use the same hyperparameters for sparse training as that we used in the dense setting as mentioned in the appendix. We have also now mentioned it in the main text. We provide an additional explanation for tuning HAM hyperparameters in S16 below.
>
> **Method comparison**
> Moreover, the direct comparison with the spred paper is unfair as they start from a pretrained 80% validation accuracy ResNet50, we instead train with spred from random initialization. For STR we report 72.4% for 90% sparsity not 68.36%. However this is still lower than the reported number in the paper. We attribute this to the sensitivity of their hyperparameters, see for example their grid search in Table 10 in [3] where very small changes in the weight decay already change the sparsity level achieved and accuracy significantly (we also faced these challenges in reproducing STR).
>
> **Questions**
>
> Q1. Main purpose of HAM: Its main purpose is being a general-purpose optimization step with the beneficial characteristics: an implicit sparsity bias and complementary sign flips to dense training as described in Table 1. We also provide more details in W1 and in the general response regarding the scope. We have clarified this point in the introduction.
>
> Q2. In comparison to the $m \odot w$ parameterization, which requires a 5% overhead, HAM does not have any overhead and can still achieve the same effects as the pointwise $m \odot w$ parameterization, which is emphasized in the abstract and the introduction. Note that with small batchsize, as common in large scale settings, this overhead becomes non-negligible again.
>
> Q3. Lemma 4.4 was restricted to $\beta = 0$, as it is mainly used for Theorem 4.5. A general beta can be included by noticing that we can put the regularization on the manifold by using $\beta \theta =  \beta\nabla^2 R^{-1}(\theta) \nabla^2 R(\theta) \theta $. Thus, the regularization on the manifold is $ \beta\nabla^2 R(\theta) \theta$. Please see also S12, where we discuss the inclusion of nonzero $\beta$.
>
> Q4. We changed the sentence to: The exponential update introduces a scaling of the weights which correspond to a metric $g(\theta) = 1/|\theta|$ for a Riemannian gradient flow as we see in S4.
>
> Q5. We would like to emphasize that the differences between Table 10 and Table 4 result from the fact that they are two different experimental settings. Table 4 reports sparse training algorithms in combination with HAM. Differently, Table 10 simply uses the sparse mask obtained by these methods to train a randomly initialized sparse network from scratch with HAM. The goal of Table 10 is to validate the robustness of HAM across different sparse mask structures. We have added further explanations in Table 10.
>
> Q6 (following from Q5). We would like to highlight that Table 10 reports training with fixed sparse mask structure identified by the different sparse training methods (differently from Table 4). The weight parameters are randomly initialized (and thus decoupled from the mask). This makes the optimization problem hard. In this context, the random pruning baseline is better, as it appears to be better trainable. Compared to the other sparse masks, the randomly pruned masks have denser first and last layers, which is known to aid training (See Figure 8 in [2]). We have updated the draft to explain these details.

---

> ### Author Response · Authors · 2025-11-20
> **Part 2**
>
> Q7. The unbalanced initialization has been utilized in [2]. However, the issue is that $ \gamma$ becomes smaller during training (and thus the balancing changes). This is caused by noise and weight decay. To address this issue, the parameters are rescaled during training, but the noise makes full control over the dynamics hard. Furthermore, the rescaling induces a perturbation to the training dynamics and leads to a decay in the inverse metric, as illustrated in Figure 4. In contrast, HAM induces more stable learning dynamics and achieves an overall higher performance than Sign-In (see Table 4).
> The method spred and deeper reparameterizations are not competitive with the general state-of-the-art sparsification methods. See for the supposed performance gap, the second part of W2 for spred about method comparisons. Also, we note that the method PILoT, explicitly exploits the unbalanced initialization to improve over spred [1].
>
> Q8. We have used a Taylor approximation of the exponential. So the equality holds upto order $\mathcal{O}(z^2)$.  We have added this information in the revised manuscript. Note when we apply this to the exponential step we get an approximation in  the second order of the learning rate $\eta$.
>
> Q9. Yes. Theorem B.6 is concerned with (GD;HYP*), which corresponds to gradient descent plus the hyperbolic step (HYP*) or HAM. We have updated Theorem B.6 to directly refer to (GD;HYP*).
>
>
>
> Q10. See S26.
>
> Q11.  The overall purpose of the theoretical analysis is to show that HAM facilitates sign flips that are complementary to dense training (and thus improve performance with the help of better optimization).
> The setup comes from [2], which studies a single neuron student teacher setup as an example for sparse training. The student with parameters $(a,w)$ can learn the ground truth reliably only if both the parameters ($a$ and $w$) can sign flip (in case of wrongly initialized signs). We thus have to consider four cases corresponding to the four initially possible parameter sign configurations ($sign(a) =\pm$ and $sign(w) =\pm$).
> We show that HAM gradient flow can learn the correct sign of $a$ in Theorem E.2 (even if the sign needs to be flipped during training). Lemma E.1 is paraphrased from [2] to illustrate the gradient flow case that cannot flip $a$.
> We have extended the exposition in the appendix of the revised draft and added a detailed explanation of the theoretical setup to make it more comprehensive. The writing has been improved, as detailed in S27.
>
> Q12. In Figure 5 (Figure 6 in revised manuscript), we used $\alpha =1000$. This corresponds to a less strong sparsity bias than an $L_1$ implicit bias. The ground truth can be reached better when we increase $\alpha$, while we also decrease the learning rate to keep training stable. In Figure 8 of the revised manuscript, we demonstrate this in an experiment. For $\alpha = 10^6$, we approximate the ground truth up to order $10^{-4}$ instead of $0.01$. This is in line with Theorem 4.6, which characterizes the implicit bias.
>
> **Suggestions**
>
> S1. We are happy to use bold face letters for vectors and matrices. We have done  so in the updated manuscript.
>
> S2. We have added in the appendix, in parentheses, next to each statement the corresponding result in the main part of the manuscript.
>
> S3. The name is derived from the hyperbolic entropy mirror map. We propose it as a controllable step to make the training aware of it. Therefore, we believe our title describes the method well.
>
> S4. We have included general references to the aforementioned methods as in [4,5,6,7]. We will focus on the birth-death and SAM optimizers in the paragraph as they are closely related.
>
> S5. In the revised manuscript we have changed all $in$ subscripts to init.
>
> S6. Thm 3.1 initialization assumption: Indeed it should be $\sqrt{|\theta_0|} we have updated it in the revised manuscript.
>
> S7. In line 84-87, we have updated the sentence regarding the Sign-In method: “Sign-In partially mitigates this issue by iteratively re-initializing $m$ and $w$ such that $m \odot w$ remains fixed. We set $\gamma: = (m^2 - w^2)^2 >> 0$.” Furthermore we now highlight in Figure 1 that this holds for $\gamma$ small. Note that the $\gamma$ does not have to be exactly zero for the inverse metric of $m \odot w$ to be small near zero.
>
> S8. We have changed the word ‘iteratively’ to ‘gradually’ in the revised manuscript to cover both the iterative pruning methods and continuous sparsification methods.
>
> S9. $|w_0| \leq m_0$ is an assumption of Theorem 2.1 in [1]. We have mentioned this in the revised manuscript. In practice, it is a choice that can be made due to the degree of freedom introduced by the reparameterization. It therefore does not pose any restrictions on the generality of the result, i.e., we can still cover all initializations for $\theta_{init} \in \mathbb{R}^n$.

---

> ### Author Response · Authors · 2025-11-20
> **Part 3**
>
> S12. The inverse metric of HAM does not depend on the regularization constant $\beta$ (even if $\beta\neq 0$). The related analysis sets $\beta =0$ only to simplify the exposition. We have added a full analysis for more general $\beta \geq 0$ in Appendix B.1 for completeness.
>
> Regarding Line 300, the inverse metric of $m \odot w$ in contrast, does depend on $\beta$ and thus the regularization. We have clarified this reference in the draft.
>
> S13. This is true, however, we have recapped these results in our appendix for completeness under Theorems A.6 and A.7. We clarify this in the revised manuscript by referring to both.
>
> S14. Convergence: We have nuanced the statement by including: “with respect to $ m \odot  w$ and small $ \gamma$”.
>
> S15.  We have updated this in the revised manuscript to make this consistent.
>
> S16. As most deep learning optimizers, HAM requires hyperparameter tuning. (The results of our grid search to tune $\alpha$ and $\beta$ for a dense ResNet50 on ImageNet are reported in Appendix G.)
> However, note that these values appear stable across different scenarios and serve therefore as strong default parameters, reducing the problem to a one time effort. Note that we used the same values across all ImageNet experiments, dense and sparse, and including Vision Transformers trained with AdamW.
>
> Generally, we have observed that performance improvements remain relatively robust for a large range of $\alpha$ values, as seen in Fig 11 and 12.
> Identifying a good $\beta$ is important to control the model capacity similar to weight decay and therefore may vary between tasks. To transfer the parameters between different neural network architectures, Reviewer vz7i proposed to use tensor programs. These indeed explain why $\alpha$ and $\beta$ are expected to be stable between architectural choices. The reason is that they are multiplied by the learning rate $\eta$, which fully captures the architectural change (and tensor program adjustments have been derived for it).
>
> Furthermore, in comparison with other (sparse) optimization methods, HAM usually requires less hyperparameter tuning. For example, the method Sign-In needs to tune the frequency of rescaling the reparameterization and the rescaling constant. Continuous sparsification with STR is highly sensitive to the weight decay strength and the threshold parameter.
>
>
> S17. Updated to $\Lambda$.
>
> S18. Fixed.
>
> S19. Fixed.
>
> S20. We have improved the clarity of the paragraph by explicitly referring to the steps and theorems in question as well as improving the wording.
>
> S21. The $\delta$ appears in the continuous process, therefore it only matters from which side we approach zero. In other words, moving through zero leads to a jump.
>
> S22. Fixed.
>
> S23. The main goal of this section is to convey a different interpretation of the proposed algorithm from a natural gradient descent perspective. This maps the metric to the Fisher information of a random variable X. To improve clarity in the mentioned paragraph, we mention directly that we match the metric of (HYP) to the Fisher information of a random variable and add an extra step, connecting it to the parameter by stating we use $\mathbb{E}[X] = 2\sqrt{|{\theta}|}$.
>
> S24. Fixed.
>
> S25. In the revised manuscript, all Gaussians are now denoted by the calligraphic N. Moreover, we have adopted the notation of using $d$ for the amount of samples in the appendix.
>
> S26. The indices should indeed be on the data. We have updated it in the revised manuscript.
>
> S27. We have updated the sentence to: Note that standard gradient flow would get stuck at zero, as it has to satisfy a balance equation in Lemma E.1, which is based on Theorem 2.1 in (Gadhikar et al., 2025). The balance equation implies that if $a_0^2 - w_0^2 = C$, then for all $t \geq 0$ we have that $a_t^2 - w_t^2 = C$. (In our case: $C = 0$.)
>
> S28. see S27.
>
> S29. These two statements refer to different scenarios. We consider  |a_0| = w_0 > 0, which is split in two cases. We made this more clear in the text by stating this explicitly.
>
> S30.  It is necessary to apply Frobenius decay, i.e. weight decay, on the original parameters $\theta$ to avoid too much sparsity, as mentioned in Appendix C.1 [1]. This leads to worse generalization for dense training. Intuitively, this mimics the standard weight decay.

---

> ### Author Response · Authors · 2025-11-20
> **Part 4**
>
> S10. We used $>$ to indicate that the algorithm changes the dynamics, as $\beta = \alpha =0$ corresponds to standard training. However, we have changed it to $\geq$ for generality.
>
> S11. In Line 42 of the introduction, we have made the formulation ‘mild sparsity’ more precise by characterizing it with respect to the performance relative to the dense baseline. It is the opposite of the high sparsity regime. In the high sparsity regime, performance usually degrades with respect to the dense baseline, while mild sparsity can even induce better performance than the dense baseline [8]. HAM facilitates this with the implicit sparsity bias. To emphasize this, we have clarified the connection in the introduction and used implicit sparsity bias in the other sections. Please also see the answer to Q2 of Reviewer PXeq, where we prune the dense models using magnitude based pruning and report the validation accuracy. We observe that the validation accuracy of the model trained with HAM is significantly better for each sparsity level.
>
> [1] Jacobs, Tom and Rebekka Burkholz. “Mask in the Mirror: Implicit Sparsification.” ICLR 2025
>
> [2] Gadhikar et al. (NeurIPS, 2025) https://arxiv.org/abs/2504.12801
>
> [3] Kusupati, Aditya et al. “Soft Threshold Weight Reparameterization for Learnable Sparsity.” International Conference on Machine Learning (2020).
>
> [4] Parikh, N., & Boyd, S. (2014). Proximal algorithms. Foundations and Trends in Optimization, 1(3), 127–239. https://doi.org/10.1561/2400000003
>
> [5] Boyd, S. P., Parikh, N., Chu, E., Peleato, B., & Eckstein, J. (2011). Distributed optimization and statistical learning via the alternating direction method of multipliers. Foundations and Trends in Machine Learning, 3, 1–122.
>
> [6] McLachlan, G. J., & Krishnan, T. (1996). The EM algorithm and extensions. Wiley.
>
> [7] Cichocki, A., Zdunek, R., Phan, A. H., & Amari, S. (2009). Nonnegative matrix and tensor factorizations: Applications to exploratory multi-way data analysis and blind source separation. Wiley.
>
> [8] Jin, Tian, et al. "Pruning’s effect on generalization through the lens of training and regularization." (NeurIPS, 2022)

---

> > ### Comment · Reviewer_qGRF · 2025-11-21
> > **Response to rebuttal**
> >
> > Dear author(s),
> >
> > Thank you for your detailed rebuttal and for meticulously addressing my points! Your response has clarified most of my questions and provided helpful context to resolve some of my misunderstandings (e.g., regarding the setting in Table 10 and how it differs from Table 4). In the following, I will reply to the points where open questions or disagreements remain. Regarding all other points, consider them adequately addressed and fully resolved.
> >
> > - W1c/Q7: I am not entirely convinced by the response. By using imbalanced initializations, one avoids the zero inverse metric problem at the origin at any finite time point $t$ (although learning might be considerably slowed down late in training). If I understand your response correctly, the argument is that $\gamma$ still approaches zero as the imbalance changes over training due to weight decay and noise. Which indeed is a theoretical disadvantage of directly using $m \odot w$. However, the referenced Sign-In paper [1] demonstrates in Figure 1 that the vast majority of sign flips occur during the first few epochs of training and remain stable thereafter. Yet, for imbalanced initializations, the inverse metric at the origin is far away from zero early in training, allowing for those important sign flips to occur despite the decaying inverse metric close to the origin. This is illustrated in Figure 5 of your submission, which also shows the same ResNet50 on ImageNet as in Fig. 1 of [1]. Clearly, the inverse metric is large enough during the first few epochs to permit sign flips. As noted by Reviewer FKu7, the importance of sign flips during sparse training for performance gain, and how much HAM improves upon $m \odot w$ in this respect in practical settings, remains not entirely clear.
> >
> > - W2: Simply using the same hyperparameters for dense and sparse training does not truly resolve my concern by itself, as the hyperparameters could potentially have been chosen to prefer results where HAM outperforms the comparison methods (e.g., tuning all hyperparameters with respect to HAM and then applying the same hyperparameters to all other methods - this would yield an unfair comparison). However, I have checked Table 5 for the non-HAM-specific hyperparameters used, and they are fairly established standard hyperparameters for these architectures and datasets. Those are then equally applied to all methods on the same architecture/dataset combination, if I understood correctly. If so, the argument for well-justified hyperparameter choices is more compelling.
> >
> > - S16: The updated draft still states that the best values of $\alpha, \beta$ are “stable across tasks” (line 428). I believe the experiments do not support this claim and that it should be qualified. In your response, you argue that the performance is stable with respect to $\alpha$ (for fixed $\beta$), which the heatmaps indeed support. You also state that the same values are then used for all models on the same dataset and infer that the hyperparameter choice is robust. But this merely shows that the choice is robust with respect to different models on the same task, not across tasks (defined as dataset + prediction objective/loss). Therefore, the “one-time effort” seems to be per task, not once across tasks. In my understanding, a faithful representation would be stating that the choice of $\alpha$ but not $\beta$ (as expected for a weight decay parameter) is robust on the same task for different architectural choices, a different and weaker statement than $\alpha, \beta$ being robust across tasks.
> >
> > - Potential typo: shouldn’t line 65 read “do not utilize” instead of “utilize” since you talk about AC/DC and RiGL in contrast to the $m \odot w$ overparameterized PILoT/Sign-In methods?
> >
> > [1] Gadhikar et al. (NeurIPS, 2025) https://arxiv.org/abs/2504.12801

---

> > > ### Author Response · Authors · 2025-11-21
> > >
> > > Thank you for your prompt response. We are happy that we could clarify and answer most of your points, which clearly improved our paper. Below, we provide a point-by-point answer to the open questions.
> > >
> > > **W1c/Q7: Inverse Metric problem:** Figure 1 in [1] only establishes that the sign flips caused by dense training or sparse training stabilize early. However, complementary to dense training, the sign flips caused by both Sign-In and HAM do not necessarily have to occur at the start of training and are potentially different from the sign flips that dense training can facilitate. This is also in line with the observations with respect to Sign-In [1], which explicitly rescales multiple times during training to facilitate this and does this for more than half of the training time. Otherwise, it would have been sufficient to train with the reparameterization and start unbalanced. Table 14 in [1] shows that this would not be competitive.
> > >
> > > **W2: Hyperparameter tuning:** Yes, as the reviewer rightly points out, we use standard hyperparameter settings in all experiments, as reported in Table 5. We also used the hyperparameters suggested by the respective authors to implement methods like AC/DC, RiGL and STR.
> > >
> > > **S16: Stability of $\alpha, \beta$ across tasks:** We have revised the statement to:
> > > “The best values for $\alpha$ are stable across different tasks, while $\beta$ needs tuning similar to weight decay.”
> > >
> > > Typo: Fixed in line 65.
> > >
> > > [1] Gadhikar et al. (NeurIPS, 2025) https://arxiv.org/abs/2504.12801

---

> > > > ### Comment · Reviewer_qGRF · 2025-11-21
> > > > **Reseponse to Comment by Authors**
> > > >
> > > > Dear author(s),
> > > >
> > > > Thank you for the further explanations. Overall, I think the paper’s exposition has improved considerably, and the explanations cleared up my questions. I think the submission in its current state constitutes a worthwhile contribution to the previous line of work (PILoT, Sign-In), investigating point-wise overparameterization and sign flips for sparse training. The technical analysis appears sound, and the exposition is now clearer.
> > > > Nevertheless, the work borders on being an incremental contribution to the previous line of work. However, the novelty of replacing direct $m \odot w$ parameterization with an optimizer step sufficiently differentiates this work in my opinion. Furthermore, some skepticism remains regarding the significance and role of sign flips in sparse training, the usefulness of HAM in mitigating the issue, as well as the claimed generalization-boosting property of HAM in general, for which the paper contains insufficient evidence.
> > > >
> > > > In light of these changes, I will raise my score to 6. I have no further questions or comments at the moment.

---

### Official Review · Reviewer_FKu7 · 2025-10-27

**Soundness:** 3
**Presentation:** 4
**Contribution:** 2
**Rating:** 6
**Confidence:** 4

**Summary:**

This paper proposes a new optimizer by adding one additional hyperbolic mirror step to the original gradient descent (GD). This modification is originally inspired by the sparsity implicit bias of point-wise overparameterization. The main contribution of this paper lies in that the newly designed approach further mitigates the vanishing inverse metric issue near zero of prior approaches. The authors analyze their method in the linear regression setup, and further demonstrate the effectiveness of their method in different benchmarks.

**Strengths:**

1. This paper is well-motivated and well-organized. Specifically,
   - The authors first state the sparsity implicit bias of point-wise overparameterization, which can be applied to achieve sparsity of training models by realizing the resulted Riemannian gradient flow. This gives the start point of this paper.
   - Then the vanishing inverse metric issue is identified, which induce the hardness of sign flip prior approaches. This gives the motivation of this paper.
   - The authors then solve this issue by adding a constant 1 to $| \theta |$ (this is in fact an approximation of $\sqrt{ \theta^2 + \gamma^2}$ when $\gamma \to 0$) in the inverse metric. This is well-motivated in my view.
   - Finally, the authors propose to use a two-step update manner to apply this newly designed inverse metric of  Riemannian flow.

2. This paper provides a theoretical foundation of their method, which can help the readers understand some basic properties of this newly proposed optimizer in a very simple setting. Then this theoretical analysis is further supported by empirical evaluations on vision benchmarks where several methods are compared and discussed.

**Weaknesses:**

1. There lacks intuitive explanation on the connection between sign flip and performance gaining. While the authors motivate that the vanishing inverse metric is an issue of prior approaches, there still lacks clear evidence that solving this issue is strongly related to the performance gaining. This makes the scope of the applicability of HAM unclear.

2. The theoretical analysis still has limitations. The theoretical analysis is based on very limited settings. This leaves several important questions unanswered, e.g., while the authors claimed a faster convergence rate compared to GD in the simple setting, the readers are still unclear whether this is true in more general setting.

3. This HAM highly depends on the value of $\nabla f$, hence how to choose a suitable $\alpha$ can be very tricky. For example, the range of $\alpha$ is not restricted so that it goes from 0 to a very large constant. This makes it very hard and non-intuitive to pick a suitable $\alpha$. This unavoidably gives parameter tuning burden in practical training.

4. Minor: The replacement of $sign(\theta_0)$ with $sign(\theta_k)$ is confusing: the study of object in Theorem 3.1 applies $\theta_0$ but this is replaced by $\theta_k$ (and further replaced by $\theta_{k + 1 / 2}$ later). I'm a bit confused about why the authors start with analyzing the version with $\theta_0$ instead of directly proposing HYP.

**Questions:**

1. The questions raised in the Weaknesses: a. What is the connection between sign flip and performance improvements? b. How do we define the range of $\alpha$?

2. The two-step update manner makes me wonder the comparison with other two-step methods, e.g., momentum methods. Can the authors discuss the comparison regarding several aspects such as convergence speed (as the authors claim that HAP has a faster convergence rate compared to GD) with some of other two-step methods?

3. Is it possible to design a one-step update optimizer inspired by HAM?

4. Prior works revealed that stochasticity can boost sparsity implicit bias of point-wise overparameterization, then how does stochasticity affect HAM?

---

> ### Author Response · Authors · 2025-11-20
>
> We would like to express our gratitude for your time and efforts in providing valuable comments on our manuscript. Below, we elaborate on your concerns in a detailed point-by-point response. In case of any open questions, we would be happy to discuss them.
>
> **Weaknesses:**
>
> **W1 and Q1a. Connection between sign flips and performance.**
> We have a comprehensive theoretical analysis in Appendix E that explains the link between sign flips and performance. It provides an intuition based on a theoretically tractable single neuron example. In summary, (S)GD struggles to flip parameter signs of the first and last layer if the model is sparse (i.e. has few inputs) and therefore cannot recover the correct solution in most cases (unless the signs are initialized correctly). However, training with overparameterization (i.e. more input dimensions) enables sign flipping in the first layer. Complementary to this, HAM enables sign flipping in the last layer. Only combining HAM and overparameterized training can solve the problem in all initialization cases and recover the ground truth. For an illustration of this see Figure 4 in the appendix.
>
> **W2. Convergence rate.**
> We agree that faster convergence in gradient flow does not necessarily translate to faster convergence with finite learning rate as limitation. Yet, we also provide empirical evidence for our theoretical claims that show that the faster convergence rate holds more generally. Figure 2a demonstrates that HAM induces more sign flips, indicating that the vanishing inverse metric problem is mitigated. Note also that HAM converges faster in practice than dense training without regularization ($\beta = 0$) (see Figure 9 in the appendix). This can be attributed to a larger inverse metric for each value $ \theta \in \mathbb{R}^n$, as visualized in Figure 1.
>
> **W3. and Q1b. Hyperparameter tuning.**
> We find alpha to be quite robust. High performance gains can be achieved across a wide range of different values. Even sometimes beta transfers to different learning settings (like DeiT). The default parameters that we obtained by tuning on ImageNet are reused across our experiments.
>
> **W4. Signs.** We acknowledge that the change in signs can be confusing. The goal was to first show how the hyperbolic step is related to the overparameterization $ m \odot  w$. In this derivation, the $\text{sign}( \theta_0)$ appears naturally. However from Corollary B.2 we know that the signs are equal. Therefore, we have replaced the $\text{sign}( \theta_0)$ in the revised manuscript with  $\text{sign}( \theta_k)$.
>
> **Questions:**
>
> Q1: Please see the answers to W1 and W3.
>
> **Q2: Momentum**. Nestorov momentum can accelerate convergence as well under certain conditions. While our method HAM provably improves convergence in the flow setting compared to Sign-In, our main goal is to transfer the benefits of a) an implicit sparsity bias and b) complementary sign flip ability. Momentum methods on their own do not induce sparsity but can potentially help with sign flips, as they act like a heavy ball allowing passing through zero. Note that in our ResNet experiments, we use SGD with momentum as a default and AdamW for transformers. Adding HAM improves both a) and b) leading to improved generalization. Please also see the answer to W2 and the implicit sparsity experiment for Reviewer PXeq in Q2.
>
> **Q3: Single step optimizer.** Yes this potentially can be accomplished using the direct discretization of the resulting Riemannian gradient flow in case of gradient descent. However this may interact differently with other optimization parts such as momentum, adam, weight decay etc. This, therefore, needs careful additional considerations.
>
> **Q4: Stochasticity.** Indeed, stochasticity induces sparsity. Similar to the regularization, it controls the inverse metric constant $ \gamma$ for the pointwise overparameterization $m \odot w$. In contrast, combined with HAM, the stochastic gradient flow would evolve on the manifold, leaving the inverse metric unchanged. Thus, it would not change the implicit sparsity bias. HAM allows us to exert more control over the amount of sparsity bias. Note, however, the noise can still have effects inside the manifold.
>
> [1] Gadhikar et al. (NeurIPS, 2025) https://arxiv.org/abs/2504.12801
>
> [2] Jin, Tian, et al. "Pruning’s effect on generalization through the lens of training and regularization." (NeurIPS, 2022)

---

### Official Review · Reviewer_q7Do · 2025-10-31

**Soundness:** 3
**Presentation:** 3
**Contribution:** 3
**Rating:** 6
**Confidence:** 1

**Summary:**

This paper analyzes the training dynamics of pointwise overparametrization, where a parameter is replaced by a product $m \odot w$.
The authors argue that this reparametrization, when studied as a Riemannian gradient flow, suffers from a small inverse metric near $\theta = 0$.
This can cause parameters to get "stuck" and prevent crucial sign flips, and existing remedies like Sign-In introduce perturbations that do not clearly improve training dynamics.
To address this, the authors propose Hyperbolic Aware Minimization (HAM), an optimization step designed to capture the benefits of pointwise overparametrization without getting stuck near zero.
The main idea in HAM is to alternate a standard optimizer step with a hyperbolic mirror step.
Experiments on vision models show that HAM improves the final generalization of state-of-the-art sparse training methods like AC/DC and RiGL.

**Strengths:**

1. The paper is well-written and clearly identifies a gap in the existing literature.

2. The theoretical contribution is clear.

3. The experiments appear to support the claims.

**Weaknesses:**

While I did not see any obvious weaknesses, I also cannot strongly endorse this paper as I am an outsider to this area.

**Questions:**

Q1. It's not entirely clear to me why HAM's hyperbolic steps on a "dense" $\theta$ should preserve the benefits of optimizing on the product $m \odot w$. Could the authors please give me an intuitive explanation as to why?

Q2. Could the authors please expand on what the advantages of a product overparametrization are compared to the others discussed?

---

> ### Author Response · Authors · 2025-11-20
>
> We would like to express our gratitude for your time and efforts in providing valuable comments on our manuscript. Below, we elaborate on your concerns in a detailed point-by-point response. In case of any open questions, we would be happy to discuss them.
>
> Questions:
>
> **Q1. Benefits of HAM for dense training.**
> As we have argued, the clear benefits of HAM and $ m \odot  w$ are a) an implicit bias towards sparsity and b) the ability to promote parameter sign flips during training. These two properties are beneficial for dense training for the following reasons:
> a) The implicit sparsity bias is a form of regularization that can improve generalization by mitigating overfitting. It is well known in the pruning literature that a mild amount of sparsity (like a sparsity of 0.4-0.5) often induces performance benefits (compared to dense training) [5].
> b) As we show in the appendix by analyzing a single neuron case, sign flips promoted by HAM are complementary to sign flips promoted by dense training. In Appendix E we consider a one neuron toy example with either multi and one dimensional input corresponding to dense and sparse training.
> HAM, similar to Sign-In, is able to flip the sign of the $a$ parameter; this is complementary to what dense training can accomplish on its own.
> We have updated the single neuron example as well with a figure that highlights these effects in the Appendix (see Fig 4).
>
> **Q2 Advantages of overparameterization $ m \odot  w$.**
> As mentioned in the answer for Q1 $ m \odot  w$ and HAM share the benefit of an implicit sparsity bias and an improved ability to sign flip. We provide more details on this, as follows:
>
> $ m \odot  w$ combined with weight decay induces sparsity in the original parameter $\theta$, as the optimization objective becomes equivalent to LASSO ($L_1$ regularization) [1]. In other words, it approximates LASSO in a continuous differentiable (smooth) way. Moreover, another benefit is that gradient flow transitions the implicit bias from $L_2$ to $L_1$. Consequently, sparsity is induced gradually so that the start of training benefits from dense training [2].
> Recently, pointwise overparameterization has been shown to allow for sign flips that are complementary to dense training. These have been shown to be especially important for sparse training to explore the parameter space better [3]. It results from changing the speed in which layers learn.
>
> **Properties of other parameterizations.**
> Other reparameterization such as deeper pointwise products [4] could lead to higher sparsity as these correspond to an $L_{2/D}$ regularization where $D$ is the depth. However, [4] shows empirically that beyond depth $D = 4$ there seems to be diminishing returns for generalization performance. Furthermore, such parameterizations would require careful initialization schemes to prevent vanishing/exploding gradients. In addition, the  mirror flow analysis becomes more complex, as the mirror flow has no analytical form for $D>2$. This is due to the solution of the flow having a finite blow up. We can see this by considering the ODE $d x_t = - x_t^{D-1} dt, \ x_0 = x_{init}$ for $D =2$. The solution $x_t = x_0 exp(-t)$ exists for all $t \geq 0$. For $D > 2$ however, we get $x_t = (x_0^{-(D-2)} + (D-2)*t)^{(-1/(D-2))}$, which blows up in finite time. Therefore,  a discrete update scheme (as we have with HAM) would be unstable and would require additional considerations.
>
> [1] Ziyin and Wang (ICML, 2023): https://proceedings.mlr.press/v202/ziyin23a/ziyin23a.pdf
>
> [2] Jacobs, Tom and Rebekka Burkholz. (ICLR, 2025) https://openreview.net/forum?id=U47ymTS3ut
>
> [3] Gadhikar et al. (NeurIPS, 2025) https://arxiv.org/abs/2504.12801
>
> [4] Kolb et al. (ICLR, 2025): https://openreview.net/pdf?id=vNdOHr7mn5
>
> [5] Jin, Tian, et al. "Pruning’s effect on generalization through the lens of training and regularization." (NeurIPS, 2022)

---

> > ### Comment · Reviewer_q7Do · 2025-11-27
> >
> > Thank you, this is very informative. I will maintain my score due to the Reviewer Confidence reasons listed above.

---

### Official Review · Reviewer_vz7i · 2025-11-01

**Soundness:** 2
**Presentation:** 2
**Contribution:** 3
**Rating:** 6
**Confidence:** 3

**Summary:**

This paper proposes a new optimizer step: Hyperbolic-Aware Minimization (HAM), where first a nominal step is taking by the base optimizer (e.g. Gradient Descent), then a "hyperbolic" step analogous to exponential gradient descent. By taking the sign of the previous iterate into account, this method reduces the updated iterate's magnitude for the components of the previous iterate and the gradient that agree, otherwise the exponential step blows up the updated iterate's magnitude to exaggerate the descent step on disagreeing components. The optimizer behavior is supported by deriving the corresponding Riemannian gradient flow, and applying it to the toy problem of underdetermined linear regression. Therein, it is found that HAM induces an in-between of $L^1$ and $L^2$-flavored regularization, with the behavior approaching $L^1$ when the hyperparameter $\alpha$ is large. It is noted that this implicit bias toward sparsity is rather weak. The efficacy of HAM is then demonstrated on a wide range of training tasks.

**Strengths:**

This paper proposes an interesting optimizer modification that attempts to reap the benefits of "point-wise" overparameterization via parameter-wise multiplication without introducing a memory overhead. The sign-adjustment using the previous iterate in the hyperbolic step seems to be novel, and has an interesting interpretation via Riemannian gradient flow. The experiments seem to demonstrate that HAM provides modest improvements across many scenarios as a drop-in adjustment, which seems to imply its general-purpose nature. Furthermore, it seems that the "sparsity-friendly" nature of HAM synergizes well with pruning or sparse training methods.

**Weaknesses:**

Overall, to me it is not made clear in Sections 3 and 4 the main motivating purpose of HAM. In particular, it is claimed via a Riemannian gradient flow analysis that HAM converges faster than gradient flow, but the main numerical results do not seem to focus on this. It is then claimed that HAM encourages a particular kind of implicit sparsity bias, but it is noted by the authors in Remark 4.7 that this is relatively weak alone. As such, it would be good to clarify the core motivation of HAM, and how the claimed benefits are supported by the theoretical derivations. Furthermore, neither theoretical intuition lends an explanation to why HAM seems to get better *generalization* on dense training tasks. Though the effect is good to see, does any of the theoretical motivation suggest this?

Secondly, though the optimizer is presented as a drop-in adjustment, it does introduce two new hyperparameters in $\alpha, \beta$ without intuitive guidance of how these should be set. Indeed, the hyperparameter sweeps in the appendix seem to suggest the choice of $\alpha$ is important, and that the optimal choice of $\alpha$ can vary to be quite large, and the optimal corresponding choice of $\beta$ also varies quite significantly on a logarithmic scale between $0$ and $1$. Without additional guidance or intuition, this means HAM introduces two additional independent hyperparameters to sweep, which is non-trivial to grid-search over.

**Questions:**

On the note of the hyperparameter scale, in line with recent lines of literature studying the effective learning rate induced by various hyperparameters, is it possible that the "correct" scale of $\alpha$ (and $\beta$) actually has an implicit dependence on the scale of the parameter block (e.g. width)? For example, the Maximum Update Parameterization framework (see e.g. [1]) suggests that layer-wise descent methods require a per-layer adjustment based on the dimensionality in order to guarantee stable evolution across model scales. It seems that an entrywise modification like HAM should also naturally admit some layerwise dimensionality scaling. In particular, did the authors experience the optimal choice of $\alpha, \beta$ changing across different model sizes?

The HAM step compares the sign of the gradient with the iterate. When using a different descent method than GD, is the hyperbolic step adjusted accordingly, or does it still compare to the gradient?

[1] Yang et al. "Tensor Programs V: Tuning Large Neural Networks via Zero-Shot Hyperparameter Transfer"

---

> ### Author Response · Authors · 2025-11-20
> **Part 1**
>
> We would like to express our gratitude for your time and efforts in providing valuable comments on our manuscript. Below, we elaborate on your concerns in a detailed point-by-point response. In case of any open questions, we would be happy to discuss them.
>
> Weaknesses:
>
> **W1. Core motivation and scope of HAM.**
> We have clarified in the introduction and discussion that HAM is a general purpose optimization step that can improve both dense and sparse training by promoting sign flips and sparsity.
>
> Sign flipping has been first identified as a severe challenge in the context of sparse training, where the problem is more pronounced. Baselines to mitigate this issue (i.e. pointwise overparameterization $ m \odot  w$ as in Sign-In) have been developed in this context. Sign-In is characterized by two advantages: a) an implicit sparsity bias and b) an ability to promote sign flips. However, Sign-In also has a few limitations, including a) increased memory demands and b) a vanishing inverse metric around 0 that hampers parameter movement. HAM inherits the two advantages while overcoming the two limitations. Accordingly, HAM is designed to boost sparse training and this is its primary purpose. We have provided ample theoretical and empirical evidence that demonstrate that it is able to do so.
>
> HAM can also improve dense training, as a) the implicit sparsity bias can mitigate overfitting (see Lines 42, 408 and also the answer to Q2 of Reviewer PXeq); and b) the promoted sign flips are complementary to the ones facilitated by dense training, as detailed in Appendix E.
>
>
> **W1. Convergence.**
> We have clarified our claim of faster convergence to faster convergence than the reparameterization $m \odot  w$, in line with our introduction and our theory regarding the vanishing inverse metric problem. We note that convergence in gradient flow (studied in theory) does not necessarily translate to faster convergence with finite learning rate.
> Yet, we observe more sign flips induced by HAM compared to $ m \odot  w$ in Figure 2a, indicating that the vanishing inverse metric problem is mitigated. Note that HAM also converges faster than dense training without regularization in practice, as shown in Figure 9 in the appendix. This can be attributed to having a larger inverse metric for each value $ \theta \in \mathbb{R}^n$ as seen in Figure 1.
>
> **W1. Implicit bias towards sparsity.**
> To motivate the implicit sparsity bias experimentally we conduct the following experiment to illustrate the difference between training with and without HAM. We take the Imagenet trained ResNet50 model with val acc 77.64 for HAM and 76.92 for the baseline. We prune the reported amount of low magnitude weights for various sparsity levels and report the resulting validation accuracy (see table below). We observe that HAM’s validation accuracy drops by a smaller percentage when pruning more parameters indicating that HAM indeed induces a sparsity bias.
>
> | Sparsity | Pruned HAM (Test Accuracy) | Dense (Test Accuracy) |
> | :--- | :--- | :--- |
> | 0.9 | 1.384 | 0.104 |
> | 0.8 | 59.008 | 0.424 |
> | 0.7 | 71.94 | 37.776 |
> | 0.5 | 76.824 | 73.404 |
>
> One shot pruning at the end of dense training for HAM vs SGD.
>
>
> **W2. Hyperparameter tuning:**
> As most deep learning optimizers, HAM requires hyperparameter tuning. (The results of our grid search to tune alpha and beta for a dense ResNet50 on ImageNet are reported in Appendix G.)
> However, note that these values appear stable across different scenarios and serve therefore as strong default parameters, reducing the problem to a one time effort. Note that we used the same values across all ImageNet experiments, dense and sparse, and including Vision Transformers trained with AdamW.
>
> Generally, we have observed that performance improvements remain relatively robust for a large range of $\alpha$ values, as seen in Fig 11 and 12.
> Identifying a good $\beta$ is important to control the model capacity similar to weight decay and therefore may vary between tasks. To transfer the parameters between different neural network architectures, the tensor program was proposed. These indeed explain why $\alpha$ and $\beta$ are expected to be stable between architectural choices. The reason is that they are multiplied by the learning rate $\eta$, which fully captures the architectural change (and tensor program adjustments have been derived for it).
>
> Furthermore, in comparison with other (sparse) optimization methods, HAM usually requires less hyperparameter tuning. For example, the method Sign-In needs to tune the frequency of rescaling the reparameterization and the rescaling constant. Continuous sparsification with STR is highly sensitive to the weight decay strength and the threshold parameter.

---

> ### Author Response · Authors · 2025-11-20
> **Part 2**
>
> Q1: As detailed in W2, beta can vary with the model choice, while alpha is more stable. Yet, beta transfers to different models such as for the DeIT. We argue that adjusting the learning rate according to the proposed principles would be sufficient, as beta defines a multiplicative factor that adapts the effective learning rate.
>
> Q2: The current version of the HAM algorithm considers the gradient for any optimizer, which has shown to be effective in practice. For example we test HAM with AdamW on a DeIT (Table 8 and 9). However, it can be extended to include other terms like momentum, which we leave for future work.
>
> [1] Gadhikar, Advait et al. “Sign-In to the Lottery: Reparameterizing Sparse Training From Scratch.” (NeurIPS,2025)

---

### Official Review · Reviewer_PXeq · 2025-11-04

**Soundness:** 3
**Presentation:** 3
**Contribution:** 3
**Rating:** 8
**Confidence:** 3

**Summary:**

- This paper studies the training of sparse networks, specifically those which learn the mask as $\theta = m \odot w$
- This paper builds on previous work which identify a limitation of these models in learning sign flips due to the small inverse Riemannian metric as $m$ or $w$ approaches zero
- Motivated by the reparameterized gradient flow of these masked models, the authors propose interleaving hyperbolic updates between regular gradient steps.
- The hyperbolic updates inherit the favourable sparsity inducing dynamics from training masked models, while the regular gradient steps handle sign flips
- Empirically, HAM outperforms methods which only use the base optimizer on ImageNet and CIFAR-100 on both dense and sparse training

**Strengths:**

- Clear motivation and presentation
- The method is compatible with existing optimizers (such as SAM), and requires little computational overhead
- Strong empirical results - with the method surprisingly also improving dense training
- Thorough theoretical analysis

**Weaknesses:**

- The method additionally adds two hyperparameters ($\alpha$, $\beta$). Based on figure 8 and 9 in the appendix, the performance of HAM is quite sensitive of these hyperparameters, and optimal hyperparameters seems to differ between models, making it difficult to tune
- It seems that although HAM is motivated from the perspective of sparse training, it doesn't actually improve sparse training directly without pairing it with another method (remark 4.7 and 4.8). Given this, the motivation behind HAM seems less clear

**Questions:**

- I'm generally interested in evaluating whether the regular optimization step is strictly to facilitate sign flips, or offers more performance beyond that. To this end, I was wondering, (1) what is the effect of only the hyperbolic step, and (2) what is the effect of only applying the regular gradient step for small magnitude weights (i.e. ones whose signs could be flipped).
- What kind of sparsity does HAM trained models get (without combining with any sparse training). I.e. is direct magnitude pruning at the end of initialization more successful with HAM than with standard training due to HAM's implicit bias to sparse networks?
- Can the authors offer any intuition why HAM improves performance in dense training? Given that the motivation of the paper was directed towards sparse training, the improvement in dense training is surprising.

---

> ### Author Response · Authors · 2025-11-20
> **Part 1**
>
> We would like to express our gratitude for your time and efforts in providing valuable comments on our manuscript. Below, we elaborate on your concerns in a detailed point-by-point response. In case of any open questions, we would be happy to discuss them.
>
> **Response to Weaknesses**
>
> **W1. Hyperparameter tuning.**
> As most deep learning optimizers, HAM requires hyperparameter tuning. (The results of our grid search to tune alpha and beta for a dense ResNet50 on ImageNet are reported in Appendix G.)
> However, note that these values appear stable across different scenarios and serve therefore as strong default parameters, reducing the problem to a one time effort. Note that we used the same values across all ImageNet experiments, dense and sparse, and including Vision Transformers trained with AdamW.
>
> Generally, we have observed that performance improvements remain relatively robust for a large range of $\alpha$ values, as seen in Fig 11 and 12.
> Identifying a good $\beta$ is important to control the model capacity similar to weight decay and therefore may vary between tasks. To transfer the parameters between different neural network architectures, Reviewer vz7i proposed to use tensor programs. These indeed explain why $\alpha$ and $\beta$ are expected to be stable between architectural choices. The reason is that they are multiplied by the learning rate $\eta$, which fully captures the architectural change (and tensor program adjustments have been derived for it).
>
> Furthermore, in comparison with other (sparse) optimization methods, HAM usually requires less hyperparameter tuning. For example, the method Sign-In needs to tune the frequency of rescaling the reparameterization and the rescaling constant. Continuous sparsification with STR is highly sensitive to the weight decay strength and the threshold parameter.
>
>
> **W2. Motivation of HAM.**
> HAM is a general purpose optimizer that can improve any dense or sparse training. Most sparse training methods (like ACDC and STR) consist of both dense and sparse training phases and are therefore compatible with HAM. HAM itself also has an implicit sparsity bias and therefore tends to regularize a deep neural network towards higher sparsity as illustrated in Q2. Furthermore it also aids the optimization by promoting complementary sign flips, these are especially beneficial for sparse training. Pruning at Initialization experiments (see Table 4) highlight this effect most clearly, as the parameters are trained given a fixed sparse mask.

---

> ### Author Response · Authors · 2025-11-20
> **Part 2**
>
> **Q1. Mechanisms of HAM’s performance.**
> Figure 2a shows for random pruning at initialization that HAM induces more sign flips during training then training without it (baseline). This does not imply that all sign flips are meaningful or that this is the only mechanism that allows HAM to boost performance. However, our theoretical investigations in Appendix E show that exactly this sign flip mechanism is critical for successful optimization in the studied case.
> The hyperbolic step on its own cannot induce sign flips, as it is just a multiplication with a positive constant at every step. This insight is stated in Corollary B.2 in the appendix. We illustrate this with an additional experiment in the context of linear regression in Appendix G.1. The table below reports the mean squared error (distance to ground truth) for a standard problem with HAM and the hyperbolic step.
>
> The final ground truth distances are given in the table below. Figure 7 shows that HAM comes close to the ground truth. The hyperbolic step alone diverges due to exploding gradients, while the gradient step alone ends up with a higher error than HAM, struggling with sign flips. For that reason, HAM relies on the combination of both steps.
> | Update        | Distance to ground truth                       |
> |---------------|------------------------------|
> | HAM    | 0.026        |
> | Hyp    | diverges                         |
> | GD    |    2.7 |
>
> In the second requested experiment, we train a ResNet50 on ImageNet by only choosing a fraction of the smallest weights and training them with SGD and find that training only the smallest fraction is not sufficient to achieve a competitive performance.
>
> |Dense baseline:| 76.92|
> | :--- | :--- |
> | 50% small weights | 73.14 |
> | 20% small weights | 69.04 |
>
> HAM can make faster progress, as it can update all parameters - also the large ones. Large weights can still move effectively. They just need less support to transition through 0. Nevertheless, if our goal is to only finetune a model, updating only the smallest weight can be an efficient way, as suggested by the reviewer and also shown by [3].
>
>
> **Q2. Implicit sparsity bias.** We conduct the following experiment to illustrate the difference in obtained sparsity between training with and without HAM. We take the Imagenet trained ResNet50 model which achieves accuracy, 77.64 for HAM and 76.92 for the baseline. Next, we prune by magnitude for various sparsity levels and report the resulting validation accuracy (without further training) (see table below). We observe that HAM’s validation accuracy stays higher when pruning more parameters indicating that HAM indeed induces a sparsity bias.
>
> |Sparsity | Pruned HAM (Test Accuracy) | Pruned SGD (Test Accuracy) |
> | :--- | :--- | :--- |
> | 0.9 | 1.384 | 0.104 |
> | 0.8 | 59.008 | 0.424 |
> | 0.7 | 71.94 | 37.776 |
> | 0.5 | 76.824 | 73.404 |
> One shot pruning at the end of dense training for HAM vs SGD.
>
> **Q3. Intuition behind generalization.**
> As discussed in our answer to W2, we identify two main mechanisms that allow HAM to improve dense training: a) implicit sparsity bias that regularizes against overfitting [2] and also illustrated for Linear regression in Appendix G.1, b) complementary sign flips to dense training.
> In Appendix E we consider a one neuron toy example with either multi and one dimensional input corresponding to dense and sparse training.
> HAM, similar to Sign-In, is able to flip the sign of the $a$ parameter; this is complementary to what dense training can accomplish on its own.
> We have updated the single neuron example as well with a figure that highlights these effects in the Appendix (see Fig 4).
>
> [1] Gadhikar, Advait et al. “Sign-In to the Lottery: Reparameterizing Sparse Training From Scratch.” (NeurIPS, 2025)
>
> [2] Jin, Tian, et al. "Pruning’s effect on generalization through the lens of training and regularization." (NeurIPS, 2022)
>
> [3] Zhou, Chao, et al. "Pay Attention to Small Weights." (NeurIPS 2025).

---

### Author Response · Authors · 2025-11-20
**General Response**

We would like to thank all the reviewers for their efforts in reviewing our work and providing valuable comments. We have incorporated the comments of all reviewers and updated our draft accordingly, which we believe makes our work stronger. To make it easy for the reviewers to go over these changes, the updated text is coloured blue.

**Rebuttal summary:**

We have proposed a novel, general purpose optimizer HAM that improves both dense and sparse training.
HAM relies on two key mechanisms for improved training: (i) HAM promotes parameter sign flipping during training and (ii) induces sparsity according to its implicit bias, as substantiated by our theoretical analysis and extensive experiments.

Reviewers highlighted the following strengths of our paper:
1) Clear motivation and presentation. (FKu7, q7Do, PXeq)
2) Novelty of our proposed optimizer HAM. (FKu7, q7Do, vz7i, qGRF, PXeq)
3) Strong theoretical analysis of its underlying mechanisms. (FKu7, q7Do, vz7i, qGRF, PXeq)
4) Our empirical results are in line with our theoretical analysis. (FKu7, q7Do, vz7i, qGRF, PXeq)
5) HAM can be combined with any first order optimizer and other complementary optimization principles (like SAM). (PXeq)

We address common concerns shared by multiple reviewers here and provide a point-by-point response to all the concerns in individual replies to each reviewer.

**Scope of HAM:**

We have clarified in the introduction and discussion that HAM is a general purpose optimization step that can improve both dense and sparse training by promoting sign flips and sparsity.

Sign flipping has been first identified as a severe challenge in the context of sparse training, where the problem is more pronounced. Baselines to mitigate this issue (i.e. pointwise overparameterization $m \odot w $ as in Sign-In) have been developed in this context. Sign-In is characterized by two advantages: a) an implicit sparsity bias and b) an ability to promote sign flips. However, Sign-In also has a few limitations, including a) increased memory demands and b) a vanishing inverse metric around 0 that hampers parameter movement. HAM inherits the two advantages while overcoming the two limitations. Accordingly, HAM is designed to boost sparse training and this is its primary purpose. We have provided ample theoretical and empirical evidence that demonstrate that it is able to do so.

HAM can also improve dense training, as a) the implicit sparsity bias can mitigate overfitting (see Lines 42, 408 and also the answer to Q2 of Reviewer PXeq); and b) the promoted sign flips are complementary to the ones facilitated by dense training, as detailed in Appendix E.


**Hyperparameter tuning:**

As most deep learning optimizers, HAM requires hyperparameter tuning. (The results of our grid search to tune alpha and beta for a dense ResNet50 on ImageNet are reported in Appendix G.)
However, note that these values appear stable across different scenarios and serve therefore as strong default parameters, reducing the problem to a one time effort. Note that we used the same values across all ImageNet experiments, dense and sparse, and including Vision Transformers trained with AdamW.

Generally, we have observed that performance improvements remain relatively robust for a large range of $\alpha$ values, as seen in Fig 11 and 12.
Identifying a good $\beta$ is important to control the model capacity similar to weight decay and therefore may vary between tasks. To transfer the parameters between different neural network architectures, Reviewer vz7i proposed to use tensor programs. These indeed explain why $\alpha$ and $\beta$ are expected to be stable between architectural choices. The reason is that they are multiplied by the learning rate $\eta$, which fully captures the architectural change (and tensor program adjustments have been derived for it).

Furthermore, in comparison with other (sparse) optimization methods, HAM usually requires less hyperparameter tuning. For example, the method Sign-In needs to tune the frequency of rescaling the reparameterization and the rescaling constant. Continuous sparsification with STR is highly sensitive to the weight decay strength and the threshold parameter.

---

### Author Response · Authors · 2025-12-03

Dear Area Chair, Senior Area Chair, and Program Chairs,

Thank you for your time and effort in reviewing our paper and the rebuttal. We would like to summarize our work's main contribution and the discussion of the review process.


**Summary of Contribution**

 We have introduced Hyperbolic Aware Minimization (HAM), a simple and general optimization step that improves both dense and sparse training. We have provided theoretical insights into the mechanisms that underlie HAM’s success: (1) enabling parameter sign flips complementary to dense training, which is especially important for sparse training, (2) inducing an implicit sparsity bias that serves as a regularizer, which is especially important for dense training phases (see also experiment for Reviewer PXeq Q2). This makes HAM a broadly applicable, theoretically grounded, and practically effective addition to modern optimizers.


**Summary of Strengths**

Reviewers consistently highlighted the novelty of the method (all reviewers), the clarity of the presentation (FKu7, q7Do, PXeq), the strength and coherence of the theoretical analysis (all reviewers), and the empirical results that align closely with the theory (all reviewers). They also appreciated HAM’s simplicity (PXeq), its compatibility with common optimizers such as SGD, AdamW, and SAM, and its wide applicability to both dense and different forms of sparse training.


**Two Main Weaknesses Addressed**

 Two issues emerged as the most central:
1) **Scope (dense vs. sparse)**. We clarified that HAM is designed as a general optimization step, not only as a sparse training method. Sparse training has motivated certain design choices, but dense training benefits from both the implicit sparsity bias and the complementary sign flip mechanism.

2) **Hyperparameter tuning**. We demonstrated that HAM’s hyperparameter $\alpha$ is stable across architectures, sparsity levels, and optimizers (Figs. 11 and 12). The other hyperparameter $\beta$ has a similar role as weight decay, i.e. controlling the model capacity. We found that it transfers relatively well between architectures and tasks. This is in line with the theory of Tensor Programmes, which was suggested by Reviewer  vz7i. The reason is that the hyperparameters are multiplied by the learning rate, which is already tuned to the task and model. Furthermore, we have argued that sparse training methods incur more hyperparameter tuning in general and are usually more unstable with respect to their hyperparameters.

We believe both issues were fully resolved in the revision by providing additional textual clarifications (in blue) in the updated draft, which even convinced our most critical reviewer (who increased their score from 4 to 6).

---

### Meta-Review · Area_Chair_Bm5G · 2026-01-06

**Summary:**

This paper introduces a Hyperbolic Aware Minimization (HAM) that improves both dense and sparse training.
The reviewers’ primary concerns initially centered on the specific scope of the proposed Hyperbolic Aware Minimization (HAM), specifically questioning whether it was strictly a sparse training method or applicable to dense settings. Additionally, several reviewers raised issues regarding the practicality of the method with respect to the additional hyperparameters. Despite these reservations, the reviewers uniformly recognized the novelty of the approach and the strength of the theoretical analysis regarding sign flipping and implicit bias.

**Reviewer Concerns:**

The rebuttal successfully addressed the confusion regarding the method's scope by clarifying that the sign-flipping and regularization mechanisms of HAM are also beneficial for general optimization. The concern regarding hyperparameter sensitivity was also effectively resolved through additional experiments. It appears there are no significant outstanding concerns.

**Reviewer Scores:**

The paper originally received scores of 8, 6, 6, 6, and 4. During the rebuttal phase, the reviewer who initially assigned a 4 explicitly indicated to increase their score to 6. Consequently, had all reviewers participated fully in the discussion, they would have reached a consensus for a positive recommendation.

---

### Decision · Program_Chairs · 2026-01-26

Accept (Poster)